**Water Masses in the Atlantic Ocean: Characteristics and Distributions**

Mian Liu[1, 2]

Toste Tanhua[2, *]

*[1] College of Ocean and Earth Sciences,*

*Xiamen University, Xiamen, 361005, China*

*[2] GEOMAR Helmholtz Centre for Ocean Research Kiel,*

*Marine Biogeochemistry, Chemical Oceanography*

*Düsternbrooker Weg 20, 24105 Kiel, Germany*

*Correspondence to: T. Tanhua (ttanhua@geomar.de)*

**Abstract:** A large number of water masses are presented in the Atlantic Ocean and knowledge of their distributions and properties are important for understanding and monitoring of a range of oceanographic phenomena. The characteristics and distributions of water masses in biogeochemical space are useful for, in particular, chemical and biological oceanography to understand the origin and mixing history of water samples. Here we define the characteristics of the major water masses in the Atlantic Ocean as Source Water Types (SWTs) from their formation areas, and map out their distributions. The SWTs are described by six properties taken from the biased adjusted data product GLODAPv2, including both conservative (Conservative Temperature and Absolute Salinity) and non-conservative (oxygen, silicate, phosphate and nitrate) properties. The distributions of these water masses are investigated with the use of the Optimal Multi-Parameter (OMP) method and mapped out. The Atlantic Ocean is divided into four vertical layers by distinct neutral densities and four zonal layers to guide the identification and characterization. The water masses in the upper layer originate from winter-time subduction and are defined as Central Waters. Below the upper layer, the intermediate layer consists of three main water masses; Antarctic Intermediate Water (AAIW), Subarctic Intermediate Water (SAIW) and Mediterranean Water (MW). The North Atlantic Deep Water (NADW, divided into its upper and lower components) is the dominating water mass in the deep and overflow layer. The origin of both the upper and lower NADW is the Labrador Sea Water (LSW), the Iceland-Scotland Overflow Water (ISOW) and the Denmark Strait Overflow Water (DSOW). The Antarctic Bottom Water (AABW) is the only natural water mass in the bottom layer and this water mass is redefined as North East Atlantic Bottom Water (NEABW) in the north of equator due to the change of key properties, especial silicate. Similar with NADW, two additional water masses, Circumpolar Deep Water (CDW) and Weddell Sea Bottom Water (WSBW), are defined in the Weddell Sea region in order to understand the origin of AABW.

**Key Words:** Atlantic Ocean, Water Mass, Source Water Types, GLODAP, Optimal-Multi-Parameter Analysis

# 1 Introduction

The ocean is composed a large number of water masses without clear boundaries but gradual transformations between each other (e.g. Castro et al., 1998). Properties of the water in the ocean are not uniformly distributed and the characteristics vary with regions and depths (or densities). The water masses, which are defined as bodies of water with similar properties and common formation history, are referred to as a body of water with a measurable extent both in the vertical and horizontal, and thus a quantifiable volume (e.g. Helland-Hansen, 1916; Montgomery, 1958). Mixing occurs inevitably between water masses, both along and across density surfaces, and result in mixtures with different properties away from their formation areas. Understanding of the distributions and variations of water masses have significance to several disciplines of oceanography, for instance while investigating the thermohaline circulation of the world ocean or predicting climate change (e.g. Haine and Hall, 2002; Tomczak and Godfrey, 2013; Morrison et al., 2015).

The concept of water masses is also important for biogeochemical and biological applications, where the transformations of properties over time can be successfully viewed in the water masses frame-work. For instance, the formation of Denmark Strait Overflow Water in the Denmark Strait was described using mixing of a large number of water masses from the Arctic Ocean and the Nordic Seas (Tanhua et al., 2005). A number of investigations show the significance of knowledge about water masses to the biogeochemical oceanography, for instance the investigation of mineralization of biogenic materials (Alvarez et al., 2014), or the change of ventilation in the Oxygen Minimum Zone (Karstensen et al., 2008). In a more recent work, Garcia-Ibanez et al. (2015) considered 14 water masses combined with velocity fields to estimate transport of water masses, and thus chemical constituents, in the north Atlantic. Similarly, Jullion et al. (2017) used water mass analysis in the Mediterranean Sea to better understand the dynamics of dissolved Barium. However, the lack of a unified definition of overview water masses on a~~n~~ basin or global scale leads to additional and repetitive amount of work by redefining water masses in specific regions. The goal of this study is to facilitate water mass analysis in the Atlantic Ocean and in particular, we aim at supporting biogeochemical and biological oceanographic work in a broad sense.

Understanding the formation, transformation, and circulation of water masses has been a research topic in oceanography since the 1920s (e.g. Jacobsen, 1927; Defant, 1929; Wüst and Defant, 1936; Sverdrup, 1942 etc.). The early studies were mainly based on (potential) temperature and (practical) salinity as summarized by Emery and Meincke (1986). The limitation of the analysis based on T—S relationship is obvious; distributions of more (than three) water masses cannot be analyzed at the same time with only these two parameters, so physical and chemical oceanographers has worked to add more parameters to the characterization of water masses (e.g. Tomczak and Large, 1989; Tomczak, 1981; 1999). The Optimum Multi-parameter (OMP) method extends the analysis so that more water masses can be considered by adding parameters/water properties (such as phosphate and silicate) and solving the equations of linear mixing without assumptions. The OMP analysis has been successfully applied in a

range of studies, for instance for the analysis of mixing in the thermocline in Eastern Indian Ocean
(Poole and Tomczak, 1999).
An accurate definition and characterization is the prerequisite for the analysis of water masses. In this
study, the concepts and definitions of water masses given by Tomczak (1999) are used and we seek to
define the key properties of the main water masses in the Atlantic Ocean and to describe their
distributions. In order to facilitate the analysis, the data product GLODAPv2 is used to identify and
define the characteristics of the most prominent water masses based on 6 commonly measured physical
and biogeochemical properties (Figure 1). The water masses are defined in a static sense, i.e. they are
assumed to be steady and not change over time, and subtle differences between closely related water
masses are not considered in this basin-scale focused study. The so defined water masses are in a
subsequent step used to estimate their distributions in the Atlantic Ocean, again based on the
GLODAPv2 data product. Detailed investigations on temporal variability of water masses, or their
detailed formation processes, for instance, may find this study useful but will certainly want to use a
more granular approach to water mass analysis in their particular areas.
**2      Data and Methods**
**2.1      The GLODAPv2 data product**
Oceanographic surveys conducted by different countries have been actively organized and coordinated
since late 1950s. WOCE (the World Ocean Circulation Experiment), JGOFS (Joint Global Ocean Flux
Study) and OACES (Ocean Atmosphere Carbon Exchange Study) are three typical representatives of
international coordination in the 1990s. The GLODAP (Global Ocean Data Analysis Project) data
product was devised and implemented in this context with the aim to create a global dataset suitable to
describe the distribution and interior ocean inorganic carbon variables (Key et al., 2004; 2010). The first
edition (GLODAPv1.1) contains data up to 1999 whereas the updated and expanded versions
GLODAPv2 (Key et al., 2015; Olsen et al., 2016) was published in 2016 and the GLODAP team is
striving for annual updates (Olsen et al., 2019; 2020).  Since GLODAPv2 is a comprehensive and, more
importantly, biased adjusted data product, this is used to quantify the characteristics of water masses.
The data in the GLODAPv2 product has passed both a primary quality control (QC), aiming at precision
of the data and unify the units, and a secondary quality control, aiming at the accuracy of the data
(Tanhua et al., 2010). The GLODAPv2 data product is adjusted to correct for any biases in data through
these QC routines and is unique in its internal consistency, and is thus an ideal product to use for this
work. Armed with the internally consistent data in GLODAPv2, we utilize previously published studies
on water masses and their formation areas to define areas and depth/density ranges that can be
considered to be representative samples of a water masses.
The variables of Absolute Salinity (SA in g kg$^{-1}$), Conservative Temperature (CT in °C) and neutral
density ($\gamma$ in kg m$^{-3}$), which consider the thermodynamic properties such as entropy, enthalpy and
chemical potential (Jackett et al., 2006; Groeskamp et al., 2016), are used in this study because they
systematically reflects the spatial variation of seawater composition in the ocean, as well as the impact
from dissolved neutral species on the density and provides a more conservative, actual and accurate
description of seawater properties (Millero et al., 2008; Pawlowicz et al., 2011; Nycander et al., 2015).
**2.2 Water Masses (WMs) and Source Water Types (SWTs)**
In practice, defining properties of water masses (WMs) is often a difficult and time-consuming part,
particularly when analyzing water masses in a region distant from their formation areas. Tomczak (1999)
defined a water mass as "a body of water with a common formation history, having its origin in a
particular region of the ocean" whereas Source Water Types (SWTs) describe "the original properties
of water masses in their formation areas". The distinction between the WMs and SWTs is that WMs
define physical extents, i.e. a volume, while SWTs are only mathematical definitions, i.e. SWTs are
defined values of properties without physical extents. Knowledge of the SWTs, on the other hand, is
essential in labeling WMs, tracking their spreading or mixing progresses, since the values from SWTs
describe their initial characteristics and can be considered as the fingerprints of WMs. The SWT of a
WM is defined by the values of key properties, while some of them, like Central Waters, require more
than one SWT to be defined (Tomczak, 1999). In this study, the terminology "water mass" is used in
the discussions, realizing that the properties of the WMs used for the further analysis actually refer to
SWTs.
**2.3 OMP Analysis**
**2.3.1 Principle of OMP Analysis**
For the analysis, six key properties are used to define SWTs, including two conservative (Conservative
Temperature and Absolute Salinity) and four non-conservative (oxygen, silicate, phosphate and nitrate)
properties. In order to determine the distributions of WMs, the OMP analysis is invoked as objective
mathematical formulations of the influence of mixing (Karstensen and Tomczak, 1997; 1998). The
starting point is the 6 key properties (Figure 1) from observations (such as $CT_{obs}$ is the observed
Conservative Temperature). The OMP model determines the contributions from predefined SWTs (such
as $CT_i$ that describes the Conservative Temperature in each SWT), which represent the values of the
"unmixed" WMs in the formation areas, through a linear set of mixing equations, assuming that all key
properties of water masses are affected similarly by the same mixing processes. The fractions ($x_i$) in
each sampling point are obtained by finding the best linear mixing combination in parameter space
defined by 6 key properties and minimizing the residuals (R, such as $R_{CT}$ is the residual of Conservative
Temperature) in a nonnegative least squares sense (Lawson and Hanson, 1974) as shown in the
following equations:
$x_1CT_1 + x_2CT_2 + \ldots\ldots + x_nCT_n = CT_{obs} + R_{CT}$
$x_1SA_1 + x_2SA_2 + \ldots\ldots + x_nSA_n = SA_{obs} + R_{SA}$
$\quad x_1O_1 \ + x_2O_2 \ + \ldots\ldots + x_nO_n \ \ = O_{obs} \ + R_O$
$\quad x_1Si_1 \ + x_2Si_2 \ + \cdots\cdots + x_nSi_n \ = Si_{obs} \ + R_{Si}$
$\quad x_1Ph_1 + x_2Ph_2 \ + \ldots\ldots + x_nPh_n = Ph_{obs} \ + R_{Ph}$
$\quad x_1N_1 \ + x_2N_2 \ \ + \ldots\ldots + x_nN_n \ \ = N_{obs} \ + R_N$
$\quad\quad x_1 \ + \ \ x_2 \ + \ldots\ldots + \ \ x_n \ \ = \ \ 1 \ \ + R$
Where the $CT_{obs}$, $SA_{obs}$, $O_{obs}$, $Si_{obs}$, $Ph_{obs}$ and $N_{obs}$ are the observed values of properties, the $CT_i$, $SA_i$, $O_i$,
$Si_i$, $Ph_i$ and $N_i$ ($i = 1, 2 \ldots, n$) represent the predetermined (known) values in each SWT for each property.
The last row expresses the condition of mass conservation.
OMP analysis represents an inversion of an overdetermined system in each sampling point, so that the
sampling points are required to be located "downstream" from the formation areas, i.e. on the spreading
pathway. The total number of WMs which can be analyzed simultaneously within one OMP run is
limited by the number of variables/key properties, because mathematically, 6 variables ($x_1 - x_6$) can be
solved with 6 equations. In our analysis, one OMP run can solve up to 6 WMs. The above system of
equations can be written in matrix notation as:
$\mathbf{G \cdot x - d = R;}$
Where $\mathbf{G}$ is a parameter matrix of defined SWTs with 6 key properties, $\mathbf{x}$ is a vector containing the
relative contributions from the "unmixed" water masses to the sample (i.e. solution vector of the SWT
fractions), $\mathbf{d}$ is a data vector of water samples (observational data from GLODAPv2 in this study) and
R is a vector of residual. The solution is to find the minimum of the residual ($\mathbf{R}$) with a linear fit of
parameters (key properties) for each data point with a non-negative values. In this study, the mixed layer
is not considered as its properties tend to be strongly variable on seasonal time-scales so that water mass
analysis is inapplicable. The solution is dependent on, and sensitive to, the prior assumptions of the
properties of the SWTs. Here we have not explicitly explored this sensitivity, but note that a common
difficulty in OMP analysis is to properly define the SWT properties, and that this study provides a
generally applicable set of SWT properties for the major water masses in the Atlantic Ocean.
**2.3.2 Extended OMP Analysis**
The prerequisite (or restriction) for using (basic) OMP analysis is that the water masses are formed close
enough to the water samples with short transport times within a limited ocean region, for instance an
oceanic front or intertidal belt, so that the mixing can be assumed not to be influenced by biogeochemical
processes (i.e. assume all the parameters to be quasi-conservative). However, biogeochemical processes
cannot be ignored in a basin-scale analysis (Karstensen and Tomczak, 1998). Obviously, this
prerequisite does not apply to our investigation for the entire Atlantic, so the "extended" OMP analysis
is required. In this concept, non-conservative parameters (phosphate and nitrate) are converted into
conservative parameters by introducing the "preformed" nutrients PO and NO, where PO and NO
denotes the concentrations of phosphate and nitrate in seawater by considering the consumption of
dissolved oxygen by respiration (in other words, the alteration due to respiration is eliminated) (Broecker,
1974; Karstensen and Tomczak, 1998). In addition, a new column should be added to the equations for
non-conservative properties ($a\Delta O_2$, $a\Delta Si$, $a\Delta Ph$ and $a\Delta N$) to express the changes in SWTs due to
biogeochemical impacts, namely, the change of oxygen concentration with the remineralization of
nutrients:
$\quad x_1 CT_1 + x_2 CT_2 + \ldots\ldots + x_n CT_n \quad\quad = CT_{obs} + R_{CT}$
$\quad x_1 SA_1 + x_2 SA_2 + \ldots\ldots + x_n SA_n \quad\quad = SA_{obs} + R_{SA}$
$\quad x_1 O_1 \ + x_2 O_2 \ + \ldots\ldots + x_n O_n - a\Delta O_2 \ = O_{obs} \ \ + R_O$
$\quad x_1 Si_1 \ + x_2 Si_2 \ + \ldots\ldots + x_n Si_n \ + a\Delta Si \ = Si_{obs} \ + R_{Si}$
$\quad x_1 Ph_1 + x_2 Ph_2 + \ldots\ldots + x_n Ph_n + a\Delta Ph = Ph_{obs} \ + R_{Ph}$
$\quad x_1 N_1 \ + x_2 N_2 \ + \ldots\ldots + x_n N_n \ + a\Delta N \ = N_{obs} \ + R_N$
$\quad\quad x_1 \ + \ \ x_2 \ + \ldots\ldots + \ \ x_n \quad\quad\quad = \quad 1 \ \ + R$
As a result, the number of water masses should be further reduced in one OMP run if the biogeochemical
processes are considered and extended OMP analysis is used. In this study, a total number of 5 water
masses are included in each OMP run.

### 2.3.3 Presence of mass residual

The fractions of WMs in each sample are obtained by finding the best linear mixing combination in
parameter space defined by 6 key properties which minimizes the residuals (R) in a non-negative least
squares sense. Ideally, a value of 100% is expected when the fractions of all the water masses are added
together. However, mass residuals where the sum of water masses for a sample differ from 100%, ~~is~~ are
inevitable during the analysis and ~~is~~ are due to sample properties outside the input SWTs to the OMP
formulation. There are two different cases. The first is that one single water mass is larger than 100%
and other water masses are all 0%. This mostly happens in the Central Waters ($\gamma < 27.10$ kg m$^{-3}$, Figure
2). The reason is that key properties, for instance CT, of Central Waters are variable. When the CT
increases beyond the range of this water mass, the OMP analysis considers the fraction is over 100%.
In this case, all such samples are set to 100% after confirming the absence of any other water mass. The
second case is that none of each single water mass is more than 100%, but the total fraction is more than
100% when added together. In this study, the total fractions are generally less than 105% ($\gamma > 27.10$ kg
m$^{-3}$, Figure 2).
In order to map the distributions of water masses, all GLODAPv2 data in the Atlantic Ocean (below the
mixed layer) are analyzed with the OMP method by using 6 key properties. In order to solve the
contradiction between the limitation of water masses in one OMP run and the total number of 16 water
masses (Figure 3), the Atlantic Ocean is divided into 17 regions (Table 1) and each with its own OMP
formulation, by only including water masses that are likely to appear in the area. In the vertical, neutral
density intervals are used to separate boxes. In the horizontal direction, the division lines are 40 °N, the
equator and 50°S where the area south of 50 °S is one region, independent of density, and additional
divisions are set between equator and 40 °N ($\gamma$ at 26.70 and 27.30 kg m$^{-3}$, latitude of 30 °N, Table 1). In
this way, we end up with a set of 17 different OMP formulations that are used for estimating the
fraction(s) of water masses in each water sample. The neutral density and the latitude of the water sample
are thus used to determine which OMP should be applied (Table 1). Note that all water masses are
present in more than one OMP so that reasonable (i.e. smooth) transitions between the different areas
can be realized.
**3      Overview of the water masses in the Atlantic Ocean and the Criteria of Selection**
In line with the results from Emery and Meincke (1986) and from our interpretation of the observational
data from GLODAPv2, the water masses in the Atlantic Ocean are considered to be distributed in four
main isopycnal (vertical) layers separated by surfaces of equal (neutral) density (Figure 4). The upper
(shallowest) layer with lowest neutral density is located within upper ~500—1000 m of the water column
(below the mixed layer and $\gamma < 27.10$ kg m$^{-3}$). The intermediate layer is located between ~1000 and 2000
m ($\gamma$ between 27.10 and 27.90 kg m$^{-3}$). The deep and overflow layer occupies the layer between ~2000—
4000m ($\gamma$ between 27.90 and 28.10 kg m$^{-3}$) whereas the bottom layer is the deepest layer and mostly
located below ~4000 m ($\gamma > 28.10$ kg m$^{-3}$).
To define the main water masses in the Atlantic Ocean, the determination of their formation areas is the
first step (Figure 5) and then the selection criteria are listed to define SWTs based on the CT—SA
distribution, pressure (P) or neutral density ($\gamma$) (Table 2). For some SWTs, additional properties such as
oxygen or silicate are also required for the definition. With these criteria, which are taken from the
literature and also based on data from GLODAPv2 product, the SWTs of all the main water masses can
be defined for further estimating their distributions in the Atlantic Ocean by using OMP analysis.
For the water masses in the upper layer, i.e. the Central Waters, properties cover a "wide" range instead
of a "narrow" point value due to their variations, especially in CT and SA space, i.e. tThe Central Waters
are labeled by two SWTs to identify the upper and lower boundaries of properties (Karstensen and
Tomczak, 1997; 1998). In order to determine these two SWTs, one property is taken as a benchmark
(neutral density in this investigation) and the relationships to the others are plotted to make a linear fit
and the two endpoints are selected as SWTs to label Central Waters (Figure 6).

During the determination of each SWT, two figures are displayed to characterize them, including a) depth profiles of the 6 key properties under consideration (same color coding), and b) bar plots from the distributions of the samples within the criteria (the blue dots in Figure 6 and 7) for a SWT with a Gaussian curve to show the statistics (Figure 7). The plots of properties vs. pressure provides an intuitive understanding of each SWT compared to other WMs in the region. The distributions of properties with the Gaussian curves are the basis to visually determine and confirm the SWT property values and associated standard deviations.

Most water masses maintain their original characteristics away from their formation areas. However, some are worthy to be mentioned as products from mixing of several original water masses (for instance, North Atlantic Deep Water is the product from Labrador Sea Water, Iceland-Scotland Overflow Water and Denmark Strait Overflow Water). Also, characteristics of some water masses changes sharply during their pathways (namely, the sharp drop silicate concentration of Antarctic Bottom Water after passing the equator). As a result, it is advantageous to redefine their SWTs. In order to distinguish such water masses from the other original ones, their defined specific areas are mentioned as "redefining" areas instead of formation areas, because, strictly speaking, they are not "formed" in these areas. The calculated water mass fractions for the Atlantic Ocean data in GLODAPv2 are available ~~in the supplementary material~~at https://www.ncei.noaa.gov/access/ocean-carbon-data-system/oceans/ndp_107/ndp107.html.

# 4 The Upper Layer, Central Waters

The upper layer is occupied by four Central Waters known to be formed by winter subduction with upper and lower boundaries of properties. All values between these boundaries are used to calculate the means and standard deviations (Figure 7 and Figure S1 – S3), and occupies two SWTs in one OMP run.

Central Waters can be easily recognized by their linear CT—SA relationships (Pollard et al., 1996; Stramma and England, 1999). In this study, the upper layer is defined to be located above the neutral density isoline of 27.10 kg m$^{-3}$ (below the mixed layer). The formations and transports of the Central Waters are influenced by the currents in the upper layer and finally form relative distinct bodies of water in both the horizontal and vertical directions (Figure 8). The concept of Mode Water is referred to as the sub-regions of Central Water, which describes the particularly uniform properties of seawater within the upper layer and more refers to the physical properties (such as: CT—SA relationship and potential vorticity). In this study, the unified name "Central Water", which more refers to the biogeochemical properties (Cianca et al., 2009; Alvarez et al., 2014), is used to avoid possible confusion.

## 4.1 Eastern North Atlantic Central Water (ENACW)

The main Central Water in the region east of the Mid-Atlantic-Ridge (MAR) is the East North Atlantic Central Water (ENACW, Harvey, 1982). This water mass is formed in the inter-gyre region during the winter subduction (Pollard and Pu, 1985). One component of the Subpolar Mode Water (SPMW) is

carried south and contributes to the properties of ENACW (McCartney and Talley, 1982). The inter-
gyre region limited by latitudes between 39 and 48 °N and longitudes between 20 and 35 °W (Pollard
et al., 1996) is considered as the formation area of ENACW (Figure 5). Neutral densities of 26.50 and
27.30 kg m$^{-3}$ are selected as the upper and lower boundaries to define the SWT of ENACW (Cianca et
al., 2009; Prieto et al., 2015), which is in contrast to Garcia-Ibanez et al. (2015) that used potential
temperature ($\theta$) as the upper limit. The core of ENACW is located within the upper 500 m of the water
column (Figure 7a) with the iconic linear T-S relationship (Figure 6b) consistent with Pollard et al.
(1996). The main character of ENACW is the large ranges of temperature and salinity and low nutrient
concentrations, especially silicate (Figure 7b).

### 4.2 Western North Atlantic Central Water (WNACW)

Western North Atlantic Central Water (WNACW) is another water mass formed through winter
subduction (Worthington, 1959; McCartney and Talley, 1982) with the formation area at the southern
flank of the Gulf Stream (Klein and Hogg, 1996). In some studies, this water mass is referred to as 18 °C
water since a temperature of around 18 °C is one symbolic feature (e.g. Talley and Raymer, 1982; Klein
and Hogg, 1996). In general, seawater in the Northeast Atlantic has higher salinity than in the Northwest
Atlantic due to the stronger winter convection (Pollard and Pu, 1985) and input of MW (Pollard et al.,
1996; Prieto et al., 2015). However, for the Central Waters, the situation is the opposite. WNACW has
a significantly higher salinity (SA) by ~0.9 g kg$^{-1}$ than ENACW (Table 4). In this study, the work from
McCartney and Talley (1982) is followed and the region 24—37°N, 50—70°W shallower than 500 m
is considered as the formation area (Figure 5). By defining the SWT of WNACW, ~~the~~ neutral density
between 26.20 and 26.70 kg m$^{-3}$ is selected due to the discrete CT—SA distribution outside this range
(Table 2). Besides the linear CT—SA relationship, another property of this water mass is, as the
alternative name suggests, a temperature of around 18 °C, which is the warmest in the four Central
Waters due to the low latitude of the formation area and the impact from the warm Gulf Stream (Cianca
et al., 2009; Prieto et al., 2015). In addition, low nutrient is also a significant property compared to other
Central Waters (Figure S1).

### 4.3 Eastern South Atlantic Central Water (ESACW)

The formation area of ESACW is located in area southwest of South Africa and south of the Benguela
Current (Peterson and Stramma, 1991). In this region the Agulhas Current brings water from the Indian
Ocean (Deruijter, 1982; Lutjeharms and van Ballegooyen, 1988) that mixes with the South Atlantic
Current from the west (Stramma and Peterson, 1990; Gordon et al., 1992). The origin of ESACW can
partly be tracked back to the WSACW, but defined as a new SWT since seawater from Indian Ocean is
added by the Agulhas Current. The mixing region of Agulhas Current and South Atlantic Current (30—
40 °S, 0—20 °E) is selected as the formation area of ESACW (Figure 5). To investigate the properties
of ESACW, results from Stramma and England (1999) is followed and consider 200—700m as the core
of this water mass. For the properties, neutral density ($\gamma$) between 26.00 and 27.00 kg m$^{-3}$ and oxygen
concentration higher than 230 µmol kg$^{-1}$ are used to define ESACW (Table 2). Similar as ENACW,
ESACW also exhibits relative large CT and SA ranges and low nutrient concentrations (especially low
in silicate) compared to the AAIW below. The properties in ESACW are similar to that of WSACW,
although with higher nutrient concentrations due to input from the Agulhas current (Figure S2).

### 4.4    Western South Atlantic Central Water (WSACW)

The WSACW is formed in the region near the South American coast between 30 and 45 °S, where
surface South Atlantic Current brings Central Water to the east (Kuhlbrodt et al., 2007). The WSACW
is formed with little directly influence from other Central Water mass (Sprintall and Tomczak, 1993;
Stramma and England, 1999), while the origin of other Central Waters (e.g. ESACW or ENACW) can
be traced back, to some extent at least, to WSACW (Peterson and Stramma, 1991). This water mass is
a product of three Mode Waters mixed together: the Brazil current brings Salinity Maximum Water
(SMW) and Subtropical Mode Water (STMW) from the north, while the Falkland Current brings
Subantarctic Mode Water (SAMW) from the south (Alvarez et al., 2014). Here we follow the work of
Stramma and England (1999) and Alvarez et al. (2014) that choose the meeting region of these two
currents (25—60 °W, 30—45 °S) as the formation area of WSACW (Figure 5). Neutral density (γ)
between 26.0 and 27.0 kg m$^{-3}$ is selected to define the SWT of WSACW and the requirement of silicate
concentrations lower than 5 µmol kg$^{-1}$ and oxygen concentrations lower than 230 µmol kg$^{-1}$ is also added
(Table 2). WSACW shows the similar hydrochemical properties to other Central Waters such as linear
T-S relationship with large T and S ranges and low concentration of nutrients, especially silicate (Figure
S3).

### 4.5    Atlantic Distribution of Central Waters

Based on the OMP analysis of the GLODAPv2 data product, the physical extent of the Central Waters
can be described over the Atlantic Ocean. The horizontal distributions of four Central Waters in the
upper layer are shown on the maps in Figure 8 and the vertical distributions along selected GO-SHIP
sections are found in Figure 9. Note that the Central Waters are found at different densities, the eastern
variations being denser, so that the there is significant overlap in the horizontal distribution. The vertical
extent of the Central Waters is clearly seen in Figure 9.
The ENACW is mainly found in the northeast part of North Atlantic, near the formation area in the inter-
gyre region (Figure 8). High fractions of ENACW is also found in a band across the Atlantic at around
40 °N, where the core of this water mass is found at close to 1000 m depth in the western part of the
basin (Figure 9).
The WNACW is predominantly found in the western basin of the North Atlantic in a zonal band between
~10 °N and 40 °N (Figure 8). The vertical extent of WNACW is significantly higher in the western basin
with an extent of about 500 meter in the west, tapering off towards the east (Figure 9).

The ESACW is found over most of the South Atlantic, as well as in the tropical and subtropical north Atlantic (Figure 8). The extent of ESACW do reach particular far north in the eastern part of the basin where it is an important component over the Eastern Tropical North Atlantic Oxygen Minimum Zone, roughly south of the Cape Verde Islands. In the vertical direction, the ESACW is located below WSACW (Figure 9).

The horizontal distribution of the WSACW does reach into the northern hemisphere but is, obviously, concentrated in the western basin (Figure 8). In the vertical scale, the WSACW also tends to dominate the upper layer of the South Atlantic above the ESACW (Figure 9).

## 5 The Intermediate Layer

The intermediate water masses have an origin in the upper 500m of the ocean and subduct into the intermediate depth (1000—1500m) during their formation process. Similar to the Central Waters, the distributions of the Intermediate Waters are significantly influenced by the major currents (Figure 10, left panel). The neutral density ($\gamma$) of the Intermediate Waters is in general between 27.10 and 27.90 kg m$^{-3}$ and selected as the definition of Intermediate Layer.

In the Atlantic Ocean, two main intermediate water masses, Subarctic Intermediate Water (SAIW) and Antarctic Intermediate Water (AAIW), are formed in the surface of sub-polar regions in north and south hemisphere, respectively. In addition to AAIW and SAIW, Mediterranean Water (MW) is also considered as an intermediate water mass due to the similarity in density ranges, although the formation history is different (Figure 10).

### 5.1 Antarctic Intermediate Water (AAIW)

The Antarctic Intermediate Water (AAIW) is the main Intermediate Water in the South Atlantic Ocean. This water mass originates from the surface region north of the Antarctic Circumpolar Current (ACC) in all three sectors of the Southern Ocean, in particular in the area east of the Drake Passage in the Atlantic sector (McCartney, 1982; Alvarez et al., 2014), then subducts and spreads northward along the continental slope of South America (Piola and Gordon, 1989).

Based on the work by Stramma and England (1999) and Saenko and Weaver (2001), the region between 55 and 40 °S (east of the Drake Passage) at depths below 100 m is selected as the formation area of AAIW as well as the primary stage during the subduction and transformation (Figure 5). Previous work is considered to distinguish AAIW from surrounding water masses, including SACW in the north and NADW in the deep. Piola and Georgi (1982) and Talley (1996) define AAIW as potential densities ($\sigma_\theta$) between 27.00/27.10 and 27.40 kg m$^{-3}$ and Stramma and England (1999) define the boundary between AAIW and SACW at $\sigma_\theta = 27.00$ kg m$^{-3}$ and the boundary between AAIW and NADW at $\sigma_1 = 32.15$ kg m$^{-3}$. The following criteria are used as selection criteria to define AAIW: neutral density between 26.95 and 27.50 kg m$^{-3}$ and depth between 100 and 300 m. In addition, high oxygen (> 260 µmol kg$^{-1}$) and low temperature (CT < 3.5 °C) are used to distinguish AAIW from Central Waters (WSACW and

ESACW), while the relative low silicate concentration ($< 30$ µmol kg$^{-1}$) of AAIW is an additional
boundary to differentiate AAIW from AABW (Table 2). The AAIW is distributed across most of the
Atlantic Ocean up to ~30 °N and the water mass fraction shows a decreasing trend towards the north
(Kirchner et al., 2009). AAIW is found at depths between 500 and 1200 m (Talley, 1996) with the two
significant characteristic features of low salinity and high oxygen concentration (Figure S4, Stramma
and England, 1999).

### 5.2 Subarctic Intermediate Water (SAIW)

The Subarctic Intermediate Water (SAIW) originates from the surface layer in the western boundary of
the North Atlantic Subpolar Gyre, along the Labrador Current (Lazier and Wright, 1993; Pickart et al.,
1997). This water mass subducts and spreads southeast in the region north of the NAC, advects across
the Mid-Atlantic-Ridge and finally interacts with MW (Arhan, 1990; Arhan and King, 1995). The
formation of SAIW is a mixture of two surface sources: Water with high temperature and salinity carried
by the NAC and cold and fresh water from the Labrador Current (Read, 2000; Garcia-Ibanez et al.,
2015). In Garcia-Ibanez et al. (2015), there are two definitions of SAIW, SAIW$_6$, which is biased to the
warmer and saltier NAC, and SAIW$_4$, which is closer to the cooler and fresher Labrador Current. Here,
only the combination of these two end-members is considered on the whole-Atlantic Ocean scale (Figure
S5).
For defining the spatial boundaries we followed Arhan (1990) and selected the region  between 35 and
55 °W and 50 and 60 °N, i.e. the region along the Labrador Current and north of the NAC as the
formation area of SAIW (Figure 5). Within this area, neutral densities higher than 27.65 kg m$^{-3}$ and CT
higher than 4.5 °C is selected to define SAIW by following Read, (2000). Samples in the depth range
from the MLD to 500 m are considered as the core layer of SAIW, which included the formation and
subduction of SAIW (Table 2).

### 5.3 Mediterranean Water (MW)

The predecessor of the Mediterranean Water (MW) is the Mediterranean Overflow Water (MOW)
flowing out through the Strait of Gibraltar, whose main component is the modified Levantine
Intermediate Water.  This water mass is recognized by high salinity and temperature and intermediate
neutral density in the Northeast Atlantic Ocean (Carracedo et al., 2016). After passing the Strait of
Gibraltar, the MOW mixes rapidly with the overlying ENACW and forms the MW, leading to a sharp
decrease of salinity (Baringer and Price, 1997). In Gulf of Cadiz, the outflow of MW turns into two
branches: One branch continues to the west, descending the continental slope, mixing with surrounding
water masses in the intermediate depth and influence the water mass composition as far west as the
MAR (Price et al., 1993). The other branch spreads northwards along the coast of Iberian Peninsula and
along the European coast and its influence can be observed as far north as the Norwegian Sea (Reid,
1978; 1979). The impact from MW is significant in almost the entire Northeast Atlantic in the
Intermediate Layer (east of the MAR, Figure S6), with high temperature and Salinity but low nutrients
compared to other water masses.
Here we followed Baringer and Price (1997) and  define the SWT of MW by the high salinity (SA
between 36.5 and 37.00 g kg$^{-1}$, Table 2) samples in the formation area west of the Strait of Gibraltar
(Figure 5).

**5.4 Atlantic Distributions of Intermediate Waters**

A schematic of the main currents in the intermediate layer ($\gamma$ between 27.10 and 27.90 kg m$^{-3}$) is shown
in Figure 10 (left panel).
The SAIW is mainly formed north of 30 °N in the western basin by mixing of two main sources, the
warmer and saltier NAC and the colder and fresher Labrador Current and characterized with relative
low CT (< 4.5 °C), SA (< 35.1 g kg$^{-1}$) and silicate (< 11 μmol kg$^{-1}$). The SAIW and MW can be easily
distinguished by the OMP analysis due to significantly different properties. The meridional distributions
of three Intermediate Waters along the A16 section are shown in Figure 11 (upper panel) as well as the
zonal distributions of SAIW and MOW along the A03 section. A "blob" of MW centered around 35°N
can be seen to separate the AAIW from the SAIW in the eastern North Atlantic. The fractions of SAIW
in the western basin are definitely higher (Figure 10, right panel).
The MW enters the Atlantic from Strait of Gibraltar and spreads in two branches to the north and the
west. MW is mainly formed close to its entry point to the Atlantic, near the Gulf of Cadiz, with low
fractions in the western North Atlantic. The distribution of MW can be seen as roughly following the
two intermediate pathways following two branches (Figure 10, left panel): One spreads to the north into
the West European Basin until ~50°N, while the other branch spreads in a westward direction past the
MAR (Figure 11), mainly at latitudes between 30 and 40 °N. The density of MW is higher than SAIW,
and the distributions of the two water masses are complementary in the North Atlantic (Figure 10, right
panel).
The AAIW has a southern origin and is found at slightly lighter densities (core neutral density ~27.20
kg m$^{-3}$, Figure 10, right) compared to SAIW and MW. The AAIW is formed in the region south of 40 °S
where it sinks and spreads to the north at depth between ~1000 and 2000 m with neutral densities
between 27.10 and 27.90 kg m$^{-3}$. The AAIW is the dominate Intermediate Water in the South Atlantic
and it is clear that the AAIW represents a reduction of fractions during the pathway to the north with
only a diluted part to be found the equator and 30 °N (Figures 10 and 11).

**6 The Deep and Overflow Layer**

The Deep and Overflow Waters are found roughly between 2000 to 4000 m with neutral densities
between 27.90 and 28.10 kg m$^{-3}$. These water masses play an indispensable role in the Atlantic
Meridional Overturning Circulation (AMOC). The source region of these waters is confined to the North

Atlantic with their formation region either south of the Greenland-Scotland ridge, or in the Labrador Sea (Figures 5 and 12). The Denmark Strait Overflow Water (DSOW) and the Iceland-Scotland Overflow water (ISOW) originate from Arctic Ocean and the Nordic Seas and enter the North Atlantic through either the Denmark Strait of the Faroe Bank Channel (Figure 12, left panel). In the North Atlantic, these two water masses descend, mainly following the topography meet and mix in the Irminger Basin (Stramma et al., 2004; Tanhua et al., 2005) and form the bulk of the lower North Atlantic Deep Water (lNADW) (Read, 2000; Rhein et al.; 2011). The Labrador Sea Water (LSW) is formed through winter deep convection in the Labrador and Irminger Seas, and makes up the bulk of the upper North Atlantic Deep Water (uNADW). Due to intense mixing processes the LSW, DSOW and ISOW are defined as the water masses in north of 40 °N whereas south of this latitude the presence of the two variations of NADW are considered (Figure 12, right panel).

In south of 40 °N, both variations of the NADW spread south mainly with the Deep Western Boundary Current (DWBC, Figure 12, left panel) (Dengler et al., 2004) through the Atlantic until ~50 °S where they meet the Antarctic Circumpolar Current (ACC). During the southward transport, the NADW also spreads significantly in the zonal direction (Lozier, 2012), so that the distribution of NADW covers mostly the whole Atlantic basin in the Deep and Overflow Layer (Figure 12, right panel). The southward flow of NADW is also an indispensable component of Atlantic Meridional Overturning Circulation (AMOC) (Broecker and Denton, 1989; Elliot et al., 2002; Lynch-Stieglitz et al., 2007).

**6.1 Labrador Sea Water (LSW)**

As an important water mass that contributes to the formation of North Atlantic Deep Water (NADW), Labrador Sea Water (LSW) is predominant in mid-depth (between 1000m and 2500m depth) in the Labrador Sea region (Elliot et al., 2002). This water mass was firstly noted by (Wüst and Defant, 1936) due to its salinity minimum and later defined and named by Smith et al. (1937). The LSW is formed by deep convection during the winter and is typically found at depth with $\sigma_\theta = $ ~27.77 kg m$^{-3}$ (Clarke and Gascard, 1983). Since then the character has been identified as a contribution to the driving mechanism of northward heat transport in the Atlantic Meridional Overturning Circulation (AMOC) (Rhein et al., 2011). The LSW is characterized by relative low salinity (lower than 34.9) and high oxygen concentration (~290 µmol kg$^{-1}$) (Talley & Mccartney, 1982). Another important criterion of LSW is the potential density ($\sigma_\theta$), that ranges from 27.68 to 27.88 kg m$^{-3}$ (Clarke and Gascard, 1983; Gascard and Clarke, 1983; Stramma et al., 2004; Kieke et al., 2006). In the large spatial scale, LSW can be considered as one water mass (Dickson and Brown, 1994), however significant differences of different "vintages" of LSW exist (Stramma et al., 2004; Kieke et al., 2006). In some references, this water mass is also broadly divided into upper Labrador Sea Water (uLSW) and classic Labrador Sea Water (cLSW) with the boundary between them at potential density of 27.74 kg m$^{-3}$ (Smethie and Fine, 2001, Kieke et al., 2006; 2007). The LSW is considered as the main origin of the upper NADW (Talley and Mccartney, 1982; Elliot et al., 2002).

On the basis of the above work, the formation area of LSW is selected to include the Labrador Sea
region between the Labrador Peninsula~~r~~ and Greenland and parts of the Irminger Basin (Figure 5). ~~The~~
N~~n~~eutral density ($\gamma$) between 27.70 to 28.10 kg m$^{-3}$ as well as CT < 4°C are used to define SWT of LSW
(Table 2) by considering Clarke and Gascard (1983) and Stramma and England (1999) with the depth
range of 500-2000m (Elliot et al., 2002). Trademark characteristics of LSW are relative low salinity and
high oxygen. The relatively large spread in properties is indicative of the different "vintages" of LSW,
in particular the bi-modal distribution of density, and partly for oxygen (Figure S7).
**6.2 Iceland-Scotland Overflow Water (ISOW)**
The Iceland Scotland Overflow Water (ISOW) flows close to the bottom from the Iceland Sea to the
North Atlantic in the region east of Iceland, mainly through the Faroe-Bank Channel (Swift, 1984; Lacan
et al., 2004; Zou et al., 2020). ISOW turn into two main branches when passing the Charlie-Gibbs
Fracture Zone (CGFZ), with the first one flowing through the Mid-Atlantic-Ridge, into the Irminger
basin, where it meets and mixes with DSOW (Figure 12). The other branch is transported southward
and mixes with Northeast Atlantic Bottom Water (NEABW) (Garcia-Ibanez et al., 2015). ISOW is
characterized by high nutrient and low oxygen concentration and its pathway closely follows the Mid-
Atlantic-Ridge in the Iceland Basin. The following criteria, Conservative Temperature between 2.2 and
3.3 °C and Absolute Salinity higher than 34.95 g kg$^{-1}$, are used to define the SWT of ISOW, and neutral
density higher than 28.00 kg m$^{-3}$ is added order to distinguish ISOW from LSW in the region west of
MAR (Table 2 and Figure S8).
**6.3 Denmark Strait Overflow Water (DSOW)**
A number of water masses from the Arctic Ocean and the Nordic Seas flows through Denmark Strait
west of Iceland. At the sill of the Denmark Strait and during the descent into the Irminger Sea, these
water masses undergo intense mixing. This overflow water mass is considered as the coldest and densest
component of the sea water in the Northwest Atlantic Ocean and constitute a significant part of the
southward flowing NADW (Swift, 1980). Samples from the Irminger Sea (Figure 5) with neutral density
higher than 28.15 kg m$^{-3}$ (Table 2 and Figure S9) are used for the definition of DSOW (Rudels et al.,
2002; Tanhua et al., 2005).
**6.4. Upper North Atlantic Deep Water (uNADW)**
The uNADW is mainly formed by mixing of ISOW and LSW and considered as a distinct water mass
south of the Labrador Sea as this region is identified as the redefining area of upper and lower NADW
(Dickson and Brown, 1994). The region between latitude 40 and 50 °N, west of the MAR is selected as
the redefining area of NADW (Figure 5) and the criteria of neutral density between 27.85 and 28.05 kg
m$^{-3}$ and CT < 4.0 °C within the depth range from 1200 to 2000 m  (Table 2 and Figure S10) are used to
define the SWT of uNADW (Stramma et al., 2004). As a mixture from LSW and ISOW, the uNADW
obviously inherits many properties from LSW, but is also significantly influenced by the ISOW. The
relative high temperature (~3.3 °C) is a significant feature of the uNADW together with relatively low
oxygen (~280 µmol kg$^{-1}$) and high nutrient concentrations, which is a universal symbol of deep water
(Table 4 and Figure S10).

**6.5. Lower North Atlantic Deep Water (lNADW)**

The same geographic region is selected as the formation area of lNADW (Figure 5). In this region, the
ISOW and DSOW, influenced by LSW, mix with each other and form the lower portion of NADW
(Stramma et al., 2004). Water samples between depths of 2000 and 3000 m with CT higher than ~2.5°C
and neutral densities between 27.95 and 28.10 kg m$^{-3}$ are selected to define the SWT of lNADW (Table
2 and Figure S11).

**6.6. Atlantic Distributions of Deep and Overflow Waters**

The water masses dominate the neutral density interval 27.90 – 28.10 kg m$^{-3}$ in the Atlantic Ocean north
of 40 °N are Labrador Sea Water (LSW), Iceland-Scotland Overflow Water (ISOW) and Denmark Strait
Overflow Water (DSOW). In the region south of 40 °N the upper and lower NADW, considered as
products from these three original overflow water masses, dominate the deep and overflow layer (Figure

537 12).

The LSW is commonly characterized as two variations, "upper" and "classic" although in this study we
consider this as one water mass in the discussion above. Our analysis indicates that the LSW dominates
the North West Atlantic Ocean in the characteristic density range. In Figure 12, we choose to display γ
= 27.95 that corresponds to the main property of the LSW (Kieke et al., 2006; 2007). The LSW spreads
east and southward in the North Atlantic Ocean, but is less dominant in the area west of the Iberian
Peninsula where the presence of MW from the Gulf of Cadiz tends to dominate that density level. Note
that although the LSW is slightly denser than the MW, their density ranges do overlap (Figures 12 and

545 13).

The ISOW is mainly found in the Northeast Atlantic north 40 °N between Iceland and Iberian Peninsula
with core at γ = ~28.05 kg m$^{-3}$. The ISOW is also found west of the Reykjanes Ridge, in the Irminger
and Labrador Seas between the DSOW and LSW (Figure 12 and 13).
The DSOW is mainly found in the Irminger and Labrador Seas as the densest layer close to the bottom
(Figure 11). Our analysis indicates a weak contribution of DSOW also east of the MAR. South of the
Grand Banks the DSOW is already significantly diluted and only low to moderate fractions are found
(Figures 12 and 13).
After passing 40 °N, the upper and lower NADW are considered as independent water masses and
dominate the most of the Atlantic Ocean in this density layer. The map in Figure 12 shows that upper
NADW covers the most area, while the lower NADW is found mainly found in the west region near the
Deep Western Boundary Current (DWBC), especially in South Atlantic. In the vertical view based on
sections (Figure 14), the southward transports of both upper and lower NADW can be seen until ~
50 °S where they meets AABW in the ACC region.
**7. The Bottom Layer and the Southern water masses**
The Bottom Waters are defined as the densest water masses that occupy the lowest layers of the water
column, typically below 4000 m depth and with neutral densities higher than 28.10 kg m$^{-3}$. These water
masses have an origin in the Southern Ocean (Figure 15, left panel) and are characterized by their high
silicate concentrations. The Antarctic Bottom Water (AABW) is the main water mass in the Bottom
Layer (Figure 15, right panel). This water mass is formed in the Weddell Sea region, south of Antarctic
Circumpolar Current (ACC) through mixing of Circumpolar Deep Water (CDW) and Weddell Sea
Bottom Water (WSBW) (van Heuven et al., 2011). After the formation, AABW spreads to the north
across the equator and further northwards until ~40 °N (Figure 16), where a new SWT is redefined as
North East Atlantic Bottom Water (NEABW) due to the drastic change in properties (sharp decrease in
silicate concentration). As the two main sources of AABW, CDW and WSBW are confined to the
Southern Ocean (Figure 15, right panel), so they are referred as the southern water masses and discussed
in this section together with bottom waters.
**7.1. Antarctic Bottom Water (AABW)**
Antarctic Bottom Water (AABW) is the symbolic Bottom Water in the whole Atlantic Ocean. As one
of the important components in Atlantic Meridional Overturning Circulation (AMOC), AABW spreads
northward below 4000m depth, mainly west of Mid-Atlantic-Ridge (MAR, Figure 15, right panel) and
plays a significant role in the Thermohaline Circulation (Rhein et al., 1998; Andrié et al., 2003). The
origin of AABW in Atlantic section can be traced back to the Weddell Sea as a product of mixing of
Weddell Sea Bottom Water (WSBW) and Circumpolar Deep Water (CDW) (Foldvik and Gammelsrod,
1988; Alvarez et al., 2014).
The definition of AABW is all water samples formed south of the Antarctic Circumpolar Current (ACC),
i.e. south of 63 °S in the Weddell Sea (Figure 5), with neutral density ($\gamma$) larger than 28.27 kg m$^{-3}$ (Weiss
et al., 1979; Orsi et al., 1999). As an additional constraint, AABW is defined as water samples with
silicate higher than 120 µmol kg$^{-1}$ to distinguish from other water masses in this region as high silicate
is a trade mark property of AABW (Table 2).
The formation process of AABW is a mixture of another two original water masses, CDW and WSBW,
which are referred to as southern water masses, in the Weddell Sea region, consistent with Orsi et al.
(1999) and van Heuven et al. (2011). The CDW, with relative warm temperature (CT > 0.4 °C), is
advected with the ACC from the north, while the extremely cold Shelf Water (CT < -0.7 °C) comes as
Weddell Sea Bottom Water (WSBW) from the south (Figure 17). AABW is found from 1000m to
5500m depth (Figure 16 and 17) with low temperature (CT < 0 °C), salinity (SA < 34.68) but high
nutrient, especially silicate, concentrations (Figure S12).

## 7.2. Northeast Atlantic Bottom Water (NEABW)

Northeast Atlantic Bottom Water (NEABW), also called lower Northeast Atlantic Deep Water (lNEADW, Garcia-Ibanez et al., 2015), is mainly found below 4000m depth in the eastern basin of the North Atlantic (Figure 5). This water mass is an extension of AABW during the way to the north, since the properties of AABW change significantly on the slow transport north. A new SWT is redefined for this water mass in north of the Equator, similar as the redefinition of NADW south of the Labrador Sea.

The region east of the MAR and between the equator and 30 °N, i.e. before NEABW enters the Iberian Basin, is selected as the redefining area of NEABW. The criteria of depth deeper than 4000 m and CT above 1.8 °C are also used (Table 2). In the CT—SA diagram in Figure 3, similar T—S distribution between NEABW and AABW can be seen but with higher CT and SA of ~1.95 °C and ~35.060 g kg$^{-1}$. Most NEABW samples have a neutral density higher than 28.10 kg m$^{-3}$ and NEABW is characterized by low CT and SA, but high silicate concentration (Figure S13). This further suggests that NEABW originates from AABW, although most properties have been changed significantly from the origin in the South Atlantic.

## 7.3. Circumploar Deep Water (CDW)

Circumpolar Deep Water (CDW), which has significance to the thermohaline circulation during the wind-driven upwelling in the Southern Ocean (Morrison et al., 2015), is the lighter of the two water masses contribute to AABW formation. The production of this water mass can be tracked to the southward flow of NADW and the large-scale mixing in the Antarctic Circumpolar Current (ACC) region (van Heuven et al., 2011). At about 50°S, NADW is deflected upward by AABW before reaching the ACC (Figure 14, upper panel), this part of NADW spreads further southward into the ACC region, where it contacts with other water masses, including AAIW above and AABW below. After passing the ACC region, CDW splits into two branches at ~60 °S. The upper branch is upwelled and partly joint into the AAIW, while the rest spreads towards the coastal region, mixes with the cold fresh shelf water, sinks to the bottom and finally forms the Weddell Sea Bottom Water (WSBW), which is another contribution to the AABW (Marshall and Speer, 2012; Abernathey et al., 2016). The lower branch sinks and mixes with the WSBW below and contributes to the formation of AABW.

In this study, the SWTs of CDW is defined by considering the lower branch and the region between 55 and 65 °S is selected as the formation area (Figure 5). To define SWT of CDW (lower branch), water samples are selected from depth between 200 and 1000m in this region and additional constraints are SA higher than 34.82 g kg$^{-1}$ and CT between -0.5 and 1.0 °C (Table 2). Similar to other bottom/southern SWTs, CDW is also defined by high nutrient (silicate, phosphate and nitrate) and low oxygen concentrations (Figure S14).

## 7.4 Weddell Sea Bottom Water (WSBW)

The Weddell Sea Bottom Water (WSBW) is the densest water mass in the bottom layer. As mentioned
in the above section, part of CDW from the upper branch cools down rapidly by mixing with extremely
cold shelf water and sinks down to the bottom along the continental slop (Gordon, 2001). WSBW is
formed in the Weddell Sea basin below the depth of 3000m before it meets and mixes with CDW. The
low temperature of WSBW compared to CDW (CT = ~ -0.8 °C) is a characteristic property (Figure S15,
van Heuven et al., 2011).
Water samples in the latitudinal boundaries of 55 - 65 °S in the Weddell Sea (Figure 5) with pressures
larger than 3000 m and CT lower than -0.7 °C and silicate higher than 105 µmol kg$^{-1}$ are selected to
define the SWT of WSBW (Table 2), following Gordon (2001) and van Heuven et al. (2011).

**7.5. Atlantic Distribution of the bottom waters and southern water masses**

AABW and NEABW dominate the bottom layer ($\gamma$ > 28.10 kg m$^{-3}$). From the horizontal distribution
(Figure 15) it can be seen that AABW and NEABW cover the most bottom area of South and North
Atlantic respectively. In fact, both water masses have the same origin but are distinguished by redefining
a new SWT due to the sharp reduction of silicate after passing the equator (Figure 16). The AABW is
formed in the Weddell Sea region south of the Antarctic Circumpolar Current (ACC). After leaving the
formation area, AABW sinks to the bottom due to the high density during the way north. After passing
the ACC, AABW suffers from water exchange with NADW between 50 °S and the equator (van Heuven
et al., 2011). Similar to AABW, NEABW also mainly contacts with lower NADW and its origin (ISOW)
in the North Atlantic (Garcia-Ibanez et al., 2015).
In the Weddell Sea region, distributions of water masse mainly reflect the formation process of AABW
as displayed based on SR04 sections (Figure 17). In the zonal section, AABW can be seen as the mixture
of CDW and WSBW, where the core of CDW distributes in the upper 1000m and WSBW origins from
the surface and sinks along the continental slope into the bottom below 4000m. Both original water
masses meet each other at depth between ~2000 and 4000m, where AABW is formed, with main core
locates at ~3000m. The meridional section shows the northward outflow of AABW into the Atlantic
Ocean. AABW is located between 2000 and 4000m as a product from CDW and WSBW. After leaving
Weddell Sea region, AABW is considered as an independent water mass and spreads further northward
as the only bottom water mass until the equator.

**8. Conclusions and Discussion**

The characteristics of the main water masses in their formation areas are defined in a 6-dimensional
hydro-chemical space in the Atlantic Ocean. The values of properties for these water masses form the
fundamental basis to investigate their transport, distribution and mixing and referred to as SWTs. Table
4 and Figure 3 provides an overview SWTs of all the 16 Atlantic Ocean main water masses considered

in this study. The distribution of water masses are estimated by using OMP analysis based on the GLODAPv2 data product, and preliminarily divided into four vertical layers based on neutral densities.

The upper layer, which covers the most shallow layer (typically down to about 500 m depth) of the ocean below the mixed layer (the mixed layer is not consider in this analysis), is occupied by Central Waters. The intermediate layer is situated between the upper layer and the deep and overflow layer at roughly 1000 to 2000m depth. Of the three water masses in this layer, AAIW and SAIW are both characterized by relative low salinity and temperature, while the MW hasve high SA and CT. The SAIW and MW show a Northwest-Southeast distribution in the North Atlantic, while the AAIW dominates the intermediate layer of the region south of 30 °N.  In the deep and overflow layer between roughly 2000 and 4000m, NADW, which contains upper and lower portions, is recognized as the dominate water mass with  a relative complex origin from LSW, ISOW and DSOW.  The bottom layer is occupied by AABW with a southern origin formed by CDW and WSBW. After passing the equator, this water mass is redefined as NEABW due to the changes in properties (silicate).

Besides the 16 main Atlantic Ocean water masses, additional water masses still exist and can be found in the Atlantic that cannot be explained by the mixing of any above listed original water masses. This tends to happen close to the coast by local oceanographic events, such water masses are not listed and considered as main water mass in this study, and no additional SWTs are defined. For instance, in the coastal region of Southern Benguela Upwelling System (15 – 20 °E, 30 – 34 °S), water samples are found with low temperature and oxygen (CT = ~8 °C, oxygen = ~150 μmol kg$^{-1}$). This cannot be explained by the mixing of ESACW and WSACW, which are the only two possible water masses in this region and depth, because the CT and oxygen of both water masses are higher than these values. One possible explanation is that low-oxygen water transported by upwelling (Flynn et al., 2020).

The here presented characteristics (property values and the standard deviations) of Atlantic Ocean water masses and their distributions are intended to guide water mass analysis of hydrographic data and expect to provide a basis for further biogeochemical research.

**Acknowledgements**

This work is based on the comprehensive and detailed data from GLODAP data set throughout the past few decades. In particular, we are grateful to the efforts from all the scientists and crews on cruises, who generated funding and dedicated time on committing the collection of data. We also would like to thank the working groups of GLODAP for their support and information of the collation, quality control and publishing of data. Their contributions and selfless sharing are prerequisites for the completion of this work. We are grateful to Johannes Karstensen for his support and advices in running OMP programs and to Marcus Dengler for the information and suggestions on physical oceanography during the writing

process. We are very thankful for the ground breaking research by the late Matthias Tomczek that made
this work possible, and for the constructive comments on a previous version of this manuscript. Thanks
gores to the China Scholarship Council (CSC) for providing the funding support to Mian Liu's PhD
study in GEOMAR Helmholtz Centre for Ocean Research Kiel. Thanks to Prof. Minggang Cai and all
the colleagues in the research group of Marine Organic Chemistry (MOC) for the help and support
during Mian Liu's post-doctoral work in College of Ocean and Earth Sciences, Xiamen University.

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

**Figures**

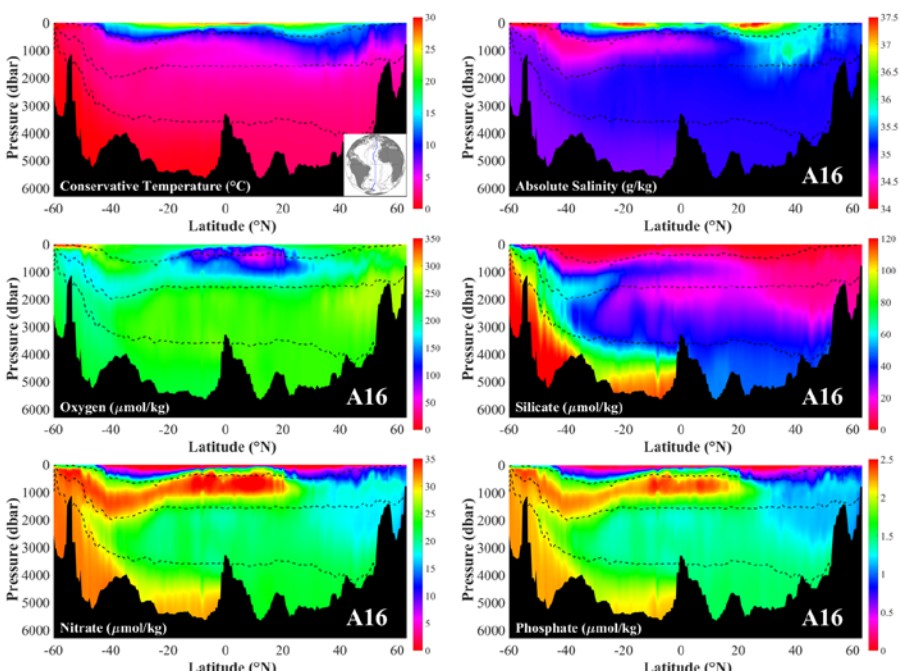


**Figure 1:** The Atlantic distribution of key properties required by the OMP analysis along the A16
section as occupied in 2013 (Expocode: 33RO20130803 in North Atlantic & 33RO20131223 in South
Atlantic). The dashed lines show the neutral densities at 27.10, 27.90 and 28.10 kg m$^{-3}$.

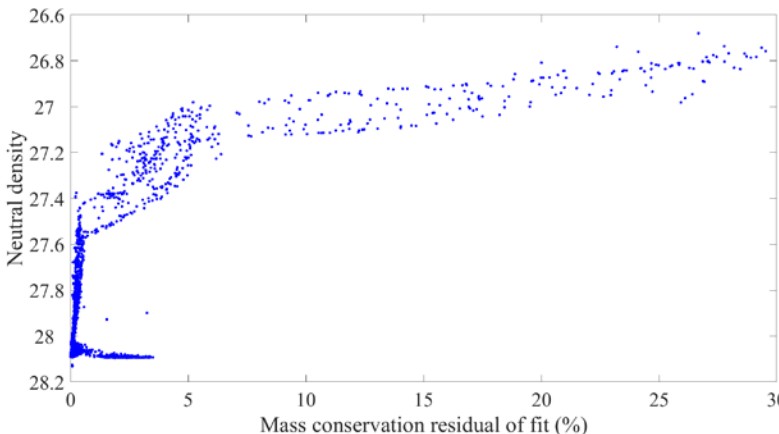


**Figure 2:** An example of a mass conservation residual in OMP analysis for the A03 section. This
figure indicates that in density layers outside of the water masses included in the analysis, we find a
high residual, i.e. the OMP analysis should only be used for a certain density interval.

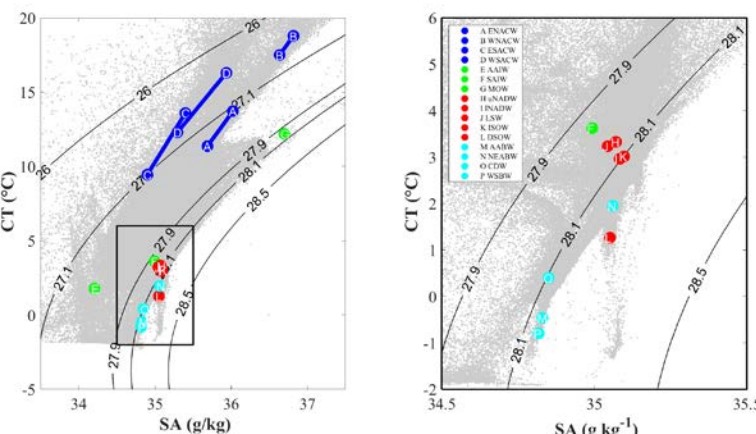


**Figure 3:** T - S diagram of all Atlantic data from the GLODAPv2 data product (gray dots) indicating the 16 main SWTs in the Atlantic Ocean discussed in this study. The colored dots with letters A--D show the upper and lower boundaries of Central Waters, and E--P show the mean values of other SWTs.

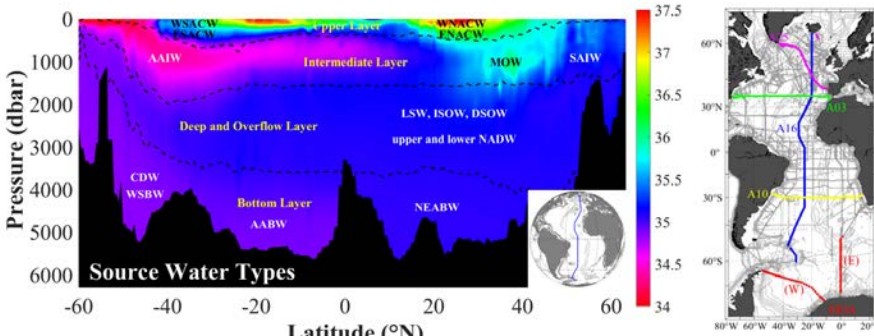

944

**Figure 4:** Left panel: Distributions of water masses in the Atlantic Ocean based on the A16 section in 2013. The background color shows the Absolute Salinity (g kg$^{-1}$). The dashed lines show the boundary of the four vertical layers divided by Neutral Density. Right panel: Five selected WOCE/GO-SHIP sections that were selected in this work to represent the vertical distribution of the main water masses.

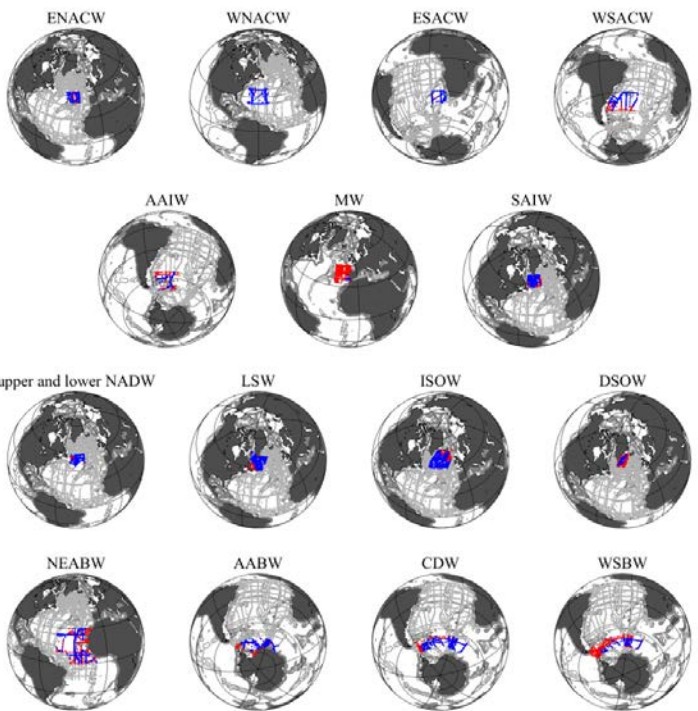


**Figure 5:** Formation/Redefining areas of the 16 main water masses in the Atlantic Ocean. The red
dots show stations in formation area and the blue dots stations ~~that~~ where the SWT was found, and the
grey dots show all the stations from GLODAPv2 dataset.

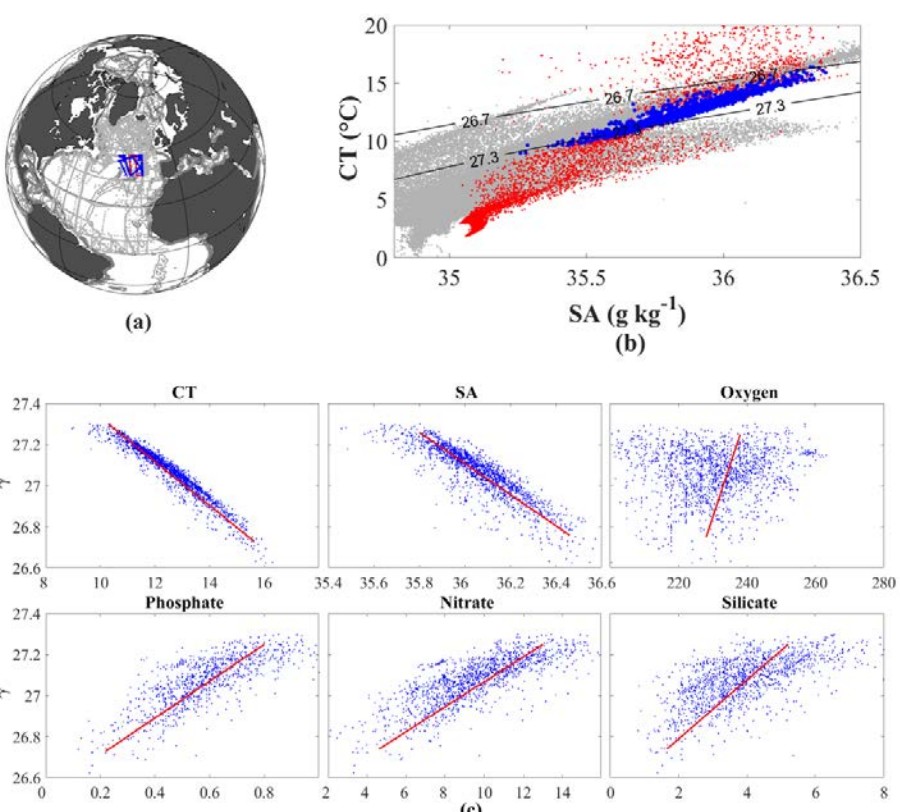


**Figure 6:** Example of a selection of water samples to define a water mass (here ENACW): Panel a) the formation area, b) the T-S diagram. The red dots show all the data in formation area, the blue dots show the selected data as ENACW and the grey dots show all the data from GLODAPv2 dataset. Panel c): Six key Properties vs Neutral Density (γ) as independent variable. Blue dots show the selected data as ENACW from Panel a) and b) and the red line shows the linear fit. The start and end points of the red line are the upper and lower boundaries of ENACW.

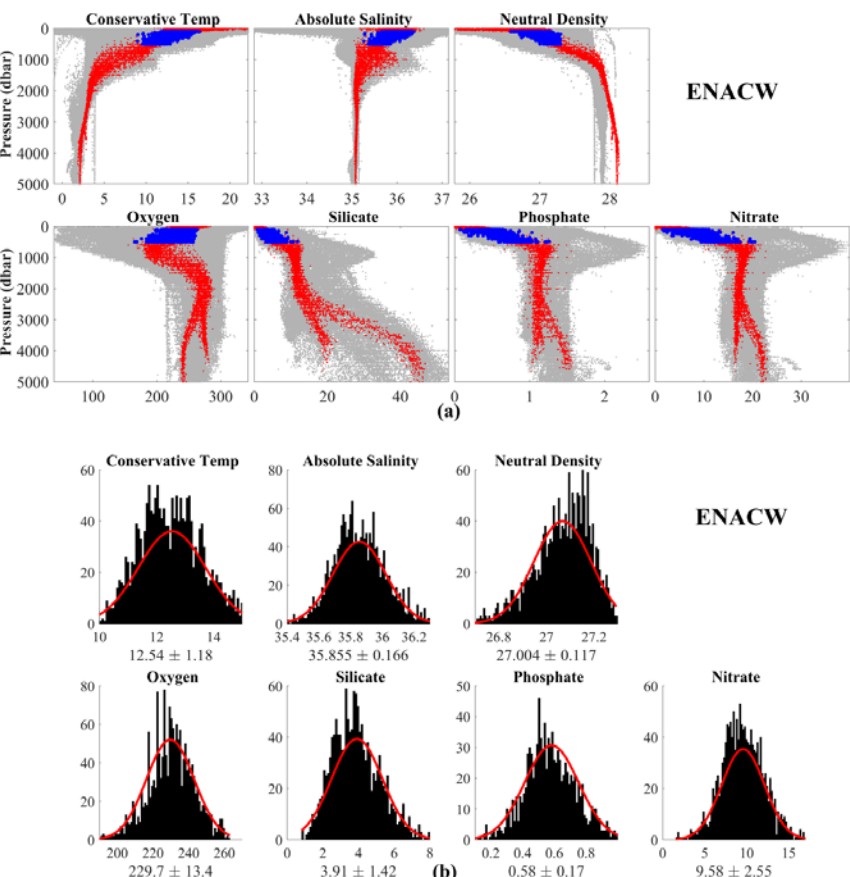

960

**Figure 7:** Example of the definition  of an SWTs (here ENACW): Panel a) The distribution of key
properties vs. pressure; Panel b) Bar plots of the data distribution of samples used to define the SWTs.
Conservative Temperature (°C), Absolute Salinity (g kg$^{-1}$), Neutral Density (kg m$^{-3}$), Oxygen and
Nutrients (µmol kg$^{-1}$). The red Gaussian fit shows mean value and standard deviation of selected data.

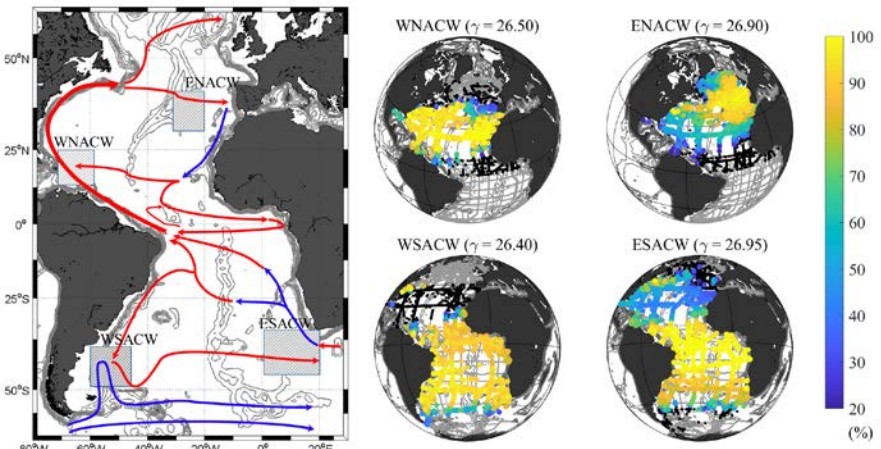

965

**Figure 8:** Currents (left) and Water Masses (right) in the Upper Layer. Left panel: The warm (red) and cold (blue) currents (arrows) and the formation areas (rectangular shadows) of water masses in the Upper Layer. Right panel: The colored dots show fractions from 20% to 100% of water masses in each station around its core neutral densities (kg m$^{-3}$). Stations with fractions less than 20% are marked by black dots while gray dots show ~~the~~ all the GLODAPv2 stations.

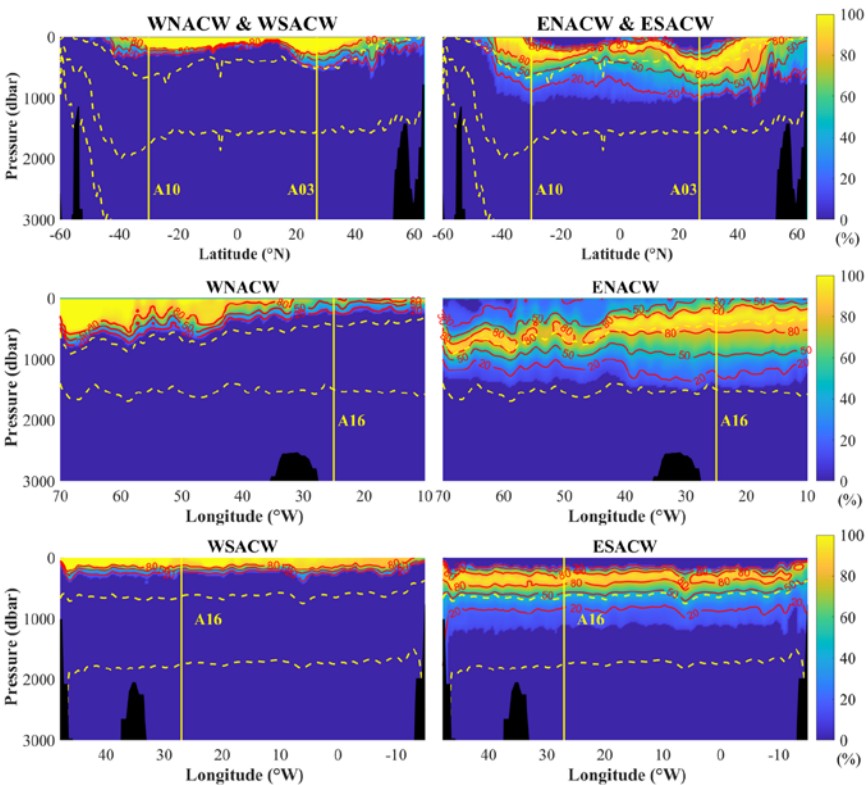

971

**Figure 9:** Distribution of Central Water Masses based on A16 (upper), A03 (middle), A10 (lower) sections for the top 3000 m depth. The contour lines show fractions of 20% 50% and 80%, yellow vertical lines show cross overs with other sections, and yellow dashed lines show ~~the boundaries of~~ vertical boundaries of ~~water columns~~ layers (neutral density at 27.10, 27.90 and 28.10 kg m$^{-3}$).

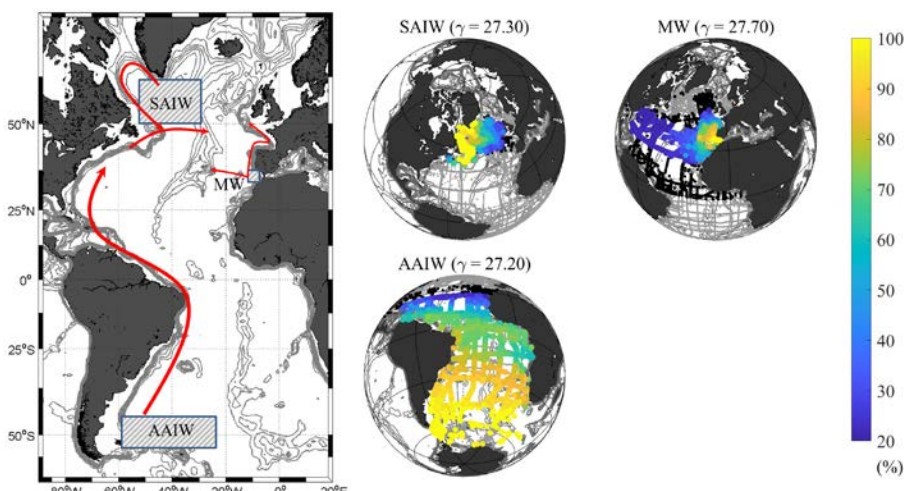

976

**Figure 10:** Currents (left) and Water Masses (right) in the Intermediate Layer. Left panel: The currents (arrows) and the formation areas (rectangular shadows) of water masses in the Intermediate Layer. Right panel: The colored dots show fractions from 20% to 100% of water masses in each station around the core neutral densities (kg m⁻³). Stations with fractions less than 20% are marked by black dots while gray dots show all the GLODAPv2 stations.

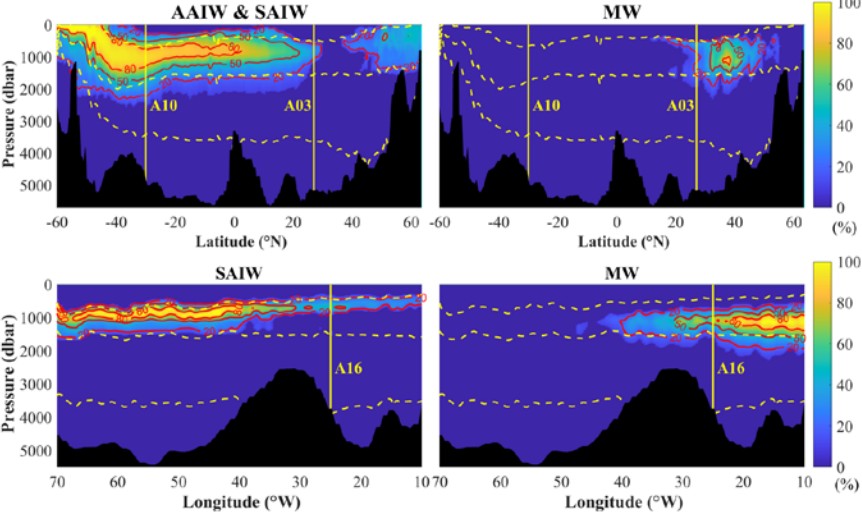

982

**Figure 11:** Distribution of Water Masses in the Intermediate Layer based on A16 (upper) and A03 (lower) sections. Contour lines show fractions of 20% 50% and 80%, yellow vertical lines show cross overs with other sections, yellow dashed lines show the ~~boundaries of vertical water columns layers~~ vertical boundaries of layers (neutral density at 27.10, 27.90 and 28.10 kg m⁻³).

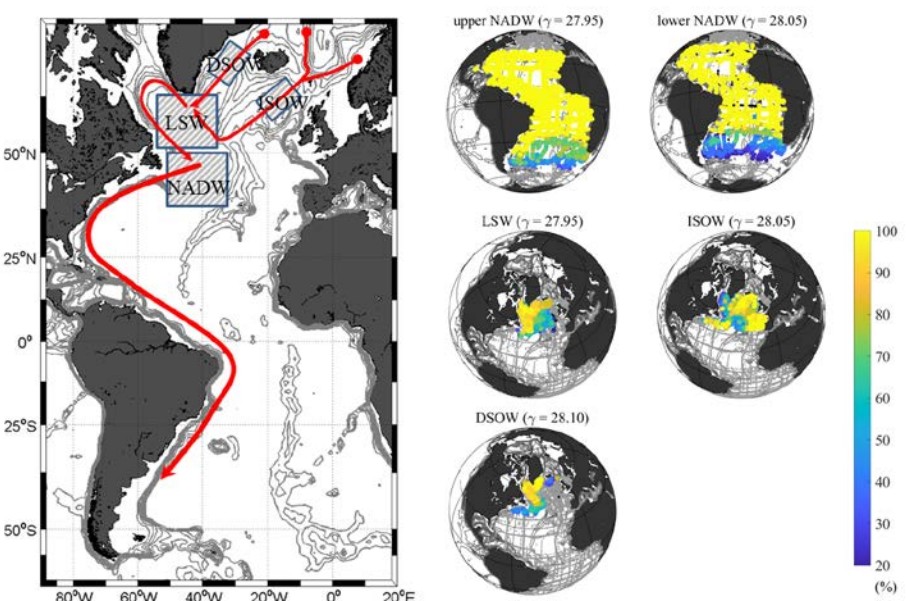

987

**Figure 12:** Currents (left) and Water Masses (right) in the Deep and Overflow Layer. Left panel: The currents (arrows) and the formation areas (rectangular shadows) of water masses in the Deep and Overflow Layer. Right panel: The colored dots show fractions (from 20% to 100%) of water masses in each station around core neutral density (kg m$^{-3}$). Stations with fractions less than 20% are marked by black dots while gray dots show all the GLODAPv2 stations.

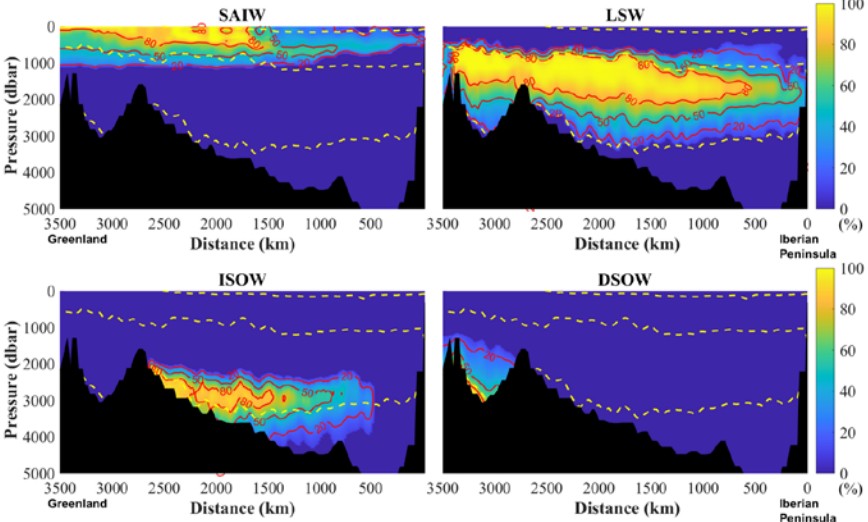


**Figure 13:** Distribution of SAIW (upper left), LSW (upper right), ISOW (lower left) and DSOW (lower right) based on the A25 section. Contour lines show fractions of 20% 50% and 80% and yellow

dashed lines show the ~~boundaries of vertical water columns layers~~ vertical boundaries of layers
(neutral density at 27.10, 27.90 and 28.10 kg m$^{-3}$).

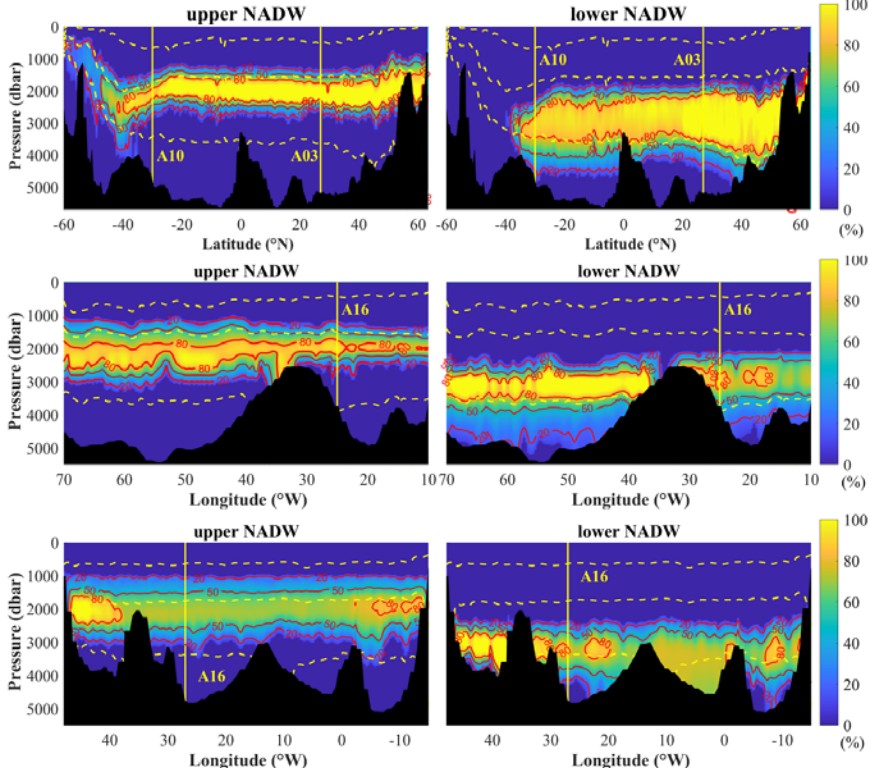


**Figure 14:** Distribution of upper and lower NADW based on the A16 (upper), A03 (middle) and A10
(lower) sections. Contour lines show fractions of 20% 50% and 80%, yellow vertical lines show cross
overs with other sections, and the ~~yellow dashed lines show the boundaries of vertical water columns~~
~~layers~~ vertical boundaries of layers (neutral density at 27.10, 27.90 and 28.10 kg m$^{-3}$).

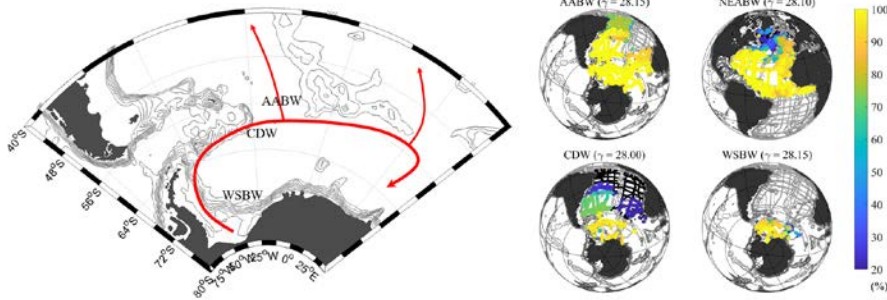


**Figure 15:** Currents (upper) and Water Masses (lower) in the Bottom Layer (AABW and NEABW)
and the Southern Area (CDW and WSBW). Left panel: The arrows show the main currents in the
Southern Area. Right panel: The colored dots show fractions (from 20% to 100%) of water masses in
each station around core neutral density (kg m$^{-3}$). Stations with fractions less than 20% are marked by
black dots while gray dots show all the GLODAPv2 stations.

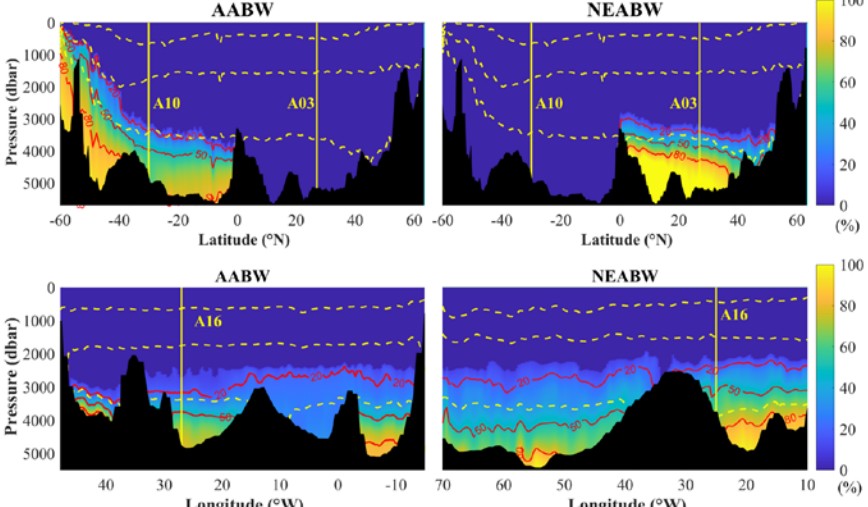


**Figure 16:** Distribution of AABW and NEABW based on A16 (upper), A10 (lower left) and A03
(lower right) sections. Contour lines show fractions of 20% 50% and 80%, yellow vertical lines show
cross overs with other sections, yellow dashed lines show the boundaries of vertical water columns
layers (neutral density at 27.10, 27.90 and 28.10 kg m$^{-3}$).

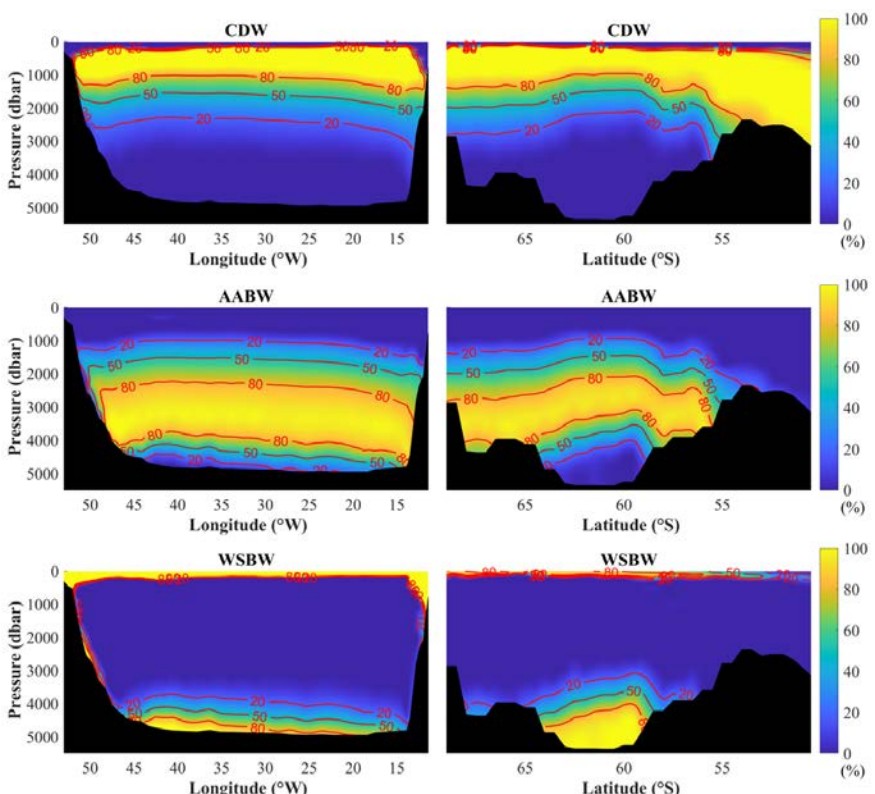


**Figure 17:** Distribution of Southern Water Masses (CDW, AABW and WSBW) based on SR04
sections. The left column shows the west (zonal) part and right column shows^ the east (meridional)
part of the section. The contour lines show fractions of 20% 50% and 80%.







**Table 1:** Schematic of the selection criteria for the OMP analysis (runs) in this study.

1026        50 °S                    Equator                        40 °N

| #17<br>AAIW AABW<br>CDW WSBW<br><br><br>(γ = 27.10 kg m⁻³) | #13<br>WSACW ESACW<br>AAIW<br><br>(γ = 26.70 kg m⁻³) | #5<br>WSACW WNACW<br>ESACW(upper) ENACW(upper) | | #1<br>WNACW ENACW<br>SAIW MOW<br><br>(γ = 26.70 kg m⁻³) |
|---|---|---|---|---|
| | | 30°N | | |
| | | #7<br>ESACW(lower)<br>ENACW(lower)<br>AAIW MOW<br>uNADW | #6<br>ESACW(lower)<br>ENACW(lower)<br>MOW SAIW<br>uNADW | |
| | #14<br>ESACW<br>AAIW<br>uNADW<br><br>(γ = 27.30 kg m⁻³) | #9<br>ENACW(lower)<br>ESACW(lower)<br>AAIW MOW<br>uNADW | #8<br>ENACW(lower)<br>ESACW(lower)<br>MOW SAIW<br>uNADW | #2<br>ENACW<br>SAIW MOW<br>LSW (uNADW)<br><br>(γ = 27.30 kg m⁻³) |
| | | 30°N | | |
| | | #10<br>AAIW MOW<br>uNADW | | |
| (γ = 27.90 kg m⁻³) | #15<br>AAIW<br>uNADW lNADW<br>CDW AABW | #11<br>AAIW MOW<br>uNADW lNADW<br>NEABW | | #3<br>SAIW<br>LSW ISOW DSOW<br>(uNADW lNADW)<br>NEABW |
| (γ = 28.10 kg m⁻³) | #16<br>lNADW<br>AABW | #12<br>lNADW (ISOW DSOW)<br>NEABW | | #4<br>ISOW DSOW<br>(lNADW)<br>NEABW |

1027        50 °S                    Equator                        40 °N

**Table 2:** Summary of the criteria used to select the water samples considered to represent the Source Water Types discerned in this study.
For convenience, they are grouped into four depth layers.

| Layer | SWT | Longitude | Latitude | Pressure (dbar) | Conservative Temperature (°C) | Absolute Salinity (g kg$^{-1}$) | Neutral Density (kg m$^{-3}$) | Oxygen (µmol kg$^{-1}$) | Silicate (µmol kg$^{-1}$) |
|---|---|---|---|---|---|---|---|---|---|
| Upper Layer | ENACW | 20°W—35°W | 39°N—48°N | 100 — 500 | --- | --- | 26.50—27.30 | --- | --- |
| | WNACW | 50°W—70°W | 24°N—37°N | 100 —500 | --- | --- | 26.20—26.70 | --- | < 2 |
| | ESACW | 0—15°E | 30°S—40°S | 200 — 700 | --- | --- | 26.00—27.50 | 200—230 | < 8 |
| | WSACW | 25°W—60°W | 30°S—45°S | 100 —1000 | --- | --- | 26.00—27.00 | < 230 | < 5 |
| Intermediate Layer | AAIW | 25°W—55°W | 45°S—60°S | 100 — 300 | < 3.5 | < 34.40 | 26.95—27.50 | > 260 | < 30 |
| | SAIW | 35°W—55°W | 50°N—60°N | 100 — 500 | > 4.5 | --- | > 27.70 | --- | --- |
| | MW | 6°W—24°W | 33°N—48°N | > 300 | --- | 36.50—37.00 | --- | --- | --- |
| Deep and Overflow Layer | uNADW | 32°W—50°W | 40°N—50°N | 1200—2000 | < 4.0 | --- | 27.85—28.05 | --- | --- |
| | lNADW | 32°W—50°W | 40°N—50°N | 2000—3000 | > 2.5 | --- | 27.90—28.10 | --- | --- |
| | LSW | 24°W—60°W | 48°N—66°N | 500 —2000 | < 4.0 | --- | 27.70—28.10 | --- | --- |
| | ISOW | 0—45°W | 50°N—66°N | 1500—3000 | 2.2—3.3 | > 34.95 | > 28.00 | --- | < 18 |
| | DSOW | 19°W—46°W | 55°N—66°N | >1500 | < 2.0 | --- | > 28.15 | --- | --- |
| Bottom Layer | AABW | --- | > 63°S | --- | --- | --- | > 28.20 | > 220 | > 120 |
| | CDW | < 60°W | 55°S—65°S | 200—1000 | -0.5—1 | > 34.82 | > 28.10 | --- | --- |
| | WSBW | --- | 55°S—65°S | 3000---6000 | < -0.7 | --- | --- | --- | --- |
| | NEABW | 10°W—45°W | 0—30°N | > 4000 | > 1.8 | --- | --- | --- | --- |


**Table 3:** *The full names of the water masses discussed in this study, and the abbreviations.*

| Full name of Water Mass | Abbreviation |
|---|---|
| East North Atlantic Central Water | ENACW |
| West North Atlantic Central Water | WNACW |
| West South Atlantic Central Water | WSACW |
| East South Atlantic Central Water | ESACW |
| Antarctic Intermediate Water | AAIW |
| Subarctic Intermediate Water | SAIW |
| Mediterranean Overflow Water | MOW |
| Upper North Atlantic Deep Water | uNADW |
| Lower North Atlantic Deep Water | lNADW |
| Labrador Sea Water | LSW |
| Iceland-Scotland Overflow Water | ISOW |
| Denmark Strait Overflow Water | DSOW |
| Antarctic Bottom Water | AABW |
| Circumpolar Deep Water | CDW |
| Weddell See Bottom Water | WSBW |
| Northeast Atlantic Bottom Water | NEABW |



**Table 4:** *Table of the mean value and the standard deviation of all variables for all the water masses*
 *(i.e. Source Water Types) in this study*

| Layer | SWTs | Conservative Temperature (°C) | Absolute Salinity | Neutral Density (kg m$^{-3}$) | Oxygen (μmol kg$^{-1}$) | Silicate (μmol kg$^{-1}$) | Phosphate (μmol kg$^{-1}$) | Nitrate (μmol kg$^{-1}$) |
|---|---|---|---|---|---|---|---|---|
| Upper Layer | ENACW (upper) | 13.72 | 36.021 | 26.887 | 243.1 | 2.49 | 0.41 | 7.03 |
| | ENACW (lower) | 11.36 | 35.689 | 27.121 | 216.3 | 5.33 | 0.75 | 12.14 |
| | WNACW (upper) | 18.79 | 36.816 | 26.344 | 213.3 | 0.72 | 0.08 | 2.00 |
| | WNACW (lower) | 17.51 | 36.634 | 26.554 | 193.9 | 1.60 | 0.24 | 4.88 |
| | ESACW (upper) | 13.60 | 35.398 | 26.500 | 217.1 | 3.68 | 0.65 | 8.26 |
| | ESACW (lower) | 9.44 | 34.900 | 26.928 | 214.2 | 6.60 | 1.19 | 16.42 |
| | WSACW (upper) | 16.30 | 35.936 | 26.295 | 222.2 | 1.60 | 0.32 | 3.15 |
| | WSACW (lower) | 12.30 | 34.294 | 26.707 | 209.8 | 3.58 | 0.80 | 10.43 |
| Intermediate Layer | AAIW | 1.78±1.02 | 34.206±0.083 | 27.409±0.111 | 300.7±16.2 | 21.09±4.66 | 1.95±0.11 | 27.33±1.92 |
| | SAIW | 3.62±0.43 | 34.994±0.057 | 27.831±0.049 | 294.6±9.7 | 8.53±0.85 | 1.04±0.07 | 15.55±1.06 |
| | MW | 12.21±0.77 | 36.682±0.081 | 27.734±0.150 | 186.2±10.7 | 7.17±1.75 | 0.74±0.11 | 12.71±1.96 |
| Deep and Overflow Layer | Upper NADW | 3.33±0.31 | 35.071±0.027 | 27.942±0.027 | 279.4±8.0 | 11.35±0.78 | 1.11±0.04 | 16.99±0.49 |
| | Lower NADW | 2.96±0.21 | 35.083±0.019 | 28.000±0.029 | 278.0±4.6 | 13.16±1.42 | 1.10±0.05 | 16.80±0.48 |
| | LSW | 3.24±0.32 | 35.044±0.031 | 27.931±0.042 | 287.4±8.5 | 9.79±0.85 | 1.08±0.06 | 16.30±0.58 |
| | ISOW | 3.02±0.26 | 35.098±0.028 | 28.001±0.044 | 277.2±3.3 | 12.21±1.18 | 1.10±0.05 | 16.58±0.48 |
| | DSOW | 1.27±0.29 | 35.052±0.016 | 28.194±0.028 | 300.3±3.6 | 8.66±0.77 | 0.95±0.05 | 13.93±0.44 |
| Bottom Layer | AABW | -0.46±0.24 | 34.830±0.009 | 28.357±0.048 | 239.0±9.3 | 124.87±2.36 | 2.27±0.03 | 32.82±0.45 |
| | CDW | 0.41±0.19 | 34.850±0.011 | 28.188±0.037 | 203.8±8.5 | 115.53±7.72 | 2.31±0.06 | 33.46±0.91 |
| | WSBW | -0.79±0.05 | 34.818±0.005 | 28.421±0.010 | 251.8±3.7 | 119.93±3.26 | 2.24±0.03 | 32.50±0.36 |
| | NEABW | 1.95±0.06 | 35.061±0.008 | 28.117±0.005 | 245.9±3.7 | 47.06±2.32 | 1.49±0.04 | 22.27±0.53 |