# Peer review of "Water Masses in the Atlantic Ocean: Characteristics and Distributions"

_Ocean Science, 2018_

## Referee Comment (RC1) · Mathias Tomczak (Referee) · 6 Feb 2019

The paper uses the most comprehensive ocean database to objectively define source water types for the Atlantic Ocean.

The paper is put sloppily together and will require a careful reading by someone not too close to the original manuscript to iron out the many grammatical errors and other inconsistencies. I do not consider it the role of the reviewer to do that, but here are a few pointers as to what has to be done:

a) The reference "Schaffer, A.J., JACOBSEN, A.W., Mikulicz's syndrome: a report of ten cases. American Journal of Diseases of Children 34, 327-346, 1927" has no place in an oceanography paper. I assume that, in an effort to be as comprehensive on the

history of water mass analysis as possible, the author meant "Jacobsen, J P. (1927) Eine graphische Methode zur Bestimmung des Vermischungskoeffizienten im Meer. Gerlands Beiträge zur Geophysik, 16, 404-412" and went astray in his Google search.

b) "The ocean is thus composed a large number of water masses" misses the word "of", and similar with many sentences.

c) "Source water type" is sometimes abbreviated as SWT, sometimes as STW.

d) "Based on the work of (Pollard and Pu, 1985)" should be "Based on the work of Pollard and Pu (1985)", and similar with many citations.

Turning to the scientific content of the paper, it may be noted that there are basically two approaches to derive objective definitions of water masses and their source water types. Assuming no knowledge of the situation under study one might try cluster analysis and discover the relevant source water types from that. If, on the other hand, sufficient information is available to predict the source water regions with certainty one might proceed with an analysis of the parameter distributions in the preselected source water regions and derive the source water types from that. Given the large amount of information available for the Atlantic Ocean the paper rightly proceeds along the second path.

Unfortunately, the paper does not fully grasp one of the basics of water mass theory, namely the distinction between water masses formed by subduction and water masses formed by winter-time deep convection. The water masses of the deep ocean are mostly formed by winter-time deep convection. Source water types of the latter variety represent the conditions in their source regions during winter and can therefore be mathematically described by a single point in parameter space, with an associated variance representing interanual changes during water mass formation. The variable values given in Table 3 for the lower three layers can thus bee seen as a listing of the SWTs for the corresponding water masses, which are defined through single points in parameter space.
The water masses of the upper ocean (known as Central Waters), on the other hand, are formed by subduction over extensive ocean regions of the subtropics and therefore characterized not by points in parameter space but by usually nearly linear parameter relationships. This is clearly seen in Figure 3, where the TS-relationship of ENACW stretches from 10 to 15°C and 35.2 to 36.2 in salinity, or in Figures 5 and 6, which show similar TS-relationships for WSACW and ESACW. (This is even occasionally acknowledged in the text, for example in section 3.1.1.) To define these Central Waters through source water types requires two SWTs for each Central Water to define a linear parameter relationship, one at the lower end of the parameter relationship and one at the upper end. The principle is demonstrated in detail in Figure 2 of Tomczak (1999).

The paper states " In this paper we use the concepts and definitions of water masses as given by Tomczak (1999)" but it does not appear to follow this through in full logic. It states when introducing the water masses of the upper layers: "The Upper Layer is occupied by four SWTs called central waters that are known to be formed by subducted [sic] into the thermocline (Sprintall and Tomczak, 1993; Tomczak and Godfrey, 2013) into the interior of the ocean (Pollard et al., 1996). Figure 2 illustrate [sic] a schematic of the main currents in this layer and the main formation regions of the central waters in the Atlantic Ocean. Water masses or SWTs in this layer can be easily recognized
by their linear T-S relationship."

There is complete confusion between water masses and SWTs here. The correct description would read: "The Upper Layer is occupied by four water masses called Central Waters that are known to be formed by subduction into the thermocline (Sprintall and Tomczak, 1993; Tomczak and Godfrey, 2013). Figure 2 illustrates a schematic of the main currents in this layer and the main formation regions of the Central Waters in the Atlantic Ocean. Water masses in this layer can be easily recognized
by their linear T-S relationship, and each water mass requires two SWTs to represent this relationship for a complete description."

The problem of the Central Waters continues right through the paper. If the criteria
used to define ENACW pick up all data points along the linear parameter relationship of Figure 2, what is the meaning of the variable values given for ENACW in Table 3? Those values do not describe a parameter relationship but a single parameter point in space not representative of Central Water. The representation of the Central Waters in Figure 22 also does not correspond to the definition ranges shown in Figures 2, 5 and 6. Central Waters should show up in Figure 22 as TS-lines, not TS-points. (As an aside, I assume that "F ISOW" in the figure caption should read "K ISOW".)

To summarize the review up to this point, the paper has to address the description of Central Waters afresh and make a clear distinction between subducted water masses and water masses formed by deep convection.

There are other points of a minor nature that nevertheless require attention. Figure 22 shows a sequence of data points near 34.8 salinity reaching down to temperatures as low as -1.5°C. Such data points cannot be produced by mixing between any of the SWTs shown, so they represent an additional SWT. At least a remark is required what these data points stand for and why they are not considered further.

The paper states "Mode Waters, on the other "hand, are considered as the precursor or the prototype of the central waters." This is not necessarily so. The quasi-linear parameter relationship of Central Water does not relate linearly to depth. Some depth ranges within the Central Waters display rapid parameter changes with depth, others show particularly uniform properties. Mode Waters are sub-regions of Central Water; they describe such layers within the Central Water range that display particularly uniform properties.

Finally, in a combined TS-diagram of data from a region SWTs are normally identified as the extreme TS-points that cannot be produced by mixing with other water masses. The definition points for AAIW in Figure 8 clearly include TS-combinations influenced by admixtures from NADW and even Central Water. A definition of AAIW based on its source region will have to concentrate on the low salinity points in the 0 to 2°C range.

Some commentary is necessary to justify the selection of definition points in Figure 8.

---

## Referee Comment (RC2) · Steven van Heuven (Referee) · 4 Apr 2019

Liu and Tanhua present an enumeration of statistical properties of selected sets of seawater samples, thereby defining 'Source Water Types', likely for further use in Optimal Multiparameter analysis (OMPa) in a separate manuscript. Although of laudable intentions (having 'true' or 'universal' SWT definitions would make many a PhD student's life easier), I do not see why this paper should be published in its current shape – or at all. My main criticism is, in increasing order of importance:

A) The ms. has not been carefully proofread, and glaring oversights remain.

B) Fundamental concepts in water mass analysis appear lost on the authors, rendering the findings of reduced usefulness for use by other investigators, who may work with a

different conceptual framework of water mass analysis / OMPa.

C) The findings are trivial, and possibly not application-appropriate. (although I did not read the companion manuscript).

The large and rather thorough central portion of the manuscript may have merit as a review of literature on Atlantic oceanography – however I do not consider myself qualified to judge whether it may hold value over existing work in this regard.

I can imagine this entire paper to constitute, in severely condensed form, the first three paragraphs of a/the application paper. Almost all figures may then be moved to supmat.

Below, I'll list for each of the above categories — non-exhaustively —- some illustration to my criticism stated above, and provide further general commentary.
* * *
*** A) "The ms. has not been carefully proofread, and glaring oversights remain". Writing is worryingly sloppy. References are made to obviously wrong papers. Abbreviations are jumbled. Example: – For GLODAP, reference is made to Lauvset et al (of the excellent mapped product). Given that the (bias-minimized) original bottle observations are used, reference should be to Olsen et al. That paper contains a fully self-contained explanation of the employed QC methods. This makes reference to Key et al., 2010 superfluous. Given the second authors exceedingly heavy involvement in these earlier publications, this slipup is surprising.

Phrasing is occasionally imprecise and terminology (e.g., "variable", "value", "definition" etc.) is used inaccurately. Please carefully re-read. As an example: the caption to Table 1 now reads: "Table 1: Table of all the water masses and the four main layers as defined in this study. The variables defined are used to select water samples that defines water masses in the formation regions." but perhaps, much less ambiguously, should read: "Table 1: Summary of the criteria used to select (from GLODPAv2) the water samples considered to represent the source water types discerned in this study.

For convenience, they are grouped into four depth layers."

*** B) "Fundamental concepts in water mass analysis appear lost on the authors, rendering the findings of reduced usefulness for use by other investigators, who may work with a different conceptual framework of water mass analysis / OMPa."

For instance, already the initial review section betrays a lack of understanding of the fundamentals of OMP. Line 67: "SWTs describe the original properties of water masses in their formation area, and can thus be considered as the original form of water masses (Tomczak, 1999)" is incomplete. For watermasses defined as originating from a single SWT, this is correct. However, for WMs defined as being on the mixing line between two distinct SWTs (i.e., central waters), the statement is incorrect. Throughout the text, the terminology "SWT" and "Water Mass" is mixed up. The authors appear to not be aware of (or to subscribe to) the very specific, and non-identical, definitions of SWT and WM as provided by Tomczak 1999 (although that paper is cited). Rather, the terminology is used loosely, often incorrectly, and certainly confusingly. Tomczak states (paraphrasing) that "a water mass may be defined as either a point in parameter space, or as a line between two such points". All water masses discussed in the manuscript as treated as point sources, while in wider literature many (notably the Central waters) are generally considered to be "line sources". Please re-read Tomczak 1999 and follow its protocol. For highly relevant examples of what a hierarchy of water masses and their constituent water types could look like for the Atlantic, again, consider [Middag et al., ESPL 2018] or the chapter 7 of the thesis of Van Heuven, 2012.

*** C) "The findings are trivial, and possibly not application-appropriate." Effectively, the paper is an enumeration of means and standard deviations of properties of seawater samples encountered in (slightly arbitrarily drawn) multidimensional boxes in the ocean. Such defining of SWT properties would likely be re-performed by any investigator of Atlantic water masses. Presenting them here thus has little added value for the community. (exceptions may be long-term tracing of changing Atlantic water mass distributions, always employing the same SWT definitions. Such a 'climate change' application

though, would require a less subjective or circular approach to the defining). The exact SWT definition used in a particular study may be very much application-dependent. E.g., where this study employs 'formations region' SWT definitions, other studies (e.g., Middag et al., 2018) employ 'edge-of-section' definitions for the watermasses that have not been sampled at their formation regions. Other 'ad hoc' definitions may be envisaged, and may be equally valid for the application at hand. That is, the definitions are subjective. Likely, for high-detail application, the results presented in this manuscript are too general. Conversely, for largest-scale application (i.e., basin-wide OMP), much more coarse approximations of the SWT's may suffice (see for example Middag et al., EPSL 2018).

*** Assorted commentary, in no specific order

– Although I'm a great fan of GLODAPv2, I do not see demonstrated that that data product is "uniquely ideal for use for SWT definition" (paraphrased from LINE 99), given its limited physical oceanographical detail. While GLODAPv2 is a very good biogeochemical data product, it features limited vertical resolution (vs. CTD), rather lax accuracy constraints for the exceedingly precise measurements of S and T. Without further corroboration of this statement, it is not evident why Gv2 should serve this purpose better than any other dataproduct that features S,T and O. Please elaborate. (Evidently, there may be additional value in having colocated values for N, P and Si, but I'd wager that will prove of little discriminating value for OMPa).

– The CDW in the Weddell Sea, I believe is more commonly locally referred to as the "Warm Deep Water". It evidently does not include freezing waters. However, such samples are visible at 34.65/-1.9 in Figure 20 (likely located on the continental shelf of the Antarctic Peninsula). Please select more carefully – these skew your averages...

– Table 3: What is the use of stating "potential density" statistics (particularly for deep and bottom waters)? These would not likely be used for OMPa, because no information is contained additional to what is already contained in S and T. Also, they are for

many SWT's pre-described through the sample selection criteria, so this is constitutes a rather circular result.

– Line 34: change "sea water type" to "source water type".

– On conceptual grounds I have some trouble with referring to CDW as an SWT. Rather, it may be considered to be an aged mixture of the other SWTs presented. Large-scale (extended-)OMP-analysis that considers CDW will prove unusably under-constrained for samples from the CDW (i.e., such samples might be found to consist of 100% CDW, or of 33% of each of the other SWT, or any possible combination between). (However, OMP users may obviously choose to not include CDW an a candidate SWT). Same goes for "North East Atlantic Bottom Water". That water mass is not "formed" (F19 even mentions its "formation region"). It is merely AABW that flowed there, aging, and with admixture from (already defined) deep water. This is an intermediate to other more extreme STWs, that would already be accounted for in OMPa. Obviously, for a local study of, say, NE Atl. water masses, the NEABW SWT may be used, but please refrain from using "formation region" in its context.

– Line 82: "our analysis is relatively course" ==> "coarse"

– I do not clearly see what the role is of the 4 density intervals discerned in this study? Are they merely to steer the reader's eye? Or are they used as additional boundaries to the vertical extent of selections of samples? Whatever the case, I believe that the two conceptually different ways of separating water masses by means of (i) OMP and (ii) density intervals are not necessarily compatible, and that these two methods should ideally not be mixed within a single paper, to avoid confusion.

– Line 120: "for some SWTs, key properties such as salinity, oxygen or silicate are also necessary". It may warrant some discussion as to why this does not constitute circular reasoning. For example, if one pre-defines the S-range for the samples, then the resultant average S is of little intrinsic value! This is pertinent for example to the definition of MOW ("36.35-36.65"), which in reality has salinities well beyond the stated

range.

– Most (panels of most) figures should be moved into supmat. Please maintain only an interesting subset of figures for the main ms.

– Figure 1 – this cruise is not drawn in on the map. Consider plotting it in F2. You can't expect readers to google the cruise track themselves.

– Figure 22: no characters in circles in legend. I believe it to be a shame that the paper does NOT present alternative property-property presentations of the definitions derived. Possibly, nothing beats a set of theta_vs_x plots (or similar) for visualizing the multidimensional separation of the various SWTs. For inspiration, please refer to, for instance, figure 7.5 in the thesis of van Heuven (2012). Also, from F22 I recon that there's some extremes of the samples in the S-T diagram that are not accounted for by any watertype. For example, in F22, panel B, the few hundred (?) samples at -1/34.9 have no closely associated SWT. These samples – while indeed part of the Atlantic Gv2 – are located in the Norwegian Sea, which is not covered in your work. Remove these from the figures to improve legibility and aid understanding. Also, many hundreds of samples are located at salinities well below that of AAIW (several standard deviation of the AAIW SWT definition). Are these surface samples? If not, how would these ever be represented accurately in an OMPa? For an example fix, please consider the SWT definitions in figure 7.5 of van Heuven (2012).

– Line 128 "[...] the standard deviation of the distribution (the amplitude of the curve defined as 2/3 of the highest bar)". This does not ring a bell as being a definition of standard deviation. Please rephrase.

---

## Author Comment (AC1) · 14 Oct 2019

As a novice in this field, I am very honored to be guided by the predecessors. Especially Matthias Tomczek, he gave me valuable guidance to this manuscript, and pointed out my shortcomings. We read the comments carefully and made the following modifications according to his suggestion. The attachment is the PDF version, the comments and our revisions are marked with different fonts and colors.
**Reply to the comments by Reviewer 1, Matthias Tomczek**

The paper uses the most comprehensive ocean database to objectively define source water types for the Atlantic Ocean.

The paper is put sloppily together and will require a careful reading by someone not too close to the original manuscript to iron out the many grammatical errors and other inconsistencies.

We have now rewrote the manuscript and did careful reading to iron out the many language

errors.

I do not consider it the role of the reviewer to do that, but here are a few pointers as to what has to be done:

a) The reference "Schaffer, A.J., JACOBSEN, A.W., Mikulicz's syndrome: a report of ten cases. American Journal of Diseases of Children 34, 327-336, 1927 has no place in an oceanography paper. I assume that, in an effort to be as comprehensive on the history of water mass analysis as possible, the author meant "Jacobsen, J P. (1927) Eine graphische Methode zar Bestimmung des Vermischungskoeffizienten im Meer. Gerlands Beiträge zur Geophysik, 16, 404-412" and went astray in his Google search.

**Thanks for pointing this error out. Reference now removed.**

b) "The ocean is thus composed a large number of water masses" misses the word "of", and similar with many sentences.

**Corrected (line 41)**

c) "Source water type" is sometimes abbreviated as SWT, sometimes as STW.

We went through the manuscript and corrected to it now is always "SWT".

d) "Based on the work of (Pollard and Pu, 1985)" should be "Based on the work of Pollard and Pu (1985)", and similar with many citations.

**We went through the manuscript and corrected the citations. (line 276)**

Turning to the scientific content of the paper, it may be noted that there are basically two approaches to derive objective definitions of water masses and their source water types. Assuming no knowledge of the situation under study one might try cluster analysis and discover the relevant source water types from that. If, on the other hand, sufficient information is available to predict the source water regions with certainty one might proceed with an analysis of the parameter distributions in the preselected source water regions and derive the source water types from that. Given the large amount of information available for the Atlantic Ocean the paper rightly proceeds along the second path.

We agree with this assessment, and the intention of our work was not to redefine the water masses in the Atlantic Ocean, but rather to use the information already available and base the work on that. In several instances, water masses has in the past been differently defined and/or named so we tried to put this information in the paper as well.

Unfortunately, the paper does not fully grasp one of the basics of water mass theory, namely the distinction between water masses formed by subduction and water masses formed by winter-time deep convection. The water masses of the deep convection.
Fig. 1.

---

## Author Comment (AC2) · 14 Nov 2019

Thank you very much for your guidance and advice. As a novice in this field, I feel very honored to get the guidance from the predecessor. According to your suggestions, we re-read and revised the manuscript and made some replies. The detail content of each reply has been uploaded to the supplementary attachment. Thank you again for your comments.

Please also note the supplement to this comment:
https://www.ocean-sci-discuss.net/os-2018-139/os-2018-139-AC2-supplement.pdf

[Figure]

**Supplement:**

**Reply to the comments by Reviewer 1, Matthias Tomczek**

The paper uses the most comprehensive ocean database to objectively define source water types for the Atlantic Ocean.

The paper is put sloppily together and will require a careful reading by someone not too close to the original manuscript to iron out the many grammatical errors and other inconsistencies.

We have now rewritten the manuscript and did careful reading to iron out the many language errors.

I do not consider it the role of the reviewer to do that, but here are a few pointers as to what has to be done:

a) The reference "Schaffer, A.J., JACOBSEN, A.W., Mikulicz's syndrome: a report of ten cases. American Journal of Diseases of Children 34, 327-346, 1927" has no place in an oceanography paper. I assume that, in an effort to be as comprehensive on the history of water mass analysis as possible, the author meant "Jacobsen, J P. (1927) Eine graphische Methode zur Bestimmung des Vermischungskoeffizienten im Meer. Gerlands Beiträge zur Geophysik, 16, 404-412" and went astray in his Google search.

Thanks for pointing this error out. Reference now removed.

b) "The ocean is thus composed a large number of water masses" misses the word "of", and similar with many sentences.

**Corrected (line 41)**

c) "Source water type" is sometimes abbreviated as SWT, sometimes as STW.

We went through the manuscript and corrected to it now is always "SWT".

d) "Based on the work of (Pollard and Pu, 1985)" should be "Based on the work of Pollard and Pu (1985)", and similar with many citations.

**We went through the manuscript and corrected the citations. (line 276)**

Turning to the scientific content of the paper, it may be noted that there are basically two approaches to derive objective definitions of water masses and their source water types. Assuming no knowledge of the situation under study one might try cluster analysis and discover the relevant source water types from that. If, on the other hand, sufficient information is available to predict the source water regions with certainty one might proceed with an analysis of the parameter distributions in the preselected source water regions and derive the source water types from that. Given the large amount of information available for the Atlantic Ocean the paper rightly proceeds along the second path.

We agree with this assessment, and the intention of our work was not to redefine the water masses in the Atlantic Ocean, but rather to use the information already available and base the work on that. In several instances, water masses has in the past been differently defined and/or named so we tried to put this information in the paper as well.

Unfortunately, the paper does not fully grasp one of the basics of water mass theory, namely the distinction between water masses formed by subduction and water masses formed by winter-time deep convection. The water masses of the deep ocean are mostly formed by winter-time deep convection.

Source water types of the latter variety represent the conditions in their source regions during winter and can therefore be mathematically described by a single point in parameter space, with an associated variance representing interanual changes during water mass formation. The variable values given in Table 3 for the lower three layers can thus bee seen as a listing of the SWTs for the corresponding water masses, which are defined through single points in parameter space.

The water masses of the upper ocean (known as Central Waters), on the other hand, are formed by subduction over extensive ocean regions of the subtropics and therefore characterized not by points in parameter space but by usually nearly linear parameter relationships. This is clearly seen in Figure 3, where the TS-relationship of ENACW stretches from 10 to 15\_C and 35.2 to 36.2 in salinity, or in Figures 5 and 6, which show similar TS-relationships for WSACW and ESACW. (This is even occasionally acknowledged in the text, for example in section 3.1.1.) To define these Central Waters through source water types requires two SWTs for each Central Water to define a linear parameter relationship, one at the lower end of the parameter relationship and one at the upper end. The principle is demonstrated in detail in Figure 2 of Tomczak (1999). The paper states "In this paper we use the concepts and definitions of water masses as given by Tomczak (1999)" but it does not appear to follow this through in full logic. It states when introducing the water masses of the upper layers: "The Upper Layer is occupied by four SWTs called central waters that are known to be formed by subducted [sic] into the thermocline (Sprintall and Tomczak, 1993; Tomczak and Godfrey, 2013) into the interior of the ocean (Pollard et al., 1996). Figure 2 illustrate [sic] a schematic of the main currents in this layer and the main formation regions of the central waters in the Atlantic Ocean. Water masses or SWTs in this layer can be easily recognized by their linear T-S relationship."

There is complete confusion between water masses and SWTs here. The correct description would read: "The Upper Layer is occupied by four water masses called Central Waters that are known to be formed by subduction into the thermocline (Sprintall and Tomczak, 1993; Tomczak and Godfrey, 2013). Figure 2 illustrates a schematic of the main currents in this layer and the main formation regions of the Central Waters in the Atlantic Ocean. Water masses in this layer can be easily recognized by their linear T-S relationship, and each water mass requires two SWTs to represent this relationship for a complete description."

**Thank you for helping us out here with a clear path forward.**

In the revised version, we made amendments to this aspect. **Central Waters** are redefined by their **upper** and **lower boundaries** by following the suggestions:

In the Section 3.1, distinctions between Central Waters and other deeper SWTs are pointed out. The most significant feature of the Central Water is that the characteristics cover a relative larger range and cannot be consider as a point value. As a result, upper and lower boundaries are set for each property of Central Waters and the values between them are all considered as SWTs of Central Waters in the further OMP analysis.

Take the potential temperature ( $\theta$ ) of ENACW as the example. After confirming the formation area (15°W—25°W, 39°N—48°N and 100 — 500 db), and the criteria ( $\sigma_{\theta}$  between 26.50 and 27.30 kg/m3), a relationship between potential temperature and density (potential density as the reference/independent parameter, detail in section 3.0) is plotted (blue dots) in Fig. 3 with linear fit (red line). The upper and lower boundaries in potential temperature of ENACW are finally determined by the choice of our own fit ranges (https://omp.geomar.de/), 14.60 °C as the upper boundary and 9.80 °C as the lower boundary. All the other properties are done with the same method and the boundaries are listed in Table 3.

In the figure plots for other deeper water masses (for example Fig. 9 for AAIW) statistics are done for all the data in the selected regions with selected criteria. For the Central Waters, statistics are done, considering all **the data between upper and lower boundaries**, and then, all the data between the boundaries are considered as one Central Water Mass and plotted in

the following figures (for example Fig. 4 for ENACW). In the figure plots, Central Water, for example ENACW, is still plotted as **one water mass**. The reason we plot figures in this way is hoping to show the reader that one Central Water, although the properties cover certain ranges and have upper and lower boundaries, is still **one water mass** instead of two.

The problem of the Central Waters continues right through the paper. If the criteria used to define ENACW pick up all data points along the linear parameter relationship of Figure 2, what is the meaning of the variable values given for ENACW in Table 3?

Those values do not describe a parameter relationship but a single parameter point in space not representative of Central Water. The representation of the Central Waters in Figure 22 also does not correspond to the definition ranges shown in Figures 2, 5 and 6. Central Waters should show up in Figure 22 as TS-lines, not TS-points. (As an aside, I assume that "F ISOW" in the figure caption should read "K ISOW".)

However, in the further OMP analysis, one Central Water **occupies two positions** of SWTs (upper and lower boundary respectively). In other words, upper and lower ENACW for example are considered two independent SWTs in the calculation of OMP analysis, but the final output in the figure plots (for example, in Fig. 4 for ENACW) is still **one water mass**, which means that all the data within the range are considered to be this central water mass.

To summarize the review up to this point, the paper has to address the description of Central Waters afresh and make a clear distinction between subducted water masses and water masses formed by deep convection.

We hope that the above modifications we have made can avoid the previous mistakes and also avoid the misunderstandings or confusions to the readers, and also conform to our original intention of quoting the literature for example, Tomczak (1999).

There are other points of a minor nature that nevertheless require attention. Figure 22 shows a sequence of data points near 34.8 salinity reaching down to temperatures as low as -1.5\_C. Such data points cannot be produced by mixing between any of the SWTs shown, so they represent an additional SWT. At least a remark is required what these data points stand for and why they are not considered further.

Thanks for pointing out this detail that we overlooked in previous version. This water mass indeed cannot be explained by the mixing of any above listed original water masses. So a new SWT should be defined. By analyzing of their distributions (blue dots in the left figure) on the map and their depth (between 2000 and 4000m), and also combining with temperature and salinity data (right plot). The properties are closer to Eurasian Basin Deep Water (e.g. Bönisch and Schlosser, 1995; Schauer et al., 2014). As a result, we preliminarily determined this water mass as from the deep Arctic (Eurasian Basin). Since this water mass originate from the Arctic Ocean, and the amount of data is too small (only 600 samples), and further there is not much interaction with other listed water masses, we did not list it as an independent water mass in this paper and also no SWT is defined so far. Thanks again for pointing out this point and also provides guidance for our next work. We explain this point in the new manuscript.

The paper states "Mode Waters, on the other â'A'lhand, are considered as the precursor or the prototype of the central waters." This is not necessarily so. The quasi-linear parameter relationship of Central Water does not relate linearly to depth. Some depth ranges within the Central Waters display rapid parameter changes with depth, others show particularly uniform properties. Mode Waters are sub-regions of Central Water; they describe such layers within the Central Water range that display particularly uniform properties.

Finally, in a combined TS-diagram of data from a region SWTs are normally identified as the extreme TS-points that cannot be produced by mixing with other water masses. The definition points for AAIW in Figure 8 clearly include TS-combinations influenced by admixtures from NADW and even Central Water. A definition of AAIW based on its source region will have to concentrate on the low salinity points in the 0 to 2\_C range. Some commentary is necessary to justify the selection of definition points in Figure 8.

In the previous version, Mode Water is considered as the precursor or the prototype of the Central Water. Such a statement is indeed not rigorous. Thanks for pointing out this key point and in the new version, such description is used. Mode Waters are described as **sub-regions** of Central Waters and also report the most significant characteristic of Central Waters: the linear T-S relationship.

---

## Author Comment (AC3) · 14 Nov 2019

Thank you very much for your time and valuable comments as our reviewer, especially your rigorous attitude to the writting. We read and revised the manuscript again according to your suggestion. The details of the replies has been uploaded to the supplementary attachment. Thanks again for your comments.

Please also note the supplement to this comment:
https://www.ocean-sci-discuss.net/os-2018-139/os-2018-139-AC3-supplement.pdf

[Figure]

**Supplement:**

**Review by van Heuven on the manuscript "Characteristics of Water Masses in the Atlantic Ocean based on GLODAPv2 data" by Mian Liu and Toste Tanhua**

Liu and Tanhua present an enumeration of statistical properties of selected sets of seawater samples, thereby defining 'Source Water Types', likely for further use in Optimal Multiparameter analysis (OMPa) in a separate manuscript. Although of laudable intentions (having 'true' or 'universal' SWT definitions would make many a PhD student's life easier), I do not see why this paper should be published in its current shape – or at all.

My main criticism is, in increasing order of importance:

A) The ms. has not been carefully proofread, and glaring oversights remain.

B) Fundamental concepts in water mass analysis appear lost on the authors, rendering the findings of reduced usefulness for use by other investigators, who may work with a different conceptual framework of water mass analysis / OMPa.

C) The findings are trivial, and possibly not application-appropriate. (although I did not read the companion manuscript).

The large and rather thorough central portion of the manuscript may have merit as a review of literature on Atlantic oceanography – however I do not consider myself qualified to judge whether it may hold value over existing work in this regard.

I can imagine this entire paper to constitute, in severely condensed form, the first three paragraphs of a/the application paper. Almost all figures may then be moved to supmat.

Below, I'll list for each of the above categories — non-exhaustively —- some illustration to my criticism stated above, and provide further general commentary.

\*\*\*

\*\*\* A) "The ms. has not been carefully proofread, and glaring oversights remain".

Writing is worryingly sloppy. References are made to obviously wrong papers.

We examined the references carefully and corrected the misquoted references.

Abbreviations are jumbled.

We went through the manuscript carefully and corrected all the abbreviations to the right way of expression.

Example: – For GLODAP, reference is made to Lauvset et al (of the excellent mapped product). Given that the (bias-minimized) original bottle observations are used, reference should be to Olsen et al. That paper contains a fully self-contained explanation of the employed QC methods. This makes reference to Key et al., 2010 superfluous. Given the second authors exceedingly heavy involvement in these earlier publications, this slipup is surprising.

We read the methods section of Olsen et al., (2016 and 2019) carefully, and make changes in the description of QC part in the new manuscript (section 2.1).

Phrasing is occasionally imprecise and terminology (e.g., "variable", "value", "definition" etc.) is used inaccurately.

We unify the relevant expressions in the whole manuscript. For example: the word "**characteristics**" is used to describe the overall water mass (for example: characteristics of water masses); "**property**" is used to describe one special features such as oxygen (for example: six key properties); the word "**variable**" is not used in the relevant description in order to avoid confusion; "**value**" is used to show specific mathematical value; "**definition**" is used to show concrete concept.

Please carefully re-read. As an example: the caption to Table 1 now reads: "Table 1: Table of all the water masses and the four main layers as defined in this study. The variables defined are used to select water samples that defines water masses in the formation regions." but perhaps, much less ambiguously, should read: "Table 1: Summary of the criteria used to select (from GLODPAv2) the water samples considered to represent the source water types discerned in this study. For convenience, they are grouped into four depth layers."

Thank you for helping us out here with a clear linguistic organization. In the new manuscript, we now use clearer expressions to make the reader understand more easily.

*** B) "Fundamental concepts in water mass analysis appear lost on the authors, rendering the findings of reduced usefulness for use by other investigators, who may work with a different conceptual framework of water mass analysis / OMPa."

For instance, already the initial review section betrays a lack of understanding of the fundamentals of OMP. Line 67: "SWTs describe the original properties of water masses in their formation area, and can thus be considered as the original form of water masses (Tomczak, 1999)" is incomplete. For water masses defined as originating from a single SWT, this is correct. However, for WMs defined as being on the mixing line between two distinct SWTs (i.e., central waters), the statement is incorrect.

Thanks for pointing out the mistakes in our previous work. We read the reference (Tomczak, 1999) again, and indeed, Central Waters, which are formed by subduction through the thermocline and into the interior of the ocean, need to be treated differently from other deep water masses due to their different formation ways. In Section 3.0, this difference is pointed out.

Throughout the text, the terminology "SWT" and "Water Mass" is mixed up. The authors appear to not be aware of (or to subscribe to) the very specific, and non-identical, definitions of SWT and WM as provided by Tomczak 1999 (although that paper is cited).

A new section (Section 2.2) is added to distinguish the differences between "Water Mass" and "STW". In general, a "Water Mass" is an objective entity with temporal and spatial distribution; while a "STW" is a set of mathematical values that describes the original properties of a "Water Mass".

Rather, the terminology is used loosely, often incorrectly, and certainly confusingly. Tomczak states (paraphrasing) that "a water mass may be defined as either a point in parameter space, or as a line between two such points". All water masses discussed in the manuscript as treated as point sources, while in wider literature many (notably the Central waters) are generally considered to be "line sources". Please re-read Tomczak 1999 and follow its protocol. For highly relevant examples of what a hierarchy of water masses and their constituent water types could look like for the Atlantic, again, consider [Middag et al., ESPL 2018] or the chapter 7 of the thesis of Van Heuven, 2012.

Thanks for the guidance in distinguishing Central Waters and other deep masses.

In the revised version, we made amendments to this aspect. Central Waters are redefined as a line between two points, which are the upper and lower boundaries. In the Section 3.1, such distinction between Central Waters and other deeper SWTs is pointed out, and all the 6 key properties of Central Water, which cover relative larger ranges and cannot be consider as point values, are redefined by upper and lower boundaries.

In the figures (Fig. 4-7), all the data between upper and lower boundaries are plotted and the Central Water is still presented to the readers as one whole water mass. In the form of figure plots, Central Water has no difference to other deep water masses. The main difference is mainly reflected in the

OMP analysis, the central water masses occupy two "positions", which are the upper and lower boundaries and the values between them all considered as SWTs of this Central Water.

For example, in Section 3.1.1, the potential temperature (θ) of ENACW has clearly two boundaries. The upper and lower boundaries in potential temperature of ENACW are finally determined by the choice of our own fit ranges (https://omp.geomar.de/), 14.60 °C as the upper boundary and 9.80 °C as the lower boundary. All the other properties are done with the same method and the boundaries are listed in Table 3.

*** C) "The findings are trivial, and possibly not application-appropriate."

Effectively, the paper is an enumeration of means and standard deviations of properties of seawater samples encountered in (slightly arbitrarily drawn) multidimensional boxes in the ocean.

We agree with this assessment, and the intention of our work was not to redefine the water masses in the Atlantic Ocean, but rather to use the information already available and base the work on that. In several instances, water masses has in the past been differently defined and/or named so we tried to put this information in the paper as well.

Such defining of SWT properties would likely be re-performed by any investigator of Atlantic water masses. Presenting them here thus has little added value for the community. (exceptions may be long-term tracing of changing Atlantic water mass distributions, always employing the same SWT definitions. Such a 'climate change' application though, would require a less subjective or circular approach to the defining).

The exact SWT definition used in a particular study may be very much application-dependent. E.g., where this study employs 'formations region' SWT definitions, other studies (e.g., Middag et al., 2018) employ 'edge-of-section' definitions for the water masses that have not been sampled at their formation regions. Other 'ad hoc' definitions may be envisaged, and may be equally valid for the application at hand. That is, the definitions are subjective. Likely, for high-detail application, the results presented in this manuscript are too general. Conversely, for largest-scale application (i.e., basin-wide OMP), much more coarse approximations of the SWT's may suffice (see for example Middag et al., EPSL 2018).

We disagree with this statement, at least partly. It is true that several investigators will go through the trouble to define their source water properties themselves, and possibly use "edge-of-section" data. We think that most do that since finding a stringent water mass definition is cumbersome. We provide in this work a comprehensive characterization of water mass properties that can be used by investigators. Obviously, the investigators focusing on a particular water mass might want to be more precise, and possibly look at temporal evolution etc. This paper is not intended for those, but rather for the chemical/biological oceanographers that would like to understand (roughly) the formation and mixing history of the water they sampled.

We explicitly aimed for being general and course in the WM characterization, realizing that "sub-water-masses" can be defined, and that spatiotemporal variability do exists.

*** Assorted commentary, in no specific order
– Although I'm a great fan of GLODAPv2, I do not see demonstrated that that data product is "uniquely ideal for use for SWT definition" (paraphrased from LINE 99), given its limited physical oceanographical detail. While GLODAPv2 is a very good biogeochemical data product, it features limited vertical resolution (vs. CTD), rather lax accuracy constraints for

the exceedingly precise measurements of S and T. Without further corroboration of this statement, it is not evident why Gv2 should serve this purpose better than any other dataproduct that features S, T and O. Please elaborate. (Evidently, there may be additional value in having colocated values for N, P and Si, but I'd wager that will prove of little discriminating value for OMPa).

Point taken. I guess another products with biogeochemical data would do the work. The main advantages of Gv2 is the internal consistency of the data, so that we can use a large set of cruises and have some confidence that the data are consistent, which might be useful for water mass analysis. We do agree that high-resolution CTD profiles would be useful, for instance one can use potential vorticity as a parameter. Since our intended main audience are chemists and biologist with sparse vertical resolution of their data (mostly), this work corresponds to that need and the expected data availability.

– The CDW in the Weddell Sea, I believe is more commonly locally referred to as the "Warm Deep Water". It evidently does not include freezing waters. However, such samples are visible at 34.65/-1.9 in Figure 20 (likely located on the continental shelf of the Antarctic Peninsula). Please select more carefully – these skew your averages...

The selection criteria have been adjusted and data with low temperature (below 0 °C) are removed.

– Table 3: What is the use of stating "potential density" statistics (particularly for deep and bottom waters)? These would not likely be used for OMPa, because no information is contained additional to what is already contained in S and T. Also, they are formany SWT's pre-described through the sample selection criteria, so this is constitutes a rather circular result.

We agree, potential density is calculated from T and S, and is not an independent property of a water mass. The column of "potential density" is removed and Table 3 shows only 6 key properties of the water masses.

– Line 34: change "sea water type" to "source water type".

Changed.

– On conceptual grounds I have some trouble with referring to CDW as an SWT. Rather, it may be considered to be an aged mixture of the other SWTs presented. Large-scale (extended-)OMP-analysis that considers CDW will prove unusably under-constrained for samples from the CDW (i.e., such samples might be found to consist of 100% CDW, or of 33% of each of the other SWT, or any possible combination between). (However, OMP users may obviously choose to not include CDW an a candidate SWT). Same goes for "North East Atlantic Bottom Water". That water mass is not "formed" (F19 even mentions its "formation region"). It is merely AABW that flowed there, aging, and with admixture from (already defined) deep water. This is an intermediate to other more extreme STWs, that would already be accounted for in OMPa. Obviously, for a local study of, say, NE Atl. water masses, the NEABW SWT may be used, but please refrain from using "formation region" in its context.

Indeed, some water masses are not original water masses, but products of spreading and mixing, for example, NEABW. Because of their special properties and may be useful for region-specific studies, we hope to still keep such water masses and consider their SWTs. In terms of expression, we distinguish such water masses from ''original'' water masses, such as AAIW. We only show the key properties of such water masses in specific region, while the definition "formation area" is only for the ''original'' water masses. This is obviously somewhat subjective, but is introduced to facilitate the process of water mass analysis.

– Line 82: "our analysis is relatively course" ==> "coarse"

Changed into "coarse", new version in line 87

– I do not clearly see what the role is of the 4 density intervals discerned in this study?
Are they merely to steer the reader's eye? Or are they used as additional boundaries to the vertical extent of selections of samples? Whatever the case, I believe that the two conceptually different ways of separating water masses by means of (i) OMP and (ii) density intervals are not necessarily compatible, and that these two methods should ideally not be mixed within a single paper, to avoid confusion.

The OMP method is the only criterion for distinguishing water masses in this study. However, there are a large number of water masses and OMP analysis can only calculate no more than 6 water masses within one OMP run (with the number of variables we have to our disposal). Therefore, the ocean is divided into 4 layers according to the density (in total 13 OMP runs, divided by density and latitude as shown in the following table) that to ensure no more than 6 water masses are present in each OMP run. Therefore, density can also be considered as an additional boundary, which will be further clarified in this article so as not to confuse the readers.

| 50°S | Equator | | 40°N |
|---|---|---|---|
| **#13** AAIW AABW CDW WSBW $(\sigma_\theta = 27 \text{ kg/m}^3)$ | **#6** WSACW ESACW AAIW | **#5** WSACW WNACW ESACW ENACW AAIW | **#1** WNACW ENACW SAIW MOW |
| | **#8** ESACW AAIW uNADW | **#7** ENACW ESACW AAIW MOW uNADW | **#2** ENACW SAIW MOW LSW |
| $(\sigma_\theta = 27.7 \text{ kg/m}^3)$ | **#10** AAIW uNADW lNADW CDW AABW | **#9** AAIW MOW uNADW lNADW NEABW | **#3** SAIW LSW ISOW DSOW NEABW |
| $(\sigma_\theta = 27.88 \text{ kg/m}^3)$ | **#12** lNADW AABW | **#11** lNADW NEABW | **#4** ISOW DSOW NEABW |

– Line 120: "for some SWTs, key properties such as salinity, oxygen or silicate are also necessary". It may warrant some discussion as to why this does not constitute circular reasoning. For example, if one pre-defines the S-range for the samples, then the resultant average S is of little intrinsic value! This is pertinent for example to the definition of MOW ("36.35-36.65"), which in reality has salinities well beyond the statedrange.

The criteria in Table 1 are listed according to historical literatures. Based on these criteria, we selected eligible data from the GLODAP dataset, and then make statistics on these eligible data to obtain a result (gaussian distribution), and these results (or values) are considered as the basis for our definition of SWT. Therefore, besides the area distribution (latitude, longitude and depth), it is also necessary to list some criteria in Table1, because there may be other water masses in the designated area, and we need to use these criteria

to eliminate the interference of external water masses. But, that being said, it is a somewhat circular argument that we define the range of a variable (e.g. S) for the definition of the water mass, and then use that to estimate the average value of that variable. However, it serves to use well-known characteristics of water mass properties to define them also in this paper.

– Most (panels of most) figures should be moved into supmat. Please maintain only an interesting subset of figures for the main ms.

This is a good point. We do think that the figures are an important part of the paper. Now, particularly since we have merged the two manuscripts into one, we do have an exceedingly large number of Figures. We will carefully make a selection to which figures will be moved to supmat. Thanks for the suggestion.

– Figure 1 – this cruise is not drawn in on the map. Consider plotting it in F2. You can't expect readers to google the cruise track themselves.

A small figure (map) is added to show the cruise.

– Figure 22: no characters in circles in legend. I believe it to be a shame that the paper does NOT present alternative property-property presentations of the definitions derived. Possibly, nothing beats a set of theta_vs_x plots (or similar) for visualizing the multidimensional separation of the various SWTs. For inspiration, please refer to, for instance, figure 7.5 in the thesis of van Heuven (2012). Also, from F22 I recon that there's some extremes of the samples in the S-T diagram that are not accounted for by any watertype. For example, in F22, panel B, the few hundred (?) samples at -1/34.9 have no closely associated SWT. These samples – while indeed part of the Atlantic Gv2 – are located in the Norwegian Sea, which is not covered in your work. Remove these from the figures to improve legibility and aid understanding. Also, many hundreds of samples are located at salinities well below that of AAIW (several standard deviation of the AAIW SWT definition). Are these surface samples? If not, how would these ever be represented accurately in an OMPa? For an example fix, please consider the SWT definitions in figure 7.5 of van Heuven (2012).

We have made corresponding modifications and adjustments to the figure as suggested.

– Line 128 "[...] the standard deviation of the distribution (the amplitude of the curve defined as 2/3 of the highest bar)". This does not ring a bell as being a definition of standard deviation. Please rephrase.

Description is reorganized, in new manuscript line 228.

---

## Referee Report (RR1)

Peer review report on "Characteristics of Water Masses in the Atlantic Ocean based on GLODAPv2 dataset" by Mian Liu and Toste Tanhua

*1) Overview and general recommendation:*

Understanding the formation, transformation, and circulation of water masses has been a hot topic in oceanography since its start. Many ways of tangling this oceanography field have been developed, from the mere description of hydrographic properties to statistical and numerical models. This manuscript presents the water mass structure of the Atlantic Ocean resulting from an Optimum Multiparameter (OMP) analysis.

Although the intention of the manuscript is honorable, trying to facilitate the interpretation of biogeochemical results, the findings of the manuscript do not add any new information to the community. I would consider reducing the extension of this work and merging it with the companion paper. In the case the work would have to stand from itself, a much deep discussion of the results would be needed. Besides, the reliability of the OMP results has not be proven, by, for example, analyzing the residuals.

The manuscript needs a very careful proofreading and the number of figures/subplots in the main text needs to be reduced.

*2) General comments:*

1. After previous reviewers highlighted the need of a careful proofread of the manuscript, the manuscript still presents grammatical errors and misspellings. To highlight some:
   a. There are still two appearances of STW instead of SWT (lines 325 and 355).
   b. The term "sea water type" still appears in the manuscript instead of "source water type" (line 160 and Table 4 caption).
   c. There are inconsistencies in units. For example, sometimes density units are written as kg m$^{-3}$ and others as kg/m$^3$.
   d. Sloppy proofread can be seen, for example, on lines 181 ("…is that he water masses…"), 306 ("…our analysi.The region…"), 423-424 ("…being important to distinguishe AAIW from Central Waters…"), and 536 ("vintages" of LSW exitst), to enumerate some. Besides, there are grammatical errors, such as those on lines 49 ("there are gradual transformation between them", where "transformation" should be plural), and 52-53 ("Also important is the concepts", it should be either "important is the concept" or "important are the concepts"); and sentences that are unfinished, such as the one on line 126 ("…water masses, since this product is.").
   e. One citation is not correctly spelled in the text (line 68, should be Jacobsen (1927)). There are also few references not cited in the text, such as Clarke et al. (1990); Ishii et al. (2011); Key et al. (2010); Lacan and Jeandel (2004); to enumerate some.
2. To solve the OMP for the whole Atlantic Ocean, the authors split the water column into different regions and layers (as summarized in Table 1), called OMP runs. Some suggested improvements for the OMP analysis:
   a. After changing the description of the Central Waters by using two SWTs, the two SWTs defining each Central Water are not allowed to mix between them between 40N and the Equator (#5 upper and lower), so the transition between the different properties of the Central Waters observed in the ocean cannot be represented by the OMP. Besides, in Table 1 there is no

specification of which of the two SWTs (or if the two SWTs) representing each one Central Waters they use that when solving the OMP system, such as in #6 and #8, for example. Please specify to avoid confusion.

b.  Lines 205-206: If the SWTs allowed to mix in each of the OMP runs have a coherent lateral and vertical distribution not such "step-like" features should appear. Each run should share at least one SWT with the adjacent OMP runs to avoid that issue, which seems to be the case according to Table 1. However, drastic disappearances/appearances of water masses can be observed in the top panels of Figure 19, highlighting the lack of a "transitional water mass" between AABW and NEADW.

c.  Lines 207-214: There is no figure supporting what is discussed there, and what is 100% is the mass conservation itself not the residual, the residual should be 10% or 20% (same for line 36-37). Besides, if all the required SWTs are defined and the weighting of the OMP equations is well performed, an error in the mass conservation of 20%, even 10% should not happen.

d.  There is no discussion about the residuals of the OMP analysis. The residuals of the least square method constrained to non-negative solutions used for an OMP analysis give insights about the reliability of the proposed mixing model, and indicate the quality of the solution.

e.  Figure 6 shows high percentages of ENACW along the Gulf Stream. That highlights the fact that that water mass is formed in the intergyre region (Pollard et al., 1996) and not close to the Iberian Peninsula. Changing the formation region for ENACW would result in a wider temperature and salinity range for ENACW than the one considered in this work.

f.  Figures 7 (WSACW) and 14 (ISOW) show water masses outside the range they should appear. Fig. 7 shows WSACW below $\sigma_\theta$ = 27.00 kg m$^3$, which should not appear according to Table 1 (below that density, OMP run #8 should be applied, which does not include WSACW). In Fig. 14, ISOW seems to extend to surface with percentages around 10% (guessing from the color scale), where it is not allowed according to Table 1. These two facts question how the OMP runs were applied to the dataset.

g.  Both reviewers highlighted the fact that some samples are not accounted by any water type, and no change has been made to solve this issue. This is clearly seen in Fig. 2 (previously Fig. 22).

3.  There is a good explanation on how the regions of water mass formation where selected to determine the SWT properties, but the discussion of the OMP results, i.e., the water mass distributions comparing them against previous works is almost inexistent. If this works wants to stand by itself, it needs a better discussion of the results, presenting what novelties have been found. Some of the information to discuss with is already in the sections describing the formation regions of the water masses.

*3) Minor comments:*

4.  Line 26 and elsewhere: Once the MOW has overflowed the Strait of Gibraltar and has mixed with Atlantic Waters, it is no longer MOW but Mediterranean Water (MW) (see, for example, Carracedo et al., 2016). As in this work the depicted area of formation of MOW west of the Strait of Gibraltar (Fig. 9), please change MOW to MW here and elsewhere.

5.  Introduction: New information has been added to the introduction, but a careful re-organization and summarization needs to be done in this section. The information is presented in a chaotic order, being some information repeated.

6. Lines 156-158: This sentence is confusing. Consider rephrasing something like: "Some WMs need more than one SWT to be defined (Tomczak, 1999), for example Central Waters present a linear temperature-salinity relationship that requires two SWTs for a complete description.".
7. There is still a misuse of the terminologies water mass vs. source water type. For example, on lines 162 and 164 it is used the term WM instead of SWT, as it is correctly used on line 176. For an OMP analysis what is defined are the properties of the SWTs and not of the WMs.
8. On lines 189-191 it is stated that the mixed layer was not considered, but on lines 192-194 it is stated that all the Atlantic data present in Glodapv2 was analyzed. That creates confusion, please merge both sentences.
9. Line 195: to a reader not familiar with OMP it is not that clear why solving the fractions of 6 SWTs is an excessive number. Please clarify that the number of SWT fractions must be lower than the number of properties defining the SWTs in order to solve an overdetermined system of equations.
10. Lines 241-251: as already pointed out by van Heuven, only key figures should be maintained in the main manuscript and the rest should be placed in the supmat. As a guidance, I would leave one of the figures referred in these lines (Figs. 5, 8, 11, 12, 16 and 17) in the main text as an example and move the others to the supplementary material.
11. Line 318: Salinity should be specified as a dimensionless quantity (Unesco, 1986).
12. Lines 726 and 730: should it be "6-dimensional" and "Six often measured" according to the number of properties defining each SWT?
13. Both reviewers rightly point to the fact samples near 34.8 salinity and -1$^{\circ}$C temperature are not represented by the SWTs used in this work. van Heuven and the authors themselves determine that those samples are located in the Norwegian Sea, therefore those samples should not be considered in this work that focusses in the Atlantic Ocean. Therefore, those data points and those of the mix layer that were not being solved by the OMP analysis performed in this work should be removed from Figs. 2, 5, 8, 11, 12, 16, and 17.
14. Figures: Pressure units should be dbar not db.
15. Figure 2: I would recommend adding all the SWTs to the plots, and not only the central points in case of the Central Waters. Adding the two extremes of the TS-relationship would help the reader to know which samples are "enveloped" by the defined SWTs.
16. Figures representing along section properties (Fig. 1) and waters mass distributions (Figs. 7, 8, 10, 14, 15, 19 and 20) will benefit from map insets (as Fig. 3) to avoid the reader to go back and forth to Fig. 2 to know where the section is located. In case such map insets are not added, please state that the cruise tracks are represented in Fig. 2.
17. Please, consider changing the color scale of Figs. 1, 3, 6-10, 12-15 and 18-20 to a colorblind-friendly one, such as the ones in the cmocean package: https://github.com/kthyng/cmocean-odv.

*4) References*

Carracedo, L.I., Pardo, P.C., Flecha, S., and Pérez, F.F.: On the Mediterranean Water Composition. J Phys Oceanogr., 46, 1339–1358, 2016.

Pollard, R.T., Grifftths, M.J., Cunningham, S.A., Read, J.F., Pérez, F.F., Ríos, A.F.: Vivaldi 1991 – a study of the formation, circulation and ventilation of Eastern North Atlantic Central Water. Progress in Oceanography 37, 167–192, 1996.

Unesco: Progress on oceanographic tables and standards 1983–1986: Work and recommendations of the Unesco/SCOR/ICES/IAPSO Joint Panel. Chapter 7.1: Practical Salinity. Unesco Technical Papers in Marine Science 50, p. 9, 1986.

---

## Referee Report (RR2)

**A review of " Characteristics of Water Masses in the Atlantic Ocean based on GLODAPv2 dataset" by Mian Liu and Toste Tanhua.**

This paper is an attempt to provide a very thorough analyses of water masses in the Atlantic Ocean. The authors have clearly done a lot of work, both in analyses, figures and text. Their objective seems to be, to provide some sort of look-up table on Atlantic Water masses that can be used for both physical and biogeochemical community in order to understand the formation and spread of water masses. This in itself is certainly useful, and in that regard, I think this paper is worthy of publication. However, this paper is not *ready* for publication.

It is not a lack of effort that is the problem. The authors have done a substantial job. It is because it is too much and yet too little. Too much because 1) a lot of the text can be shortened and 2) in the sense that they try and cover a lot of water-masses and therefore have to cover a lot of literature. On the other hand, too little because 1) a lot of the method and the science is not well explained or clearly laid out, and 2) they don't cover all the literature for all the WM because that is perhaps impossible.

So, to me, there are two issues: 1) science, 2) presentation and text. The first requires a lot of work, as explained below. Even to just make this work reproducible by others. The second, I'm not sure what is the best option, but currently I think it may be too long and maybe still incomplete. Overall, I think this paper can be reduced to at least 75% of its current length just by being more precise, concise and to the point. Because this paper is already long, this is important. Some comments are also provided below.

**The science**

It is unclear which salinity is used. I assume Practical Salinity. These days we do not use potential temperature and "salinity". We use Conservative Temperature and Absolute Salinity. These can easily be obtained using the TEOS-10 gsw software. Please use these or provide clear arguments why you do not use those variables.

Section 2.2 and 2.3. Line 136 to 174. What exactly is the message of all this text? Basically, I read; "it's difficult to define water masses, but Tomczak did a good job and we use his method". If so, I think this can be a lot shorter. This would free up some space to then properly explain the method. You provide one equation (L175), with little explanation. It is not clear where and how source waters are defined and how the related **G**-matrix would look like and what kind of numbers go into that matrix. Then it is unclear which data go's into **d** and it is also unclear which method is used to find a minimum for **R**. Is this a least-squares inversion? If so, have you looked at the sensitivity to choices in the input parameters, such has how water masses are defined, and how much variables are used? Is any weighting used for the solution? These are all unanswered question that are important for reproducibility of the results.

L181-184 This paragraph is unclear. Please provide numbers. What are "short" transport times, and how "close" is close enough and how to these numbers influence your results.

Section 2.3 and beyond. Nitrate and phosphate seem to have a very similar distribution. Using them both may not add that much information. In line 181-191 you then say you use them to construct a conserved variable. So, are you then using 5 instead of 6 variables? If so, this should be made very clear in the manuscript. Also, for the conclusion section you talk about 7-dimensional space. Is it still if you combined tracers into one? On top of that, what do you do with Oxygen, as that is also non-conservative and, in these lines, you mention this could be a problem.

Because the explanation of L185-L214 is not always clear and the manner by which numbers are obtained is not well explained, I don't understand L207-214. In addition, the authors talk about something in A16 in L208-209, but do not refer to where we can see this.

You define 4 vertical layers based on surface-referenced potential density. First of all, how is this calculated? Do you use the TEOS-10 software? Second, why surface referenced potential density. This is not accurate beyond 500 meters depth for WM analyses. Please use Neutral Density, which is perhaps the best we currently have.

L261-278 Is this about figure 5 and beyond. It is not clear what the message of this paragraph is and where it belongs. Either remove it or clarify what the purpose is.

**The writing and presentation**

Th authors have provided many figures with a lot of information. A lot of work has been done to do this properly. Still some improvements can be made. Overall, I think that the text needs to be written more concise, precise and to the point and can reduce to 75% of its current length.

**Introduction:** The authors attempt to write a little bit of history on the subject. It seems incomplete and maybe not necessary to the extend done here. It is partly a matter of style, but partly also a matter of being precise, concise and to the point. So, I think the introduction can shrink at last 25% and still convey the same information. Perhaps consider reading Groeskamp et al 2019, it provides a history on WM analyses and WM transformation.

**Section 4,5,6,7**

Each WM is introduced with some literature background. That is great. However, because so many WMs are considered, this of course requires a lot of literature study. I think the current references are all pretty old and some new insights can be included, from more recent studies. I can give one example of a WM which I'm more familiar with. For AAIW. Consider these papers: Over all

- Saenko, O. A., and A. J. Weaver (2001), Importance of wind-driven sea ice motion for the formation of antarctic intermediate water in a global climate model, Geophys. Res. Lett., 28(21), 4147–4150, doi:10.1029/2001GL013632.
- Sallee, J.-B., K. Speer, S. Rintoul, and S. Wijffels (2010), Southern Ocean thermocline ventilation, J. Phys. Ocean., 40(3), 509–529, doi:10.1175/2009JPO4291.1.
- Nycander, J., M. Hieronymus, and F. Roquet (2015), The nonlinear equation of state of sea water and the global water mass distribution, Geophys. Res. Lett., 42(18), 7714–7721, doi:10.1002/2015GL065525.
- Abernathey, R. P., I. Cerovecki, P. R. Holland, E. Newsom, M. Mazloff, and L. D. Talley (2016), Watermass transformation by sea ice in the upper branch of the southern ocean overturning, Nat. Geosci., 9, 596–601, doi:10.1038/ngeo2749.
- Groeskamp, S., R. P. Abernathey, A. Klocker (2016), Water Mass Transformation by Cabbeling and Thermobaricity. Geophysical Research Letters

I'm sure such additional work could be done for most WMs considered here. Now I'm not sure how much of this work you need to do to provide a reasonable background. Eventually, I'll leave it up to the authors to decide if the current version is god enough or needs more work on that.

Line 115-134 can be merged into one brief paragraph half the size.

L181 - What is internally consistent? I don't think this is a useful description.

L241 – During the narrative of each water mass. What does that mean?

L244 – which colour coding?

L256-260 – Good point, but not very clearly explained. Please try again.

**Figure 2:** When Figure 2 is first mentioned in text, SWT is not yet defined. But it is used in the caption. That should be clarified. It is unclear where the colours stand for. Please provide link to abbreviations in caption, they have not been discussed yet. Please provide in caption, the clarification that the middle panel is a zoom of the box in the left panel. The letters in the light blue can't be read.

A few examples of incomplete, misspelled, or weird sentences. This needs work:

- L181 that the
- L194..
- L211 weird sentence
- Where is section 8?

---

## Referee Report (RR3)

**A review of "Water Masses in the Atlantic Ocean: Characteristics and Distributions" by Mian Liu and Toste Tanhua [version 5].**

This is a second review after revisions of the previous manuscript " Characteristics of Water Masses in the Atlantic Ocean based on GLODAPv2 dataset" [version 4].

This paper uses an inverse-model to define water masses in the Atlantic Ocean. I think it is much better than the previous version. Results and figures are clear, writing is clear, and order is logic. The main thing to do is to add a more details about the method used, and some furthermore minor changes. This paper is almost ready for publication.

**Method**

The paper uses a least square analyses, equivalent to an inverse method without any weights or constraints, but with prior choices. Hence, normally one would write an inverse method including prior constraints as (McIntosh and Rintoul 1997, Menke 1984, Groeskamp et al 2014 [from which this equation is copy pasted]):

$$\mathbf{x} = \mathbf{x}_0 + \mathbf{W}_c^2 \mathbf{A}^{\mathrm{T}} (\mathbf{A}\mathbf{W}_c^2 \mathbf{A}^{\mathrm{T}} + \mathbf{W}_r^{-2})^{-1}(\mathbf{b} - \mathbf{A}\mathbf{x}_0).$$

The results are this sensitive to weighting of both the equations (Wr, now all equally weighted), weighing of the unknowns (Wc, also equally weights), and choice of prior knowledge (x0). In this paper, the prior knowledge is inputted by the SWT's. Hence, the results can be sensitive to this, and no such sensitivity is currently explored. Regardless I'm reasonably convinced that the results are fine because it looks good. But with an inverse method, you have to explore the sensitivity to the input variables that we choose. Long story short: you might want to either study this sensitivity or otherwise mention that this is not been looked at.

- McIntosh, P. C., and S. R. Rintoul, 1997: Do box inverse models work? J. Phys. Oceanogr., 27, 291–308, doi:10.1175/1520-0485(1997)027,0291:DBIMW.2.0.CO;2.
- Menke, W., 1984: Geophysical Data Analysis: Discrete Inverse Theory. Academic Press, 260 pp.
- Groeskamp, S., J. D. Zika, , B. M. Sloyan, T. J. McDougall, and P. C. McIntosh, 2014: A thermohaline inverse method for estimating diathermohaline circulation and mixing. J. Phys. Oceanogr., 44, 2681–2697, doi:10.1175/JPO-D-14-0039.1.

More minor comments can be found annotated in PDF of the paper.

---

## Author Response (AR2)

**Reply to the comments from Report 1 (Anonymous Referee #3)**

Peer review report on "Characteristics of Water Masses in the Atlantic Ocean based on GLODAPv2 dataset" by Mian Liu and Toste Tanhua

**1) Overview and general recommendation:**

Understanding the formation, transformation, and circulation of water masses has been a hot topic in oceanography since its start. Many ways of tangling this oceanography field have been developed, from the mere description of hydrographic properties to statistical and numerical models. This manuscript presents the water mass structure of the Atlantic Ocean resulting from an Optimum Multiparameter (OMP) analysis.

Although the intention of the manuscript is honorable, trying to facilitate the interpretation of biogeochemical results, the findings of the manuscript do not add any new information to the community. I would consider reducing the extension of this work and merging it with the companion paper. In the case the work would have to stand from itself, a much deep discussion of the results would be needed. Besides, the reliability of the OMP results has not be proven, by, for example, analyzing the residuals.

The manuscript needs a very careful proofreading and the number of figures/subplots in the main text needs to be reduced.

**2) General comments:**

1. After previous reviewers highlighted the need of a careful proofread of the manuscript, the manuscript still presents grammatical errors and misspellings. To highlight some:

   We have carefully proofread the whole manuscript, hopefully with fresh eyes so that we caught any remaining errors.

   a. There are still two appearances of STW instead of SWT (lines 325 and 355).

      Two mistakes have been fixed and all "STW" in this manuscript are guaranteed to be replaced by SWT.

   b. The term "sea water type" still appears in the manuscript instead of "source water type" (line 160 and Table 4 caption).

      Two mistakes have been fixed and all "sea water types" in this manuscript are guaranteed to be replaced by "source water type".

   c. There are inconsistencies in units. For example, sometimes density units are written as kg m$^{-3}$ and others as kg/m$^{3}$.

      All the unit formats have been unified as kg m$^{-3}$ or μmol kg$^{-1}$

   d. Sloppy proofread can be seen, for example, on lines 181 ("…is that he water masses…")

      Checked, "the water masses".

      306 ("…our analysi. The region…"),

Checked, "our analysis".

423-424 ("…being important to distinguishe AAIW from Central Waters…"),

Checked, "being important to distinguish AAIW".

and 536 ("vintages" of LSW exitst), to enumerate some.

Checked, "vintages" of LSW exist".

Besides, there are grammatical errors, such as those on lines 49 ("there are gradual transformation between them", where "transformation" should be plural),

Checked, "gradual transformations between them" in plural form.

and 52-53 ("Also important is the concepts", it should be either "important is the concept" or "important are the concepts");

Checked, "Also important is the concept" in singular form.

and sentences that are unfinished, such as the one on line 126 ("…water masses, since this product is.").

Checked and the sentence has been completed.

e. One citation is not correctly spelled in the text (line 68, should be Jacobsen (1927)).

Checked, Jacobsen (1927).

There are also few references not cited in the text, such as Clarke et al. (1990); Ishii et al. (2011); Key et al. (2010); Lacan and Jeandel (2004); to enumerate some.

The references are checked again.

2. To solve the OMP for the whole Atlantic Ocean, the authors split the water column into different regions and layers (as summarized in Table 1), called OMP runs. Some suggested improvements for the OMP analysis:

a. After changing the description of the Central Waters by using two SWTs, the two SWTs defining each Central Water are not allowed to mix between them between 40N and the Equator (#5 upper and lower), so the transition between the different properties of the Central Waters observed in the ocean cannot be represented by the OMP. Besides, in Table 1 there is no specification of which of the two SWTs (or if the two SWTs) representing each one Central Waters they use that when solving the OMP system, such as in #6 and #8, for example. Please specify to avoid confusion.

Additional text is added to explain the divisions of OMP runs, including in both horizontal and vertical directions and also the reasons.

b. Lines 205-206: If the SWTs allowed to mix in each of the OMP runs have a coherent lateral and vertical distribution not such "step-like" features should appear. Each run should share at least one SWT with the adjacent OMP runs to avoid that issue, which seems to be the case according to Table 1. However, drastic disappearances/appearances of water masses can be observed in the top panels of Figure 19, highlighting the lack of a "transitional water mass" between AABW and NEADW.

Thanks for the suggestion. The OMP calculations have been redone with new criteria and now the "step-like" features between each runs were removed, especially in the A16 section. In the bottom layer, distributions of AABW and NEABW are more coherent (top panels of Figure 19). In the case of AABW and NEABW the steps do occur since we, for reasons outlined in the paper, consider a northern version of AABW north of the equator. Normally we have water masses in several boxes to avoid this, but the OMP can only handle so many water masses in one run.

c. Lines 207-214: There is no figure supporting what is discussed there, and what is 100% is the mass conservation itself not the residual, the residual should be 10% or 20% (same for line 36-37). Besides, if all the required SWTs are defined and the weighting of the OMP equations is well performed, an error in the mass conservation of 20%, even 10% should not happen.

This part is discussed together with an additional section. Mass residual is inevitable in the central water, even in the paradigm of the OMP founders (see figure below). The reason is also explained in the data and method section. The key properties, for instance CT, of Central Waters are variable. When the CT increases beyond the range of this water mass, the OMP analysis considers the fraction is over 100%, even more than 120%. In the other case, the new OMP calculation has been limited this situation in our manuscript within a small number (~8% of the total samples) of samples and values no more than 105%.

[Figure]

d. There is no discussion about the residuals of the OMP analysis. The residuals of the least square method constrained to non-negative solutions used for an OMP analysis give insights about the reliability of the proposed mixing model, and indicate the quality of the solution.

This point is discussed in the same paragraph together with c.

e. Figure 6 shows high percentages of ENACW along the Gulf Stream. That highlights the fact that that water mass is formed in the intergyre region (Pollard et al., 1996) and not close to the Iberian Peninsula. Changing the formation region for ENACW would result in a wider temperature and salinity range for ENACW than the one considered in this work.

Thanks for this helpful suggestion. The formation area is changed to the west according to Pollard et al., 1996. (Figure 5a and Figure 7 left panel), and the property range and distribution of ENACW become wider.

f. Figures 7 (WSACW) and 14 (ISOW) show water masses outside the range they should appear. Fig. 7 shows WSACW below $\sigma_\theta = 27.00$ kg m3, which should not appear according to Table 1 (below that density, OMP run #8 should be applied, which does not include WSACW). In Fig. 14, ISOW seems to extend to surface with percentages around 10% (guessing from the color scale), where it is not allowed according to Table 1. These two facts question how the OMP runs were applied to the dataset.

The drawing process of Figure 7 and all the other section plots is divided into two steps. In the first step, OMP analysis is used to calculate the water mass fractions in each sampling point and then splicing together. (The figure below shows an example from A16 section.) In the second step, objmap function in MATLAB is used for interpolation calculation to draw the section plot. Therefore, the reason why WNACW appears below $\sigma_\theta = 27.00$ kg m$^{-3}$ (new boundary is $\gamma = 27.10$ kg m$^{-3}$) is not because of OMP analysis, but the result from the objmap in the interpolation calculations from all the sampling points.

[Figure]

Thanks to the reviewer for pointing out my negligence in my work. I think it is reasonable for about 10-20% of WNACW still to appear on the boundary of $\sigma_\theta = 27.00$ kg m$^{-3}$ (new boundary is $\gamma = 27.10$), while it is definitely wrong for ISOW to appear on the surface at any fraction. But this is not the error of OMP analysis, but the vulnerability of the objmap interpolation calculation. Therefore, the calculation method of objmap was adjusted in the new calculation, and such mistake has been avoided.

g. Both reviewers highlighted the fact that some samples are not accounted by any water type, and no change has been made to solve this issue. This is clearly seen in Fig. 2 (previously Fig. 22).

One paragraph is added in the discussion section to explain this point.

3. There is a good explanation on how the regions of water mass formation where selected to determine the SWT properties, but the discussion of the OMP results, i.e., the water mass distributions comparing them against previous works is almost inexistent. If this works wants to stand by itself, it needs a better discussion of the results, presenting what novelties have been found. Some of the information to discuss with is already in the sections describing the formation regions of the water masses.

The text is reorganized in the new manuscript by adding the results and comparisons to the previous investigations and references.

**3) Minor comments:**

4. Line 26 and elsewhere: Once the MOW has overflowed the Strait of Gibraltar and has mixed with Atlantic Waters, it is no longer MOW but Mediterranean Water (MW) (see, for example, Carracedo et al., 2016). As in this work the depicted area of formation of MOW west of the Strait of Gibraltar (Fig. 9), please change MOW to MW here and elsewhere.

   Accepted, Mediterranean Overflow Water and MOW are changed into Mediterranean Water and MW.

5. Introduction: New information has been added to the introduction, but a careful re-organization and summarization needs to be done in this section. The information is presented in a chaotic order, being some information repeated.

   The introduction section has been reorganized.

6. Lines 156-158: This sentence is confusing. Consider rephrasing something like: "Some WMs need more than one SWT to be defined (Tomczak, 1999), for example Central Waters present a linear temperature-salinity relationship that requires two SWTs for a complete description.".

   Thanks for the suggestion, the expressions has been changed.

7. There is still a misuse of the terminologies water mass vs. source water type. For example, on lines 162 and 164 it is used the term WM instead of SWT, as it is correctly used on line 176. For an OMP analysis what is defined are the properties of the SWTs and not of the WMs.

   Accepted, the distinguish between SWTs and WMs is further clarified in the new version.

8. On lines 189-191 it is stated that the mixed layer was not considered, but on lines 192-194 it is stated that all the Atlantic data present in Glodapv2 was analyzed. That creates confusion, please merge both sentences.

   Checked and improved.

9. Line 195: to a reader not familiar with OMP it is not that clear why solving the fractions of 6 SWTs is an excessive number. Please clarify that the number of SWT fractions must be lower than the number of properties defining the SWTs in order to solve an overdetermined system of equations.

   Accepted, and an explanation is added after consulting Karstenson, the founder of the OMP method .

10. Lines 241-251: as already pointed out by van Heuven, only key figures should be maintained in the main manuscript and the rest should be placed in the supmat. As a guidance, I would leave one of the figures referred in these lines (Figs. 5, 8, 11, 12, 16 and 17) in the main text as an example and move the others to the supplementary material.

    Accepted, and the figure plots are reorganized.

11. Line 318: Salinity should be specified as a dimensionless quantity (Unesco, 1986).

Absolute Salinity (g kg$^{-1}$) is now used in the new version.

12. Lines 726 and 730: should it be "6-dimensional" and "Six often measured" according to the number of properties defining each SWT?

Accepted and changed.

13. Both reviewers rightly point to the fact samples near 34.8 salinity and -1oC temperature are not represented by the SWTs used in this work. van Heuven and the authors themselves determine that those samples are located in the Norwegian Sea, therefore those samples should not be considered in this work that focusses in the Atlantic Ocean. Therefore, those data points and those of the mix layer that were not being solved by the OMP analysis performed in this work should be removed from Figs. 2, 5, 8, 11, 12, 16, and 17.

These points are now removed from the OMP analysis and additional explanation is added in the end of the discussion section.

14. Figures: Pressure units should be dbar not db.

Accepted and changed.

15. Figure 2: I would recommend adding all the SWTs to the plots, and not only the central points in case of the Central Waters. Adding the two extremes of the TS-relationship would help the reader to know which samples are "enveloped" by the defined SWTs.

Accepted and changed.

16. Figures representing along section properties (Fig. 1) and waters mass distributions (Figs. 7, 8, 10, 14, 15, 19 and 20) will benefit from map insets (as Fig. 3) to avoid the reader to go back and forth to Fig. 2 to know where the section is located. In case such map insets are not added, please state that the cruise tracks are represented in Fig. 2.

Accepted and added.

17. Please, consider changing the color scale of Figs. 1, 3, 6-10, 12-15 and 18-20 to a colorblind-friendly one, such as the ones in the cmocean package: https://github.com/kthyng/cmocean-odv.

The color scale is changed and we hope this time can be seen more clearly, especially the fractions of water masses, which is the main point of this study.

**4) References**

Carracedo, L.I., Pardo, P.C., Flecha, S., and Pérez, F.F.: On the Mediterranean Water Composition. J Phys Oceanogr., 46, 1339–1358, 2016.

Pollard, R.T., Grifftths, M.J., Cunningham, S.A., Read, J.F., Pérez, F.F., Ríos, A.F.: Vivaldi 1991 – a study of the formation, circulation and ventilation of Eastern North Atlantic Central Water. Progress in Oceanography 37, 167–192, 1996.

Unesco: Progress on oceanographic tables and standards 1983–1986: Work and recommendations of the Unesco/SCOR/ICES/IAPSO Joint Panel. Chapter 7.1: Practical Salinity. Unesco Technical Papers in Marine Science 50, p. 9, 1986.

Checked and corrected.

**Reply to the comments from Report 2 (Referee #4: Groeskamp, Sjoerd)**

This paper is an attempt to provide a very thorough analyses of water masses in the Atlantic Ocean. The authors have clearly done a lot of work, both in analyses, figures and text. Their objective seems to be, to provide some sort of look-up table on Atlantic Water masses that can be used for both physical and biogeochemical community in order to understand the formation and spread of water masses. This in itself is certainly useful, and in that regard, I think this paper is worthy of publication. However, this paper is not ready for publication.

It is not a lack of effort that is the problem. The authors have done a substantial job. It is because it is too much and yet too little. Too much because 1) a lot of the text can be shortened and 2) in the sense that they try and cover a lot of water-masses and therefore have to cover a lot of literature. On the other hand, too little because 1) a lot of the method and the science is not well explained or clearly laid out, and 2) they don't cover all the literature for all the WM because that is perhaps impossible.

So, to me, there are two issues: 1) science, 2) presentation and text. The first requires a lot of work, as explained below. Even to just make this work reproducible by others. The second, I'm not sure what is the best option, but currently I think it may be too long and maybe still incomplete. Overall, I think this paper can be reduced to at least 75% of its current length just by being more precise, concise and to the point. Because this paper is already long, this is important. Some comments are also provided below.

**The science**

It is unclear which salinity is used. I assume Practical Salinity. These days we do not use potential temperature and "salinity". We use Conservative Temperature and Absolute Salinity. These can easily be obtained using the TEOS-10 gsw software. Please use these or provide clear arguments why you do not use those variables.

Thanks for the suggestion. In the new version, Conservative Temperature (CT) and Absolute Salinity (SA) are used instead of Potential Temperature and Practical Salinity.

Section 2.2 and 2.3. Line 136 to 174. What exactly is the message of all this text? Basically, I read; "it's difficult to define water masses, but Tomczak did a good job and we use his method". If so, I think this can be a lot shorter. This would free up some space to then properly explain the method.

Thanks for the suggestion, the text is now simplified.

You provide one equation (L175), with little explanation. It is not clear where and how source waters are defined and how the related G-matrix would look like and what kind of numbers go into that matrix. Then it is unclear which data go's into d and it is also unclear which method is used to find a minimum for R. Is this a least-squares inversion? If so, have you looked at the sensitivity to choices in the input parameters, such has how water masses are defined, and how much variables are used? Is any weighting used for the solution? These are all unanswered question that are important for reproducibility of the results.

A detailed explanation is added in the manuscript.

L181-184 This paragraph is unclear. Please provide numbers. What are "short" transport times, and how "close" is close enough and how to these numbers influence your results.

A specific number or transport time is difficult to define, in the new manuscript a general range or scale, for instance an oceanic front or basin-wide scale, is given according to Karstensen and Tomczak, (1998).

Section 2.3 and beyond. Nitrate and phosphate seem to have a very similar distribution. Using them both may not add that much information. In line 181-191 you then say you use them to construct a conserved variable. So, are you then using 5 instead of 6 variables? If so, this should be made very clear in the manuscript.

Correct, in the analysis, 5 are used instead of 6, but was not reflected and explained in the previous manuscript. This was a mistake in the work. Run 5 and 7 are now more subdivided in Table 1 and explained in the text in the new manuscript.

Also, for the conclusion section you talk about 7-dimensional space. Is it still if you combined tracers into one? On top of that, what do you do with Oxygen, as that is also non-conservative and, in these lines, you mention this could be a problem.

Accepted, changed to 6.

Because the explanation of L185-L214 is not always clear and the manner by which numbers are obtained is not well explained, I don't understand L207-214. In addition, the authors talk about something in A16 in L208-209, but do not refer to where we can see this.

One additional section is added to give a detailed explanation for the OMP analysis.

You define 4 vertical layers based on surface-referenced potential density. First of all, how is this calculated? Do you use the TEOS-10 software? Second, why surface referenced potential density. This is not accurate beyond 500 meters depth for WM analyses. Please use Neutral Density, which is perhaps the best we currently have.

Thanks for the suggestion, Neutral Density is now used instead of Potential Density.

L261-278 Is this about figure 5 and beyond. It is not clear what the message of this paragraph is and where it belongs. Either remove it or clarify what the purpose is.

The purpose of this paragraph is to clarify some information in Figure 6 and 7 and beyond, including the Expocode, Stations, Sections, and also the source of the information. But this doesn't seem to help much with the topic of the study, and most of the information is available to the reader from the figure legends, so it might be a better choice to remove this paragraph.

The writing and presentation

The authors have provided many figures with a lot of information. A lot of work has been done to do this properly. Still some improvements can be made. Overall, I think that the text needs to be written more concise, precise and to the point and can reduce to 75% of its current length.

**Introduction:** The authors attempt to write a little bit of history on the subject. It seems incomplete and maybe not necessary to the extend done here. It is partly a matter of style, but partly also a matter of being precise, concise and to the point. So, I think the introduction can shrink at last 25% and still convey the same information. Perhaps consider reading Groeskamp et al 2019, it provides a history on WM analyses and WM transformation.

Thanks for the suggestion, the Introduction section are reorganized and the text has been shortened.

**Section 4,5,6,7**

Each WM is introduced with some literature background. That is great. However, because so many WMs are considered, this of course requires a lot of literature study. I think the current references are all pretty old and some new insights can be included, from more recent studies. I can give one example of a WM which I'm more familiar with. For AAIW. Consider these papers: Over all

- Saenko, O. A., and A. J. Weaver (2001), Importance of wind-driven sea ice motion for the formation of antarctic intermediate water in a global climate model, Geophys. Res. Lett., 28(21), 4147–4150, doi:10.1029/2001GL013632.

- Sallee, J.-B., K. Speer, S. Rintoul, and S. Wijffels (2010), Southern Ocean thermocline ventilation, J. Phys. Ocean., 40(3), 509–529, doi:10.1175/2009JPO4291.1.

- Nycander, J., M. Hieronymus, and F. Roquet (2015), The nonlinear equation of state of sea water and the global water mass distribution, Geophys. Res. Lett., 42(18), 7714–7721, doi:10.1002/2015GL065525.

- Abernathey, R. P., I. Cerovecki, P. R. Holland, E. Newsom, M. Mazloff, and L. D. Talley (2016),Water-mass transformation by sea ice in the upper branch of the southern ocean overturning, Nat. Geosci., 9, 596–601, doi:10.1038/ngeo2749.

- Groeskamp, S., R. P. Abernathey, A. Klocker (2016), Water Mass Transformation by Cabbeling and Thermobaricity. Geophysical Research Letters

Thanks for the helpful suggestion, and new criteria are redefined in the OMP analysis.

I'm sure such additional work could be done for most WMs considered here. Now I'm not sure how much of this work you need to do to provide a reasonable background. Eventually, I'll leave it up to the authors to decide if the current version is god enough or needs more work on that.

Correct, there is further work to be done on the water masses. In the next step, authors intend to introduce transient tracers (CFCs and $SF_6$) based on this work. For now, the current work can support the next step and after that, the authors will try to investigate the water masses in a meticulous and deep-going way in order to support the demand of biogeochemistry research.

Line 115-134 can be merged into one brief paragraph half the size.
Accepted, and the text is reorganized in the new version of manuscript.

L181 - What is internally consistent? I don't think this is a useful description.
The description is reorganized.

L241 – During the narrative of each water mass. What does that mean?
Expression changed, during definition of SWTs.

L244 – which colour coding?
Explained, all the selected data (blue dots) in Figure 6 and 7.

L256-260 – Good point, but not very clearly explained. Please try again.
Accepted, and the explanation is reorganized in the new manuscript.

**Figure 2:** When Figure 2 is first mentioned in text, SWT is not yet defined. But it is used in the caption. That should be clarified. It is unclear where the colours stand for. Please provide link to abbreviations in caption, they have not been discussed yet. Please provide in caption, the clarification that the middle panel is a zoom of the box in the left panel. The letters in the light blue can't be read.

Correct, the order of the figures has been rearranged and more clear instructions have been added, in addition, the details of the figures have been changed.

A few examples of incomplete, misspelled, or weird sentences. This needs work:

- L181 that the
- L194 . .
- L211 weird sentence
- Where is section 8?

Checked and corrected.

**Water Masses in the Atlantic Ocean**
**: Characteristics and Distributions**

Mian Liu[1, 2]

Toste Tanhua[2, *]

*[1] College of Ocean and Earth Sciences,*

*Xiamen University, Xiamen, 361005, China*

*[2] GEOMAR Helmholtz Centre for Ocean Research Kiel,*

*Marine Biogeochemistry, Chemical Oceanography*

*Düsternbrooker Weg 20, 24105 Kiel, Germany*

*Correspondence to: T. Tanhua (ttanhua@geomar.de)*

[revised manuscript text omitted]

 (e.g. Montgomery, 1958, Helland-Hansen, 1916)

horizontal, and thus a quantifiable volume. Since each water mass is surrounded by other water masses, mixing occurs inevitably between, them both along and across density surfaces. As a result, mixtures of water masses with different properties tend to be found away from their formation areas. Early work by Jacobson (1927) and Defant (1929) clarified the application of T-S relationship in the oceanography, and Wüst and Defant (1936) illustrated the stratification and circulation of water masses in the Atlantic Ocean based on the observational data from Meteor Cruise 1925-1927. Since the first publication of global distributions of water masses (Sverdrup, 1942), early studies on water masses are mainly based on potential temperature ($\theta$) and salinity (S). Emery and Meincke (1986) made a summary and review on this kind of analysis. The limitation of this method is that distribution of more (more than three) water masses cannot be calculated at the same time with only these two parameters. So during the same time as the development of this theory, physical and chemical oceanographers also tried to add more parameters to the calculation and the Optimum Multi-parameter (OMP) analysis is one of the typical products (Poole & Tomczak, 1999). This concept of water masses has been redefined over time, and in Emery and Meincke (1986) the water masses were divided into upper, intermediate and deep/abyssal layers based on the depth and the T-S relationship. With the development of observational capacities for a range of variables and the data base to evaluate them, definition of water masses is not limited to the T-S-P relationship. Additional physical and chemical properties, both conservative and non-conservative, are added to the water mass concept (e.g. Tomczak, 1981). 
[revised manuscript text omitted]

volume in the ocean. In their formation region they have exclusive occupation of a particular part of the
ocean; elsewhere they share the ocean with other water masses with which they mix. The total volume
of a water mass is given by the sum of all its elements regardless of their location."

Tomczak (1999) also introduces the concept of "whereas Source Water Types (SWTs). The SWTs)
describe "the original properties of water masses in their formation areas, and can thus be considered as
the original characteristics of water masses and are also indispensable in labeling distributions of water
masses (Tomczak, 1999). One difference". The distinction between the WMWMs and SWT
conceptsSWTs is that water massesWMs define a physical extentextents, i.e. a volume, while SWTs are
only mathematical definitions, i.e. SWTs are defined values of properties, or values without physical
extent. In other words, water masses, as an objective existence in the ocean, have their temporal and
spatial distributions, while SWTs, as an artificial definition, are only a concept of values. Knowledge of
the SWTs, on the other hand, is essential in labeling water massesWMs, tracking their spreading or
mixing progresses, since thesethe values from SWTs describe thetheir initial characteristics of water
masses. Or rather, the SWTs canand can be considered as the fingerprints of water masses and all the
water masses can still be identified by their own SWTs, even when they spread distance away from their
formation areas. The SWT of a water mass can be defined by several numbers and each number shows
one key property (WMs. The SWT of a WM is defined by the values of key properties, while some of
them, like Central Water covers a range between two numbers) by quantifying the concentration
(Tomczak, 1999).Waters, require more than one SWT to be defined (Tomczak, 1999). In this studywe
will use, the terminology "water mass" is used in the discussions, realizing that the properties of the
water massesWMs used for the water massfurther analysis reallyactually refer to sea water types. SWTs.

**2.3   OMP Analysis**

**2.3.1 Principle of OMP Analysis**

For the analysis in this study we used . six key properties are used to define water masses.  Those
includeSWTs, including two conservative properties (potential (conservative temperature and absolute
salinity) and four non-conservative (oxygen, silicate, phosphate and nitrate) properties to define the
water masses.. In order to determine the distribution of water masses,distributions of WMs, the OMP
analysis is invoked as objective mathematical formulations of the influence of mixing is used. As a
summary(Karstensen and practical use of Tomczak, 1997; 1998).  The starting point is the above results,
the OMP analysis was developed and successfully applied in the analysis of water masses in specific
regions (e.g. Karstensen & Tomczak, 1998, Karstensen & Tomczak, 1997). Observed properties6 key
properties (Figure 1) from a water sample are compared with observational sampling (such as $CT_{obs}$ is
the observational conservative temperature). The OMP model determines the contributions from
predefined SWTs (such as $CT_i$ shows the properties of conservative temperature in each "pure" water
masses to quantifySWT), which represent the mix of water masses that constitutes a water sample. The
theory and formulas in values of the OMP analysis are described in detail in Tomczak and Large (1989)

and the website http://omp.geomar.de/. Here we make a brief introduction to the OMP calculation that relates directly to our research. OMP analysis is based on a simple model of "unmixed" WMs in formation areas, through a linear set of following mixing equations, assuming that all key properties of water masses are affected by the same mixing process, and then to determine the distribution and of water masses through the following linear equationprocesses. 
[revised manuscript text omitted]

as the benchmark (Figure 4) instead of potential temperature since density plays a more significant role
in the vertical stratification. This way we define a range for the variable within each water mass. SWTs
to label Central Waters (Figure 6).

During the narrativedetermination of each water mass we display fourSWT, two figures are displayed
to characterize them. 1) Maps where all GLODAPv2 station locations are marked as light gray dots,
including a) Depth profiles of the 6 key properties under consideration (same color coding), and stations
within the area of formation that we consider are marked in red and stations with anyb) Bar plots from
the distributions of the samples within the desired properties as defined by Table 2 in blue. 2) The T-S
relationship with the same color coding. 3) Depth profiles of the 6 variables under consideration (same
color coding), and 4) Bar plots of the distribution of the samples within the criteriacriteria (the blue dots
in Figure 6 and 7) for a SWT. In the bar plot, with a Gaussian curve is added to show the statistic results
(the mean value and standard deviation) to the selected data (blue dots), and the amplitude of the curve
is set to be 2/3 of the highest bar.statistics (Figure 7). The plots of properties vs pressure provides an
intuitive understanding of each SWT compared to othersother WMs in the same region. The
distributions of properties distribution and with the Gaussian curve will be helpfulcurves are the basis
to visually determine and confirm the SWT property values and associated standard deviationdeviations.

Most water masses maintain their original characteristics from thetheir formation areas. However, it
issome are, worthy to mention that there are also some water masses, especially in the deep and bottom
layer, which arebe mentioned as products from mixing of several original water masses (for instance,
North Atlantic Deep Water is the product from Labrador Sea Water, Iceland-Scotland Overflow Water
and Denmark Strait Overflow Water). Also, characteristics of some water masses changes sharply
during their pathways (for instancenamely, the sharp drop silicate concentration of Antarctic Bottom
Water after passing the equator). As a result, it is usefulnecessary to redefine these water masses.their
SWTs. In order to distinguish such water masses from the other original water masses, theones, their
defined specific areas are mentioned as "redefining" areas instead of formation areas, because these
water masses, strictly speaking, they are not "formed" in these areas.

In this section, the horizontal (map views) and vertical (sections) distributions of the main water masses
are also displayed. On the maps of horizontal view, fractions of water masses are plotted at each station
with the interpolated format at their core densities. In order to avoid large interpolation errors, a station
is considered as without data and plotted as grey rather than colored dots if there is no data within ±0.1
kg/m³ from core density. To exemplify the vertical distribution of the water masses, sections from five
selected WOCE/GO-SHIP cruises, which together provide a reasonable representation of the Atlantic
Ocean, are displayed (Figure 2 right panel). The A16 section, covering the full north-south extend of the
Atlantic Ocean (Expocodes: 33RO20130803 & 33RO20131223), shows the meridional overview of all
the main water masses. The A03 (Expocode: 74AB20050501) and A10 (Expocode: 33RO20110906)

sections displays the zonal distribution of the water masses in the North (A03) and South (A10) Atlantic
separately. The A25 (Expocode: 06MM20060523) section is located at a relative higher latitude region
compared to the A03 section and better represent the deep and overflow waters in particular. From this
cruise, we focus on the investigation of LSW, ISOW and DSOW, with the purpose to show the origin
of upper and lower NADW. The SR04 (Expocode: 06AQ20101128) is a section in the Antarctic region
near Weddell Sea with certain significance to show the origin and formation of AABW. We also show
a rough overview figure illustrating the main currents in that density layer, and with the main formation
region of each water mass indicated as striped boxes.

**4 The Upper Layer, Central Waters**

The Upper Layerupper layer is occupied by four water masses called Central Waters that are known to
be formed by winter subduction. Central Waters have with upper and lower boundaries instead of point
values. Inproperties. Statistics are done for all the figures, values between boundaries are and statistics
are done to calculate the meanmeans and standard deviations and plotted as the other deep water masses.
However, in the further OMP analysis, all the values between the boundaries are considered as Central
Waters, that means each Central Water (Figure 7 and Figure 1—4 in Supplement) and occupies two
''positions''SWTs in selected one OMP analysis. run.

Central Waters can be easily recognized by their linear T-—S relationships (Pollard et al., 1996,
Stramma & England, 1999).(Pollard et al., 1996; Stramma and England, 1999). In this study, the Upper
Layerupper layer is defined to be located above the potentialneutral density isoline of 27.010 kg/m³ m⁻³
(below the mixed layer). The formationformations and transporttransports of the Central Water isWaters
are influenced by the currents in thisthe upper layer and finally forms aform relative distinct bodybodies

[revised manuscript text omitted]

 (Peterson & Stramma, 1991).  (Deruijter, 1982, Lutjeharms & van Ballegooyen, 1988) (Gordon et al., 1992, Stramma & Peterson, 1990)

(Harvey, 1982, Alvarez et al., 2014,
Emery & Meincke, 1986)

**4.5     Atlantic Distribution of Central Waters**

Based on the OMP analysis on the GLODAPv2 data product, the physical
extent of the Central Waters can be described over the Atlantic Ocean. The horizontal
distributions of four Central Waters in the upper layer are shown on the
maps in Figure 8 and the vertical distributions along selected GO-SHIP sections are found
in Figure 9. Note that the
Central Waters are found at different densities,
the eastern
variations being denser, so that the there is significant overlap in the horizontal distribution. The vertical
extent of the Central Waters is clearly seen in Figure 9.

The ENACW is mainly found in the northeast part of North
Atlantic, near the formation area in the inter-gyre region (Figure 8). High
fractions of ENACW is also found in a band across the Atlantic at around 40 °N, where the core of this
water mass is found at close to 1000 m depth in the western part of the basin (Figure 9).

The WNACW is predominantly found in the western basin of the
North Atlantic in a zonal band between ~-10 °N and 40 °N (Figure 8). The vertical extent
of WNACW is significantly higher in the western basin with an extent of about 500 meter in the west,
tapering off towards the east (Figure 9).

The ESACW is found over most of the South Atlantic, as well as
in the tropical and subtropical north Atlantic (Figure 8). The extent of ESACW do reach particular  far
north in the eastern part of the basin where it is an important component over the Eastern Tropical North
Atlantic Oxygen Minimum Zone, roughly south of the Cape Verde Islands. In the vertical
direction, the ESACW is located below WSACW (Figure 9).

The horizontal distribution of the WSACW does also
reach into the northern hemisphere but is, obviously, concentrated in the western basin (Figure 8). In the vertical scale, the WSACW
also tends to dominate the upper layer of the South Atlantic above the ESACW (Figure 9).

**5    The Intermediate Layer**

The intermediate water masses have an origin in the upper 500m of the ocean and subduct into the
intermediate depth (1000 1500m) during their formation process. Similar to the Central

Waters, the distributions of the Intermediate Waters are significantly influenced by the major currents (Figure 10, left). The neutral density ($\sigma_\theta$)$\gamma$) of the Intermediate Waters is in general between 27.10 and 27.90

kg m$^{-3}$ and selected as the definition of Intermediate Layer.

In the Atlantic Ocean, two main intermediate water masses  Subarctic Intermediate Water (SAIW)  and Antarctic Intermediate Water (AAIW

), are found to be~~ are formed in the surface of sub-polar regions in north and south hemisphere respectively. In addition to AAIW and SAIW, Mediterranean

MW) is also considered as an intermediate water mass due to the similarity of density ranges, although the formation history is different (Figure 10).

**5.1     Antarctic Intermediate Water (AAIW)**

(McCartney, 1982, Alvarez et al., 2014).

(Piola & Gordon, 1989)

~~The AAIW covers most of the Atlantic Ocean until ~40 °N and the percentage shows a decrease trend~~

(Stramma &

England, 1999)

The
Antarctic Intermediate Water (AAIW) is the main Intermediate Water in the South Atlantic Ocean. This
water mass originates from the surface region north of the Antarctic Circumpolar Current (ACC) in all three sectors of the Southern Ocean, in particular in the area east of the Drake Passage in the Atlantic sector (McCartney, 1982; Alvarez et al., 2014), then subducts and spreads northward along the continental slope of South America (Piola and Gordon, 1989).

Based on the work by Stramma and England (1999) and Saenko and Weaver (2001), the region between 55 and 40 °S (east of the Drake Passage) at depths below 100 m is selected as the formation area of AAIW as well as the primary stage during the subduction and transformation (Figure 5). Previous work is considered to distinguish AAIW from surrounding water masses, including SACW in the north and NADW in the deep. Piola and Georgi (1982) and Talley (1996) define AAIW to haveas potential densities ($\sigma_\theta$) between 27.00-/27.10 and 27.40 kg/m³. On the other hand, m⁻³ and Stramma and England (1999) define the boundary between AAIW and SACW at $\sigma_\theta\sigma_\theta$ = 27.00 kg/m³ m⁻³ and the boundary between AAIW and NADW at $\sigma_1\sigma_1$ = 32.15 kg/m³. Furthermore, we used these m⁻³. The following criteria in our are used as the selection ofcriteria to define AAIW: potentialneutral density between 26.95 and 27.50 kg/m³ m⁻³ and pressure less thandepth between 100 and 300 barm. In addition, high oxygen (> 230260 µmol/ kg) as being important to distinguishe⁻¹) and low temperature (CT < 3.5 °C) are used to distinguish AAIW from Central Waters (WSACW and ESACW). The), while the relative high potential temperature (> -0.5 °C) and low silicate concentration (< 30 µmol/ kg) are⁻¹) of AAIW is an additional trade-marks of boundary to differentiate AAIW that we use to distinguish between AAIW andfrom AABW. As shown in Figure 8, (Table 2). The AAIW covers most of the AAIW samples haveAtlantic 
[revised manuscript text omitted]
 WS, 1974. No a Conservative Water-Mass Tracer. *Earth and Planetary Science Letters* **23**, 100-7.

Broecker WS, Denton GH, 1989. The Role of Ocean-Atmosphere Reorganizations in Glacial Cycles. *Geochimica Et Cosmochimica Acta* **53**, 2465-501.

Carracedo L, Pardo PC, Flecha S, Pérez FF, 2016. On the Mediterranean Water Composition. *Journal of Physical Oceanography* **46**, 1339-58.

Castro CG, Perez FF, Holley SE, Rios AF, 1998. Chemical characterisation and modelling of water masses in the Northeast Atlantic. *Progress in Oceanography* **41**, 249-79.

Cianca A, Santana R, Marrero J, Rueda M, Llinás O, 2009. Modal composition of the central water in the North Atlantic subtropical gyre. *Ocean Science Discussions* **6**, 2487-506.

Clarke RA, Gascard J-C, 1983. The Formation of Labrador Sea Water. Part I: Large-Scale Processes. *Journal of Physical Oceanography* **13**, 1764-78.

Defant A, 1929. *Dynamische Ozeanographie*. Springer.

Dengler M, Schott FA, Eden C, Brandt P, Fischer J, Zantopp RJ, 2004. Break-up of the Atlantic deep western boundary current into eddies at 8° S. *nature* **432**, 1018.

Deruijter W, 1982. Asymptotic Analysis of the Agulhas and Brazil Current Systems. *Journal of Physical Oceanography* **12**, 361-73.

Dickson RR, Brown J, 1994. The Production of North-Atlantic Deep-Water - Sources, Rates, and Pathways. *Journal of Geophysical Research-Oceans* **99**, 12319-41.

Elliot M, Labeyrie L, Duplessy JC, 2002. Changes in North Atlantic deep-water formation associated with the Dansgaard-Oeschger temperature oscillations (60-10 ka). *Quaternary Science Reviews* **21**, 1153-65.

Emery WJ, Meincke J, 1986. Global Water Masses - Summary and Review. *Oceanologica Acta* **9**, 383-91.

Foldvik A, Gammelsrod T, 1988. Notes on Southern-Ocean Hydrography, Sea-Ice and Bottom Water Formation. *Palaeogeography Palaeoclimatology Palaeoecology* **67**, 3-17.

Garcia-Ibanez MI, Pardo PC, Carracedo LI*, et al.*, 2015. Structure, transports and transformations of the water masses in the Atlantic Subpolar Gyre. *Progress in Oceanography* **135**, 18-36.

Gascard J-C, Clarke RA, 1983. The Formation of Labrador Sea Water. Part II. Mesoscale and Smaller-Scale Processes. *Journal of Physical Oceanography* **13**, 1779-97.

Gordon AL, 2001. Bottom Water Formation. In: Steele JH, ed. *Encyclopedia of Ocean Sciences.* Oxford: Academic Press, 334-40.

Gordon AL, Weiss RF, Smethie WM, Warner MJ, 1992. Thermocline and Intermediate Water Communication between the South-Atlantic and Indian Oceans. *Journal of Geophysical Research-Oceans* **97**, 7223-40.

Haine TWN, Hall TM, 2002. A generalized transport theory: Water-mass composition and age. *Journal*
*of Physical Oceanography* **32**, 1932-46.
Harvey J, 1982. Theta-S Relationships and Water Masses in the Eastern North-Atlantic. *Deep-Sea*
*Research Part a-Oceanographic Research Papers* **29**, 1021-33.
Helland-Hansen BR, 1916. Nogen hydrografiske metoder. *Scand. Naturforsker Mote. Kristiana. Oslo*.
Jullion L, Jacquet S, Tanhua T, 2017. Untangling biogeochemical processes from the impact of ocean
circulation: First insight on the Mediterranean dissolved barium dynamics. *Global Biogeochemical*
*Cycles* **31**, 1256-70.
Karstensen J, Tomczak M, 1997. Ventilation processes and water mass ages in the thermocline of the
southeast Indian Ocean. *Geophysical Research Letters* **24**, 2777-80.
Karstensen J, Tomczak M, 1998. Age determination of mixed water masses using CFC and oxygen
data. *Journal of Geophysical Research-Oceans* **103**, 18599-609.
Key RM, Kozyr A, Sabine CL*, et al.*, 2004. A global ocean carbon climatology: Results from Global Data
Analysis Project (GLODAP). *Global Biogeochemical Cycles* **18**.
Kieke D, Rhein M, Stramma L, Smethie WM, Bullister JL, Lebel DA, 2007. Changes in the pool of
Labrador Sea Water in the subpolar North Atlantic. *Geophysical Research Letters* **34**.
Kieke D, Rhein M, Stramma L, Smethie WM, Lebel DA, Zenk W, 2006. Changes in the CFC inventories
and formation rates of Upper Labrador Sea Water, 1997-2001. *Journal of Physical Oceanography* **36**,
64-86.
Kirchner K, Rhein M, Huttl-Kabus S, Boning CW, 2009. On the spreading of South Atlantic Water into
the Northern Hemisphere. *Journal of Geophysical Research-Oceans* **114**.
Kissel C, Laj C, Lehman B, Labyrie L, Bout-Roumazeilles V, 1997. Changes in the strength of the
Iceland-Scotland Overflow Water in the last 200,000 years: Evidence from magnetic anisotropy
analysis of core SU90-33. *Earth and Planetary Science Letters* **152**, 25-36.
Klein B, Hogg N, 1996. On the variability of 18 Degree Water formation as observed from moored
instruments at 55 degrees W. *Deep-Sea Research Part I-Oceanographic Research Papers* **43**, 1777-&.
Kuhlbrodt T, Griesel A, Montoya M, Levermann A, Hofmann M, Rahmstorf S, 2007. On the driving
processes of the Atlantic meridional overturning circulation. *Reviews of Geophysics* **45**.
Lauvset SK, Key RM, Olsen A*, et al.*, 2016. A new global interior ocean mapped climatology: the
1°× 1° GLODAP version 2. *Earth Syst. Sci. Data* **8**, 325-40.
Lazier JRN, Wright DG, 1993. Annual Velocity Variations in the Labrador Current. *Journal of Physical*
*Oceanography* **23**, 659-78.
Lozier MS, 2012. Overturning in the North Atlantic. *Ann Rev Mar Sci* **4**, 291-315.
Lutjeharms JR, Van Ballegooyen RC, 1988. Anomalous upstream retroflection in the agulhas current.
*Science* **240**, 1770.
Lynch-Stieglitz J, Adkins JF, Curry WB*, et al.*, 2007. Atlantic meridional overturning circulation during
the Last Glacial Maximum. *Science* **316**, 66-9.
Mccartney MS, 1982. The subtropical recirculation of Mode Waters. *Journal of Marine Research* **40**,
427-64.
Mccartney MS, Talley LD, 1982. The subpolar mode water of the North Atlantic Ocean. *Journal of*
*Physical Oceanography* **12**, 1169-88.
Montgomery RB, 1958. Water characteristics of Atlantic Ocean and of world ocean. *Deep Sea*
*Research (1953)* **5**, 134-48.
Olsen A, Key RM, Heuven SV*, et al.*, 2016. The Global Ocean Data Analysis Project version 2
(GLODAPv2) - an internally consistent data product for the world ocean. *Earth System Science Data*
**71**, 300-8.
Orsi AH, Johnson GC, Bullister JL, 1999. Circulation, mixing, and production of Antarctic Bottom
Water. *Progress in Oceanography* **43**, 55-109.
Peterson RG, Stramma L, 1991. Upper-Level Circulation in the South-Atlantic Ocean. *Progress in*
*Oceanography* **26**, 1-73.
Pickart RS, Spall MA, Lazier JRN, 1997. Mid-depth ventilation in the western boundary current system
of the sub-polar gyre. *Deep-Sea Research Part I-Oceanographic Research Papers* **44**, 1025-+.

Piola AR, Georgi DT, 1982. Circumpolar properties of Antarctic intermediate water and Subantarctic
Mode Water. *Deep Sea Research Part A. Oceanographic Research Papers* **29**, 687-711.
Piola AR, Gordon AL, 1989. Intermediate Waters in the Southwest South-Atlantic. *Deep-Sea Research
Part a-Oceanographic Research Papers* **36**, 1-16.
Pollard RT, Griffiths MJ, Cunningham SA, Read JF, Perez FF, Rios AF, 1996. Vivaldi 1991-A study of the
formation, circulation and ventilation of Eastern North Atlantic Central Water. *Progress in
Oceanography* **37**, 167-92.
Pollard RT, Pu S, 1985. Structure and Circulation of the Upper Atlantic Ocean Northeast of the Azores.
*Progress in Oceanography* **14**, 443-62.
Poole R, Tomczak M, 1999. Optimum multiparameter analysis of the water mass structure in the
Atlantic Ocean thermocline. *Deep-Sea Research Part I-Oceanographic Research Papers* **46**, 1895-921.
Price JF, Baringer MO, Lueck RG*, et al.*, 1993. Mediterranean outflow mixing and dynamics. *Science*
**259**, 1277-82.
Prieto E, Gonzalez-Pola C, Lavin A, Holliday NP, 2015. Interannual variability of the northwestern
Iberia deep ocean: Response to large-scale North Atlantic forcing. *Journal of Geophysical Research-
Oceans* **120**, 832-47.
Read J, 2000. CONVEX-91: water masses and circulation of the Northeast Atlantic subpolar gyre.
*Progress in Oceanography* **48**, 461-510.
Reid JL, 1978. On the middepth circulation and salinity field in the North Atlantic Ocean. *Journal of
Geophysical Research: Oceans* **83**, 5063-7.
Reid JL, 1979. On the contribution of the Mediterranean Sea outflow to the Norwegian-Greenland
Sea. *Deep Sea Research Part A. Oceanographic Research Papers* **26**, 1199-223.
Rhein M, Kieke D, Huttl-Kabus S*, et al.*, 2011. Deep water formation, the subpolar gyre, and the
meridional overturning circulation in the subpolar North Atlantic. *Deep-Sea Research Part Ii-Topical
Studies in Oceanography* **58**, 1819-32.
Rhein M, Stramma L, Krahmann G, 1998. The spreading of Antarctic bottom water in the tropical
Atlantic. *Deep-Sea Research Part I-Oceanographic Research Papers* **45**, 507-27.
Smethie WM, Fine RA, 2001. Rates of North Atlantic Deep Water formation calculated from
chlorofluorocarbon inventories. *Deep-Sea Research Part I-Oceanographic Research Papers* **48**, 189-
215.
Smith EH, Soule FM, Mosby O, 1937. *The Marion and General Greene Expeditions to Davis Strait and
Labrador Sea, Under Direction of the United States Coast Guard: 1928-1931-1933-1934-1935:
Scientific Results, Part 2: Physical Oceanography*. US Government Printing Office.
Stramma L, England MH, 1999. On the water masses and mean circulation of the South Atlantic
Ocean. *Journal of Geophysical Research-Oceans* **104**, 20863-83.
Stramma L, Kieke D, Rhein M, Schott F, Yashayaev I, Koltermann KP, 2004. Deep water changes at the
western boundary of the subpolar North Atlantic during 1996 to 2001. *Deep Sea Research Part I:
Oceanographic Research Papers* **51**, 1033-56.
Stramma L, Peterson RG, 1990. The South-Atlantic Current. *Journal of Physical Oceanography* **20**,
846-59.
Sverdrup, 1942. The Oceans: Their Physics, Chemistry and General Biology.
Swift JH, 1984. The circulation of the Denmark Strait and Iceland Scotland overflow waters in the
North-Atlantic. *Deep-Sea Research Part a-Oceanographic Research Papers* **31**, 1339-55.
Swift SM, 1980. Activity patterns of pipistrelle bats (Pipistrellus pipistrellus) in north‐east Scotland.
*Journal of Zoology* **190**, 285-95.
Talley L, 1996. Antarctic intermediate water in the South Atlantic. In. *The South Atlantic.* Springer,
219-38.
Talley L, Raymer M, 1982. Eighteen degree water variability. *J. Mar. Res* **40**, 757-75.
Talley LD, Mccartney MS, 1982. Distribution and Circulation of Labrador Sea-Water. *Journal of
Physical Oceanography* **12**, 1189-205.
Tanhua T, Olsson KA, Jeansson E, 2005. Formation of Denmark Strait overflow water and its hydro-
chemical composition. *Journal of Marine Systems* **57**, 264-88.

Tomczak M, 1981. A multi-parameter extension of temperature/salinity diagram techniques for the
analysis of non-isopycnal mixing. *Progress in Oceanography* **10**, 147-71.
Tomczak M, 1999. Some historical, theoretical and applied aspects of quantitative water mass
analysis. *Journal of Marine Research* **57**, 275-303.
Tomczak M, Large DG, 1989. Optimum multiparameter analysis of mixing in the thermocline of the
eastern Indian Ocean. *Journal of Geophysical Research: Oceans* **94**, 16141-9.
Van Heuven SMaC, Hoppema M, Huhn O, Slagter HA, De Baar HJW, 2011. Direct observation of
increasing CO2 in the Weddell Gyre along the Prime Meridian during 1973–2008. *Deep Sea Research*
*Part II: Topical Studies in Oceanography* **58**, 2613-35.
Weiss RF, Ostlund HG, Craig H, 1979. Geochemical Studies of the Weddell Sea. *Deep-Sea Research*
*Part a-Oceanographic Research Papers* **26**, 1093-120.
Worthington L, 1959. The 18 water in the Sargasso Sea. *Deep Sea Research (1953)* **5**, 297-305.
Wüst G, Defant A, 1936. *Atlas zur Schichtung und Zirkulation des Atlantischen Ozeans: Schnitte und*
*Karten von Temperatur, Salzgehalt und Dichte*. W. de Gruyter.

Zou S.J., Bower A., Furey H., Lozier M.S., Xu X.B.: Redrawing the Iceland-Scotland Overflow Water
pathways in the North Atlantic. *Nature Communications* 11, 2020.

---

## Author Response (AR3)

**Reply to the reviewers comments to the manuscript "Water Masses in the Atlantic Ocean: Characteristics and Distributions" by Mian Liu and Toste Tanhua**

Kiel, December 8, 2020

We thank the reviewer for the patience with us and for the helpful comments to improve the manuscript.

I did yet another thorough read-through of the manuscript and corrected the manuscript for language and style. No changes to the content was done during this step, and all changes are available in the track-changes mode.

Comment made by the reviewer in the pdf:
Page 7. The reason this happens is that the individual sample properties outside the input SWTs to the OMP formulation. This is now explained in the text, and is eluted to in more detail in the same section.

Page 8: Corrected the double up of SA, and went through the manuscript to be consequent with the use of SA vs. S, and CT vs T. when appropriate (for instance not when we refer to a previous work that use different units). We also checked the capitalization of Absolute Salinity and Conservative Temperature

Comments to the manuscript:
We highly appreciate the insight on the more general expressions for this kind of inversions. We have indeed not done a comprehensive study of the effect of any errors in the prior assumptions. We do touch on it when we discuss the general approach taken making this study useful for the basin-scale distribution of water masses. That is, we do not take into account temporally changing properties of the SWTs, for instance.
To make this clear we have added the following sentence on page 6, end of section 2.3.1. "The solution is dependent on, and sensitive to, the prior assumptions of the properties of the SWTs. Here we have not explicitly explored this sensitivity, but note that a common difficulty in OMP analysis is to properly define the SWT properties, and that this study provide a generally applicable set of SWT properties for the major water masses in the Atlantic Ocean."

---

## Author Response (AR4)

Kiel December 30, 2020

Dear Editor,

I apologize that I did not upload the track-changes file in the previous submission.

Note that there are a few modifications to the file I uploaded on December 8, I did change the reference to figures in the supplementary material to be "Figure S1" etc. and I added a reference to a data file that we would like to add as a supplementary material. (The new file with those additions, but without track changes is also uploaded.

The data file has been requested from a few colleges that are aware of this submission, and will be a file of the GLODAPv2 Atlantic data with the water mass fractions of the water masses we discuss in this paper. We think that this will be a useful complement to this article.

In addition, I found that by mistake Figure S1 was already present as a figure in the main body of the article, so I removed that and renumbered the figures accordingly.

The first author of the paper, Mian Liu, is currently on a long sea going expedition that have made exchange of information and sending of figures complicated and slow.

With best regards,
Toste Tanhua

---

## Author Response (AR5)

Dear Editor of Ocean Science,

We thank you very much for your patience with us, and for the careful read of the manuscript and the suggested edits.

We have now attended to all those edits.

In addition we added a link to a repository at NOAA NCEI with the data of the calculated water mass fractions in the Atlantic. This is the same repository as where the GLODAP data product is stored.

With best regards,

Toste Tanhua and Mian Liu